# Impact of Intercontinental Pollution Transport on North American Ozone Air Pollution: An HTAP Phase 2 Multi-model Study

Min Huang[1,2], Gregory R. Carmichael[3], R. Bradley Pierce[4], Duseong S. Jo[5], Rokjin J. Park[5], Johannes Flemming[6], Louisa K. Emmons[7], Kevin W. Bowman[8], Daven K. Henze[9], Yanko Davila[9], Kengo Sudo[10], Jan Eiof Jonson[11], Marianne Tronstad Lund[12], Greet Janssens-Maenhout[13], Frank J. Dentener[13], Terry J. Keating[14], Hilke Oetjen[8,*], Vivienne H. Payne[8]

[1]George Mason University, Fairfax, VA, USA
[2]University of Maryland, College Park, MD, USA
[3]University of Iowa, Iowa City, IA, USA
[4]NOAA National Environmental Satellite, Data, and Information Service, Madison, WI, USA
[5]Seoul National University, Seoul, Korea
[6]European Center for Medium range Weather Forecasting, Reading, UK
[7]National Center for Atmospheric Research, Boulder, CO, USA
[8]Jet Propulsion Laboratory, California Institute of Technology, Pasadena, CA, USA
[9]University of Colorado-Boulder, Boulder, CO, USA
[10]Nagoya University, Furocho, Chigusa-ku, Nagoya, Japan
[11]Norwegian Meteorological Institute, Oslo, Norway
[12]Center for International Climate and Environmental Research, Oslo, Norway
[13]European Commission, Joint Research Center, Ispra, Italy
[14]US Environmental Protection Agency, Washington, DC, USA
[*]Now at: University of Leicester, Leicester, UK

*Correspondence to*: Min Huang (mhuang10@gmu.edu)

**Abstract**

The recent update on the US National Ambient Air Quality Standards of the ground-level ozone ($O_3$) can benefit from a better understanding of its source contributions in different US regions during recent years. In the Hemispheric Transport of Air Pollution experiment Phase 1 (HTAP1), various global models were used to determine the $O_3$ source-receptor relationships among three continents in the Northern Hemisphere in 2001. In support of the HTAP Phase 2 (HTAP2) experiment that studies more recent years and involves higher-resolution global models and regional models' participation, we conduct a number of regional scale Sulfur Transport and dEposition Model (STEM) air quality base and sensitivity simulations over North America during May-June 2010. STEM's top and lateral chemical boundary conditions were downscaled from three global chemical transport models' (i.e., GEOS-Chem, RAQMS, and ECMWF C-IFS) base and sensitivity simulations in which the East Asian (EAS) anthropogenic emissions were reduced by 20%. The mean differences between STEM surface $O_3$ sensitivities to the emission changes and its corresponding boundary condition model's are smaller than those among its boundary condition models, in terms of the regional/period mean (<10%) and the spatial distributions. An additional STEM simulation was performed in which the boundary conditions were downscaled from a RAQMS simulation without EAS anthropogenic emissions. The scalability of $O_3$ sensitivities to the size of the emission perturbation is spatially varying, and the full (i.e., based on 100% emission perturbation) source contribution obtained from linearly scaling the North American mean $O_3$ sensitivities to a 20% reduction in the EAS anthropogenic emissions may be underestimated by at least 10%. The three boundary condition models' mean $O_3$ sensitivities to the 20% EAS emission perturbations are ~8% (May-June 2010)/~11% (2010 annual) lower than those estimated by eight global models, and the multi-model ensemble estimates are higher than the HTAP1 reported 2001 conditions. GEOS-Chem sensitivities indicate that the EAS anthropogenic $NO_x$ emissions matter more than the other EAS $O_3$ precursors to the North American $O_3$, qualitatively consistent with previous adjoint sensitivity calculations.

In addition to the analyses on large spatial/temporal scales relative to the HTAP1, we also show results on subcontinental- and event-scale that are more relevant to the US air quality management. The EAS pollution impacts are weaker during observed $O_3$ exceedances than on all days in most US regions except over some high terrain western US rural/remote areas. Satellite $O_3$ (TES, JPL-IASI, and AIRS) and carbon monoxide (TES and AIRS) products, along with surface measurements and model calculations, show that during certain episodes stratospheric $O_3$ intrusions and the transported EAS pollution influenced $O_3$ in the western and the eastern US differently. Free-running (i.e., without chemical data assimilation) global models underpredicted the transported background $O_3$ during these episodes, posing difficulties for STEM to accurately simulate the surface $O_3$ and its source contribution. Although we effectively improved the modeled $O_3$ by incorporating satellite $O_3$ (OMI and MLS) and evaluated the quality of the HTAP2 emission inventory with the KNMI OMI nitrogen dioxide, using observations to evaluate and improve $O_3$ source attribution still remains to be further explored.

## 1.      Introduction

Tropospheric ozone ($O_3$), a short-lived trace gas with a lifetime ranging from hours in the boundary layer to weeks in the free troposphere, affects tropospheric chemistry, harms human and ecosystem health, and induces climate change on local, regional and global scales (Jerrett et al., 2009; Smith et al., 2009; Anenberg et al., 2010; Mauzerall and Wang, 2001; Avnery et al., 2011a, b; Shindell et al., 2009, 2013; Bowman and Henze, 2012; Stevenson et al., 2006, 2013; Monks et al., 2015). It has been recognized that the uneven distributions of tropospheric $O_3$ can be attributed to the stratosphere as well as local, regional and distant emission sources, through complicated processes that occur on synoptic, meso- and micro-scales (Task Force on Hemispheric Transport of Air Pollution (HTAP), 2010; National Research Council (NRC), 2009; Maas and Grennfelt, 2016). The mitigation of $O_3$'s climate and health impacts would benefit from efforts to control the emissions of its precursors from the various emission sources (United Nations Environment Programme (UNEP) and World Meteorological Organization (WMO), 2011), such as nitrogen oxides ($NO_x$), carbon monoxide (CO), methane ($CH_4$), and non-methane volatile organic compounds (NMVOCs).

Ground-level $O_3$ is one of the six criteria air pollutants regulated by the US Environmental Protection Agency (EPA), and the US National Ambient Air Quality Standards (NAAQS) has recently been lowered to 70 ppbv to better protect Americans' health and the environment. Issues regarding making accurate estimates of the total $O_3$ as well as the background $O_3$ level (defined as the concentration that is not affected by recent locally-emitted or produced anthropogenic pollution) (e.g., McDonald-Buller et al., 2011; Zhang et al., 2011; Fiore et al., 2014; Huang et al., 2015), have been recently discussed as part of the implementation of the new US $O_3$ standard (US EPA, 2016a, b). This includes assessing the impacts of various components of the background $O_3$, such as stratospheric $O_3$, local natural sources such as biogenic, lightning and wildfire emissions, as well as the long-range transport (LRT) of pollution. The impact of the trans-Pacific pollution transport on US air quality has been evaluated in numerous studies over the past decades (e.g., Fiore et al., 2009; Reidmiller et al., 2009; Zhang et al., 2008, 2009; Huang et al., 2010, 2013a; Lin et al., 2012a, 2015, 2016; US EPA, 2016a). It has been found that the increasing trends of pollution in the upwind continents, especially the populated East Asia (e.g., Zhang et al., 2014; Susaya et al., 2013; Wang et al., 2012), may partially offset the US air quality improvements in recent decades due to the regional and local emission controls (e.g., Jacob et al., 1999; Verstraeten et al., 2015; Ambrose et al., 2011; Wigder et al., 2013; Cooper et al., 2010; Parrish et al., 2009, 2012; Gratz et al., 2014). A better understanding of the processes that determine the $O_3$ pollution levels, as well as an improved capability of attributing the air pollution to nearby or distant sources is needed to assist with designing and implementing effective local emission control strategies to comply with the tighter air quality standards.

Chemical transport models are often used to reproduce and attribute the observed $O_3$ levels, including assessing the impacts of the internationally transported $O_3$ on the US air quality. In the HTAP modeling experiment Phase 1 (HTAP1), various global models with horizontal resolutions ranging from $1° \times 1°$ to $5° \times 5°$, only around half of which are finer than $3° \times 3°$, were used to determine the $O_3$ source-receptor (SR) relationships among three continents in the Northern Hemisphere in 2001 (Chapter 4 in HTAP, 2010). The global model based SR relationships in HTAP1 determined using the emission perturbation approach (i.e., calculating the changes of $O_3$

at the receptor regions in response to a 20% reduction in the emission inputs in a given source
region) were reported as either monthly 24h mean values or policy-relevant metrics such as the
maximum daily 8h average (MDA8) for the US (e.g., Fiore et al., 2009; Reidmiller et al., 2009).
Large intermodel diversity was found in the simulated total $O_3$ and the intercontinentally
transported pollution for the chosen SR pairs in the northern midlatitudes, indicating the challenges
with model simulations to accurately represent the key atmospheric processes. Multi-model mean
results were the foci of in these studies with the assumption that this approach can reduce the
uncertainty from the single model estimates for monthly or seasonal means. "Ensemble" model
analyses have been suggested by some US stakeholders as one of the methods for helping with the
characterization of the background $O_3$ components (US EPA, 2016b). Although the multi-model
approach can help identify some of the weaknesses of the individual models and may produce
more reliable estimates, it is necessary to well understand the uncertainties inherent in using the
same set of anthropogenic emissions in all these model simulations. Satellite observations over the
regions with limited in-situ measurements such as the East Asia can be particularly helpful for
quantifying such uncertainties.

The 20% emission perturbation in the HTAP1 modeling experiment was chosen to produce
a sizeable (i.e., larger than numerical noise) and realistic impact, but small enough in the assumed
near-linear atmospheric chemistry regime. The scalability of the modeled $O_3$ sensitivities to the
size of the emission perturbations has been assessed on continental scale (Wu et al., 2009; Fiore et
al., 2009; HTAP, 2010; Wild et al., 2012; Emmons et al., 2012). The receptor $O_3$ responses to the
source-region emission perturbations are found to be fairly linear within ~50% of the perturbations.
However, due to the chemical non-linearity, the full source contribution obtained by linearly
scaling the receptor regional mean $O_3$ sensitivity to the 20% reduction in the source region
emissions may be underestimated, and the scalability depended on seasons and the perturbed
emission species. Huang et al. (2013b) investigated the scalability of the $O_3$ sensitivity between
the southern California-US intermountain west SR pair for May 2010, in which study the southern
California anthropogenic emissions were perturbed by multiple amounts of +50%, -50%, -100%.
They reported that the scalability of the $O_3$ sensitivities changed with the distance from the source
regions. Further analyses on the scalability of these modeled $O_3$ sensitivities during recent years
especially for the East Asia-NAM SR pair, as well as their spatial variability, are still needed.
Furthermore, results generated using the emission perturbation approach need to be compared with
those based on the other methods (e.g., tagged tracers, adjoint sensitivity).

Previous studies have demonstrated the advantages of high resolution chemical transport
modeling for understanding SR relationships (e.g., Lin et al., 2010 for Europe and the East Asia;
Lin et al., 2012a; Huang et al., 2010, 2013a for Asia and NAM). Using observations (satellite,
sondes, aircraft) along with single model simulations, a few studies have reported that the US $O_3$
sensitivities to extra-regional sources is time- and region-dependent (e.g., Lin et al., 2012a, b;
Langford et al., 2011; Ott et al., 2016), and therefore the necessity of evaluating the extra-regional
source impacts on event scale has been emphasized in these studies as well as in US EPA (2016a,
b). The HTAP Phase 2 (HTAP2) multi-model experiment, initiated in 2012, is designed to advance
the understanding of the impact of intercontinental pollution transport during more recent years
(i.e., 2008-2010) involving a number of global and regional models' participation (Galmarini et
al., 2017; Koffi et al., 2016). The regional models are anticipated to help connect the analyses over
global and regional scales and enable discussions on small spatial (e.g., subcontinental) and

temporal scales (i.e., event based analyses). The use of satellite products for identifying the transport events as well as for quantitative model evaluation is also encouraged in the work plan. The HTAP2 modeling experiment was sequentially conducted in two steps. First, similar to the HTAP1 experiment, a group of global models with different resolutions conducted base and emission perturbation sensitivity simulations to determine the pollutants' SR relationships. All models in their base simulations used the same set of harmonized sector-based global anthropogenic emissions developed specifically for the HTAP2 modeling experiment (Janssens-Maenhout et al., 2015). Most of these global models recorded only key chemical species from their base and sensitivity simulations in varied temporal frequencies. Several global models saved the three-dimensional (3D) chemical fields of more species with a 3- or 6-hour interval, which are suitable for being used as regional models' chemical boundary conditions. In the second step, regional models conducted base and sensitivity simulations to analyze the pollutants' SR relationships in greater detail. The regional model simulations used the same set of anthropogenic emissions as the global models within their simulation domains, and the chemical boundary conditions in these regional simulations were downscaled from the base and sensitivity simulations from the selected boundary condition model outputs. For regional simulations over the North America and Europe, boundary conditions were mostly taken from a single model such as the ECMWF C-IFS or GEOS-Chem.

This study aims to address: 1) comparing the differences in $O_3$ sensitivities generated from the HTAP2 and HTAP1 experiments, which could help address how the LRT impacts on NAM changed through time; 2) how the refined modeling experiment design in HTAP2 can help advance our understanding of the LRT impacts on NAM, particularly the involvement of regional models and the inclusion of small spatial/temporal scale analysis during high $O_3$ episodes that are more relevant to air quality management;  3) the usefulness of satellite observations for better understanding the sources of uncertainties in the modeled total $O_3$ (e.g., from the emission and regional models' boundary condition inputs) as well as for reducing the uncertainties in some of these model inputs via chemical data assimilation. We performed a number of regional scale STEM (Sulfur Transport and dEposition Model) base and sensitivity simulations over the NAM during May-June 2010, during which period strong trans-Pacific pollution transport were shown to episodically impact the US (Lin et al., 2012a). Extending the HTAP2 regional simulations' basic setup, the STEM top and lateral chemical boundary conditions were downscaled from three global models' (i.e., the Seoul National University (SNU) GEOS-Chem, RAQMS, and the ECMWF C-IFS) base and sensitivity simulations in which the East Asian anthropogenic emissions were reduced. The STEM surface $O_3$ sensitivities over the NAM region based on different boundary condition models were inter-compared, in terms of the regional averages and the spatial patterns on monthly basis and during a selected event identified by satellite $O_3$ and CO products. These were also compared with the sensitivities estimated by their corresponding boundary condition models as well as all HTAP2 participating global models and the results from HTAP1.

## 2.    Methods

### 2.1.    Anthropogenic emission inputs

Identical anthropogenic emissions were used in all global and regional chemical transport models' base and sensitivity simulations. This monthly-varying harmonized sectoral (i.e., power, industry, transportation, residential, shipping, aircraft, agriculture) emission inventory was

provided on a gridded 0.1°×0.1° resolution for the years of 2008 and 2010, by compiling the
officially reported emissions at the national scale (Janssens-Maenhout et al., 2015;
http://edgar.jrc.ec.europa.eu/htap_v2). The temporal profiles for developing the monthly-varying
emissions differ by region and sector. The amount of emissions of key $O_3$ precursors (CO, $NO_x$,
NMVOCs) from both years are summarized in Table S1 for the four major emissions sectors, over
the NAM (US+Canada, based on data from the US EPA and the Environmental Canada, which
shows lower emissions from the previous years as also discussed in Pouliot et al., 2015), MICS-
Asia regions (south, southeast, and east Asia, based on country inventory for China and from the
Clean Air Policy Support System and the Regional Emission inventory in ASia 2.1, more
information also in Li et al., 2017), and for over the world. For all of these species, global total
emissions in 2008 and 2010 are similar. The $NO_x$, NMVOC, and CO emissions decreased from
2008 to 2010 over the NAM by 10.7%, 9.4%, and 15.7%, respectively. In 2008, NAM $NO_x$,
NMVOC and CO contributed to 18.0%, 11.7% and 11.9% of the global total, respectively, and in
2010, these contributions became 15.8%, 10.5% and 10.2%. For 2010, the transportation sector
contributed more than the other sectors to NAM anthropogenic $NO_x$ and CO emissions; industrial
sector contributed more than the other sectors to NMVOCs emissions. Over East Asian countries,
these emissions are ~2-5 times higher than the US emissions, and the $NO_x$, NMVOC and CO
emissions increased over Asia by 7.3%, 7.2% and 1.0%, with the dominant emission sectors in
2010 of transportation, industry, and residential, respectively. For both years, the emissions over
the MICS-Asia regions contribute to over 40% of the global emissions. For these key $O_3$ precursors,
the East Asian countries contribute to 45% (NMVOCs)-70% ($NO_x$) of the emissions in the MICS-
Asia domain in both years, and the south Asian countries contribute to ~22% ($NO_x$)-34%
(NMVOCs) of the MICS-Asia emissions. The uncertainty of the emission estimates differs by
emission sector and species: i.e., the emissions from large-scale combustion sources (e.g., $NO_x$
and CO from power and industry sectors) are less uncertain than those from small-scale and
scattered sources (e.g., CO and NMVOCs from transportation and residential sources). Non-
anthropogenic emission inputs used in different models' simulations may differ, and their impacts
on the modeled total $O_3$ and the SR relationships will be compared in detail in future studies.
*2.2.    Region definitions for the SR study and the model base and sensitivity simulations*
2.2.1.   Base and 20% emission perturbation simulations from global and regional models
The HTAP2 simulations from eight global models, used in this study, are listed in Table
1a, including the relevant references. Horizontal and vertical resolutions of these models range
from finer than 1° to coarser than 2.5°, and from 20 to 60 layers, respectively. Overall these
resolutions are higher than the HTAP1 participating models'. Figure 1 defines the source regions
used in the HTAP2 SR relationship study and we will focus in this study on assessing the East
Asia (EAS), S Asia (SAS), Europe (EUR), and non-NAM anthropogenic source (interchangeable
in this paper with "(all) foreign") impacts on the NAM $O_3$ levels in 2010. Specifically, each model
performed a base simulation and a number of sensitivity simulations in which the original HTAP2
anthropogenic emissions for all species and sectors in a defined source region were perturbed by
a certain amount (referring to 20% as in most cases) and these cases are defined in Table 1a-b as
*source region*ALL(*perturbation*), where "ALL" refers to "all species and sectors", consistent
with HTAP1 and HTAP2's naming convention. The $O_3$ differences R($O_3$, *source region*,
*perturbation*) over the NAM were then calculated between each model's base and sensitivity
simulations:

250 R(O$_3$, EAS, 20%) = BASE O$_3$ - EASALL(-20%) O$_3$   (1a)
251 R(O$_3$, SAS, 20%) = BASE O$_3$ - SASALL(-20%) O$_3$   (1b)
252 R(O$_3$, EUR, 20%) = BASE O$_3$ - EURALL(-20%) O$_3$   (1c)
253 R(O$_3$, non-NAM, 20%) = NAMALL(-20%) O$_3$ - GLOALL(-20%) O$_3$   (1d)
254 Where "GLO" stands for the "global" source region.

256   The monthly-mean R(O$_3$, *source region*, 20%) values were averaged over the NAM
257 region for the analysis and compared with the findings in the HTAP1 study (e.g., Fiore et al., 2009).
258 It is worth mentioning that the rectangular source regions defined in HTAP1 were modified in
259 HTAP2 to align with the geo-political borders. For EAS and SAS, the regions not overlapped by
260 HTAP1 and HTAP2 are mostly in the less populated/polluted regions such as the northwestern
261 China, according to the HTAP2 emission maps (http://edgar.jrc.ec.europa.eu/htap_v2/index.php).
262 HTAP2's EUR domain excludes certain regions in Russia/Belarussia/Ukraine, Middle East and
263 North Africa that are included in HTAP1's EUR domain. The impact of emissions over these
264 regions on comparing the NAM R(O$_3$, EUR, 20%) values in HTAP1 and HTAP2 will be discussed
265 in Section 3.2.1.

267   A unitless "Response to Extra-Regional Emission Reductions (RERER)" metric
268 (Galmarini et al., 2017), as defined in eq. (2), was also calculated to measure the importance of
269 local versus non-local sources to NAM's O$_3$ levels:

270 $$\text{RERER (O}_3\text{, NAM)} = \frac{R(o3, non-NAM, 20\%)}{R(o3, global, 20\%)} = \frac{(NAMALL(-20\%)\ O3 - GLOALL(-20\%)\ O3)}{(BASE\ O3 - GLOALL(-20\%)\ O3)}$$ (2)

271 The denominator and numerator terms of RERER represent the impacts of global and non-NAM
272 anthropogenic emissions on NAM O$_3$, respectively. The higher the NAM RERER value is, the
273 stronger impact from non-local sources on NAM is indicated. The RERER value can exceed 1,
274 when emission reductions led to increasing concentrations (e.g. O$_3$ titration by nitrogen monoxide
275 (NO)).

277   The STEM (version 2K3) regional simulations were then performed on a 60 km×60 km
278 horizontal resolution (a typical coarse regional model resolution) grid over NAM within the
279 domain defined in Figure 2a during May-June 2010. The meteorological conditions in spring 2010
280 were compared with the climatology from the NCEP/NCAR reanalysis data for the 1981-2010
281 period (Kalnay et al., 1996) in Huang et al. (2013b), concluding that this spring represents a period
282 of stronger-than-climatological average spring trans-Pacific transport, based on a stronger
283 meridional gradient in the North Pacific and higher Pacific/North American (PNA) indexes. This
284 is consistent with the findings by Lin et al. (2014) that the El Niño conditions during the 09/10
285 winter strengthened the trans-Pacific transport of Asian pollution in spring 2010. The mean near-
286 surface air temperatures in the western US in this spring were lower than the climatology, with
287 larger anomalies in the mountain states, which may have led to weaker local O$_3$ production and
288 decomposition of the transported peroxyacyl nitrates (PAN). In contrast, higher-than-normal
289 temperatures were found in the eastern US that favored anomalously strong local O$_3$ production.

291   STEM has been used to interpret the observations collected by satellites and during aircraft
292 campaigns in the past decade (e.g., Carmichael et al., 2003a, b; Huang et al., 2010, 2013a, b, 2014,
293 2015). STEM calculates gas-phase chemistry reactions based on the SAPRC 99 gaseous chemical
294 mechanism (Carter, 2000) with thirty photolysis rates calculated online by the Tropospheric
295 Ultraviolet-Visible radiation model (Madronich et al., 2002). Most of the key configurations of the

60 km base simulations are the same as those described in Lapina et al. (2014), i.e., meteorological
fields were pre-calculated by the Advanced Research Weather Research and Forecasting Model
(WRF-ARW, Skamarock et al., 2008) version 3.3.1 forced by the North American Regional
Reanalysis data (Mesinger et al., 2006), using a similar set of the physics configuration to those in
Huang et al. (2013a). Biomass burning emissions are from the Fire INventory from NCAR (FINN)
inventory version 1.0 (Wiedinmyer et al., 2011). Biogenic emissions were calculated by the Model
of Emissions of Gases and Aerosols from Nature (MEGAN) version 2.1 (Guenther et al., 2012),
driven by the WRF meteorology. Lightning $NO_x$ emissions are generated following the method in
Allen et al. (2012), with the flash rates determined by the WRF convective precipitation and scaled
to the National Lightning Detection Network flash rates. A major difference of the STEM
simulations in this study from the Lapina (2014) study is that the anthropogenic emissions were
replaced with the monthly-mean HTAP2 inventory with no weekday-weekend variability applied,
rather than the earlier National Emission Inventory (NEI) 2005 in which the weekday-weekend
variability exists. This change can introduce uncertainty for some US regions where weekday-
weekend variability of some $O_3$ precursors' emissions was notable during the studied period (e.g.,
weekend $NO_x$ emissions in southern California during spring/summer 2010 were 0.6-0.7 of the
weekday emissions as reported by Kim et al. (2016) and Brioude et al. (2013)), but this was done
to ensure consistency with the HTAP2 global model simulations, that also didn't use daily variable
emissions for any regions in the world. The VOC speciation for the SPRAC 99 chemical
mechanism in the NEI 2005 (ftp://aftp.fsl.noaa.gov/divisions/taq/emissions_data_2005) were
applied to break down the total NMVOC emissions provided in the HTAP2 inventory. The VOC
speciation based on the year of 2005 can be unrealistic for 2005 as well as 2010 as studies have
reported variable temporal changes of different VOC species in some US cities (e.g., Warneke et
al., 2012). The time-varying lateral and top boundary conditions in the STEM base simulations
were downscaled from three global models (i.e., 3 hourly SNU GEOS-Chem, 3 hourly ECMWF
C-IFS, and 6 hourly RAQMS) base simulations. In support of the SR relationship study to quantify
the East Asia anthropogenic impacts on the NAM, three STEM sensitivity simulations were also
conducted in which the STEM boundary conditions were downscaled from the EASALL(-20%)
sensitivity simulations by these three global models (Table 1b). All STEM simulated 3D chemical
fields were saved hourly for the convenience of calculating the US primary $O_3$ standard metric
MDA8 as well as the quantitative comparisons against the satellite Level 2 (L2) $O_3$ products. The
STEM base case surface $O_3$ performance and its $O_3$ sensitivities were also compared with those of
its boundary condition models as well as the multi- global model means. The latitude/longitude
ranges (20-50°N/130-65°W) of NAM for the global and regional model based sensitivity
calculations were selected to mainly account for the coverage of the STEM domain, which are
slightly different from the definition of North America in HTAP1.
Note that non-anthropogenic emission inputs used in STEM and its boundary condition
models differed, as summarized in Table 1c. Figure S1 shows detailed comparisons between
STEM and GEOS-Chem's non-anthropogenic (i.e., soil, lightning, biomass burning) $NO_x$
emission inputs, and their impacts on the modeled NAM background $O_3$ were included in Lapina
et al. (2014). Such quantitative comparisons will also be carried out between STEM and its other
boundary condition models in future studies.
2.2.2.   Additional base and sensitivity simulations from selected models

In addition to the base and 20% EAS all-category emission perturbation simulations, the global RAQMS model conducted a sensitivity simulation in which the East Asian anthropogenic emissions were zeroed out, which was also used as STEM's boundary conditions (Table 1b). We calculate the "$S_{O3}$" metric (eq. (3)) using the $O_3$ sensitivities in STEM and RAQMS at the receptor regions in response to both 20% and 100% of emission reductions, to explore the relationships between the $O_3$ sensitivity and the size of the emission perturbation. A closer-to-one "$S_{O3}$" value indicates higher scalability of the sensitivity based on the 20% emission perturbation method for obtaining the full "contribution" of the East Asian anthropogenic emissions on the NAM $O_3$.

$$S_{O3} = R(O_3, EAS, 100\%)/R(O_3, EAS, 20\%)/5 \tag{3}$$
Where: $R(O_3, EAS, 100\%) = BASE\ O_3 - EASALL(-100\%)\ O_3$

The RAQMS model also provided a base simulation that assimilated satellite $O_3$ products from the Ozone Monitoring Instrument (OMI, Levelt et al., 2006) and Microwave Limb Sounder (MLS, Livesey et al., 2008) (Pierce et al., 2007), which was used to help better understand the regional model base run error sources, as well as for demonstrating the use of satellite observations to help improve the representation of the trans-boundary pollution.

We also used a number of sensitivity simulations produced by the GEOS-Chem adjoint model v35f in which the emissions from selected anthropogenic emission sectors (power&industry, transportation, residential) or individual $O_3$ precursor chemical species ($NO_x$, VOC, CO) over the East Asia were reduced by 20%. Additional simulations for the 2008-2009 periods by the SNU GEOS-Chem were also utilized to quantify the East Asia and non-NAM anthropogenic source impacts in comparison with the 2010 conditions that we mainly focus on in this study.

### 2.3.    *In-situ and satellite observations*

#### 2.3.1.   In-situ observations

Over the receptor NAM, the hourly $O_3$ observations at the Clean Air Status and Trends Network (CASTNET, http://epa.gov/castnet/javaweb/index.html) sites were used to evaluate the global and regional models' base simulations in four subregions: western US (i.e., the EPA regions 8, 9, 10); southern US (i.e., the EPA regions 4 and 6), the Midwest (i.e., the EPA regions 5 and 7), and the northeast (i.e., the EPA regions 1-3). The numbers of sites used in global and regional models' evaluation in each US subregion are summarized in Tables 2-3. The locations of these sites and the subregions they belong to are indicated in Figure 2a, overlaid on a model-based terrain height map. A majority of the CASTNET sites in the western US are located at high elevation (>1 km) remote or rural regions, more susceptible to the trans-boundary pollution (e.g., Jaffe, 2011). Most of the sites in the other three subregions are located in low elevation regions, mainly affected by local and regional pollution. The model-based terrain heights fairly well represent the reality on subregional scale – the differences between the actual and model-based subregional mean terrain heights at the CASTNET sites are smaller than 0.1 km (Table 3).

During May-June 2010, intense ozonesonde measurements were made at multiple California locations (Cooper et al., 2011), in support of the NOAA "California Nexus (CalNex): Research at the Nexus of Air Quality and Climate Change" field experiment (Ryerson et al., 2013). They have been used to evaluate the simulated $O_3$ vertical profiles by the HTAP2 participating

models. The detailed evaluation results have been shown by Cooper et al. (2016), and will be
covered by subsequent publications.
Over HTAP2's EAS source region, the global models' $O_3$ performance was evaluated
against the monthly-mean surface in-situ $O_3$ measurements at 11 sites within the Acid Deposition
Monitoring Network in East Asia (EANET, http://www.eanet.asia) that had data throughout the
year of 2010. These include eight Japanese and three Korean sites (Figure 3a), all of which are
located at low elevation regions (2-150 m). The reported monthly mean observations at these sites
were based on weekly or daily sampled data, varying among sites.
2.3.2.  Satellite products
In two case studies of high $O_3$ episodes, L2 and L3 $O_3$ and CO retrievals from several
satellite instruments were used to assess the impacts of trans-Pacific pollution transport and
stratospheric $O_3$ intrusions on NAM $O_3$ levels in early May. These include: 1) the early afternoon
$O_3$ and CO profiles version 5 from the Tropospheric Emission Spectrometer (TES) (Beer et al.,
2001; Beer, 2006) on the Aura satellite; 2) the mid-morning $O_3$ profiles from the METOP-Infrared
Atmospheric Sounding Interferometer (IASI), which were retrieved using the Jet Propulsion
Laboratory (JPL) TES optimal estimation retrieval algorithm (Bowman et al., 2006) for selected
areas including the western US (Oetjen et al., 2014, 2016); as well as 3) the early afternoon L3 $O_3$
and CO maps (version 6, 1°×1°) from the Aqua Atmospheric Infrared Sounder (AIRS) instrument.
The TES tropospheric $O_3$ retrieval is often sensitive to the mid- to lower free troposphere, and $O_3$
at these altitudes in the Eastern Pacific is known to possibly impact the downwind US surface air
quality at later times (Huang et al., 2010; Parrish et al., 2010). TES $O_3$ is generally positively
biased by <15% relative to high accuracy/precision reference datasets (e.g., Verstraeten et al.,
2013). Although IASI is in general less sensitive than TES due to its coarse spectral resolution, the
681–316 hPa partial column-averaged $O_3$ mixing ratios in the JPL product agree well with TES
$O_3$ for the 2008–2011 period with a -3.9 ppbv offset (Oetjen et al., 2016). Note that IASI $O_3$ data
are processed operationally in Europe using a different algorithm. For this work we used $O_3$
profiles from TES and IASI processed using a consistent algorithm at JPL, although the latter set
of data represents only a small subset of the full set of the IASI radiance measurements. The IASI
and TES L2 $O_3$ profiles (screened by the retrieval quality and the C-Curve flags) were used to
evaluate the STEM $O_3$ vertical distributions in the different base simulations, and the satellite
observation operators were applied in these comparisons. Taking TES as an example, its
observation operator $h_z$ for $O_3$ is written in (4):
$h_z = z_c + A_{TES} \left( \ln(F_{TES}(c)) - z_c \right)$                                           (4)
where $z_c$ is the natural log form of the TES constraint vector (a priori) in volume mixing ratio.
$A_{TES}$ is the averaging kernel matrix reflecting the sensitivity of retrieval to changes in the true state
(Rodgers, 2000). $F_{TES}$ projects the modeled $O_3$ concentration fields c to the TES grid using spatial
and temporal interpolation. The exponential of $h_z$ is then used to compute the mismatches between
the model and TES $O_3$ retrievals as the model evaluation. A small mismatch between model with
the satellite observation operators and the satellite retrievals may indicate either good model
performance or may be the low sensitivity of the retrievals to the true $O_3$ profile. AIRS $O_3$ is
sensitive to the altitudes near the tropopause, with positive biases over the ozonesondes in the
upper troposphere (e.g., Bian et al., 2007); AIRS CO is most sensitive to 300–600 hPa (Warner et
al., 2007) and is frequently used together with the AIRS $O_3$ to distinguish the stratospheric $O_3$
intrusions from long-range transported anthropogenic or biomass burning pollution. We use the
L3 AIRS products in this study to get a broad overview of the areas that are strongly impacted by
the stratospheric $O_3$ intrusions or/and LRT of pollution.
The bottom-up $NO_x$ emissions from the HTAP2 inventory were assessed on a monthly base
by comparing the GEOS-Chem nitrogen dioxide ($NO_2$) columns with the de-striped KNMI (Royal
Netherlands Meteorological Institute) OMI column $NO_2$ product version 2.0 (Boersma et al.,
2011a, b). For this model evaluation against the OMI L2 products, the $NO_2$ fields calculated by the
GEOS-Chem adjoint model were saved daily at 13:30 local solar time, roughly coinciding with
the Aura and Aqua overpassing times. Other parameters used in the model column calculations
came from the GEOS-5/GEOS-Chem monthly mean conditions. The OMI data that passed the
tropospheric quality flag at 13-14 local time were selected based on the following screening criteria:
surface albedo<0.3; cloud fraction<0.2; solar zenith angle <75°; and viewing zenith angle <45°.
The averaging kernels (Eskes and Boersma, 2003) and Air Mass Factors (AMFs) in the KNMI
product were used to calculate the modeled tropospheric $NO_2$ vertical columns comparable to the
OMI's. Details of the method to compare the model-based $NO_2$ columns with the KNMI OMI's
can be found in Huang et al. (2014).

## 3.    Results and Discussions

*3.1.    Evaluation of the HTAP2 bottom-up $NO_x$ emissions and the model base simulations*

3.1.1.   Evaluation of the bottom-up $NO_x$ emissions

The comparison of the GEOS-Chem adjoint $NO_2$ columns with the OMI product was used
to help assess the bottom-up HTAP2 $NO_x$ emissions. Figure 4 shows that $NO_2$ columns from
GEOS-Chem's base simulations over the US are overall overestimated. While grid-scale
differences in $NO_2$ columns may not be directly indicative of emissions biases (Qu et al., 2016),
these discrepancies are possibly due to a positive bias in the bottom-up emissions, mainly from the
anthropogenic sources, which have also been pointed out by Anderson et al. (2014) and Travis et
al. (2016). Larger OMI-model disagreement was found over the central/eastern US in June 2010
than in May, likely also due to the uncertainty in GEOS-Chem's soil or lightning $NO_x$ emissions,
which appear to be high over these regions (Figure S1). The $NO_2$ columns in the GEOS-Chem
base simulation were overestimated in many northern China rural areas and underpredicted in a
few urban areas in the East Asia as well as a broad area in the southwestern China. The mismatches
between model and OMI $NO_2$ fell within the ranges of the comparison between the GOME2 $NO_2$
column product and six models' simulations over China in summer 2008 (Quennehen et al., 2016).
Also, the use of monthly-mean anthropogenic emissions as well as the overall rough treatment of
emission height and temporal profiles can be sources of uncertainty. These global model
evaluation results suggest that the EAS-NAM SR relationships analyzed using this inventory may
overall overestimate the NAM local contribution and underestimate the EAS contribution—Under
different chemical regimes, this statement would also rely on the quality of other $O_3$ precursors'
emissions in the HTAP2 inventory, and they may be associated with variable uncertainties
depending on the species or emission sector as introduced in Section 2.1. Therefore, careful
assessment of other key $O_3$ precursors' emissions in the inventory is needed in the future work. It
is important to note that uncertainty in satellite retrievals can prevent us from producing accurate
assessment on emissions (e.g., van Noije et al., 2006), and this comparison does not account for
the biases in the used OMI data, and would be further validated by using other OMI $NO_2$ products
as well as the bias-corrected (if applicable) in-situ $NO_2$ measurements. We also recommend more
global models to save their calculations more frequently, at least near the satellite overpassing
times, for a more comprehensive assessment of the emission inventory and a better understanding
of the model biases.
3.1.2.  Evaluation of the global model $O_3$ performance in NAM and EAS
The monthly-mean surface $O_3$ from multiple global models' free runs was evaluated with
the CASTNET observations, at the stations with 95% of the hourly $O_3$ observation completeness
for the 1 May-30 June 2010 period. The mean biases and RMSEs for these two months were
summarized in Table 2a by US subregions. The three boundary condition-model as well as the
eight-model ensembles overall underpredicted $O_3$ in the western US (by ~3-6 ppbv), similar to the
HTAP1 model performance over these regions for May-June 2001 presented in Fiore et al. (2009).
This can be due to the underestimated trans-boundary pollution (as indicated by the evaluation of
modeled $O_3$ profiles with ozonesondes and satellite $O_3$ products). In addition, the coarser model
resolutions are less capable of resolving the local features that influence the pollutants' import
processes, chemical transformation, as well as regional processes such as the cross-state pollution
transport over complex terrains. The global RAQMS base simulation with satellite assimilation
improved the free tropospheric $O_3$ structure as its comparisons with the ozonesondes shows, which
also enhanced the simulated monthly-mean surface $O_3$ by up to >10 ppbv in the western US and
some coastal areas in the southeastern US (Figure S2, left). The global models overall significantly
overestimated $O_3$ in the other three subregions (by 8-12 ppbv), close to HTAP1 model performance
for May-June 2001 over the similar areas (Fiore et al., 2009) and in the Lapina et al. (2014) study
for 2010, in large part due to the uncertainties in the bottom-up emissions as discussed in Section
3.1.1.  Satellite  assimilation  led  to  2-6  ppbv  higher  RAQMS  surface  $O_3$  in  the
central/southern/eastern US than in its free simulation, which are associated with higher positive
biases.
The surface $O_3$ performance by individual global models varies significantly, e.g., with the
RMSEs at all CASTNET sites ranging from ~9 ppbv to >15 ppbv (Table 2b). As reported in the
literature (e.g., Geddes et al., 2016; Travis et al., 2016), the representation of land use/land cover,
boundary layer mixing and chemistry can be sources of uncertainty for certain global model (i.e.,
GEOS-Chem), but how serious these issues were in the other models need to be investigated
further. Some other possible reasons include the variation of these models' non-anthropogenic
emission inputs and chemical mechanisms (Table 1c). Future work should emphasize on
evaluating and comparing all models on process level to better understand their performance.
Except in the northeastern US, the eight-model ensembles show better agreement with the
CASTNET $O_3$ observations than the three boundary condition-model ensemble. Overall the three-
model ensemble only outperforms one model but the eight-model ensemble outperforms seven
individuals. This reflects that averaging the results from a larger number of models in this case
more effectively cancelled out the positive or negative biases from the individual models.
The monthly-mean surface $O_3$ from multiple global models' free runs was also evaluated
with the EANET observations. Among the three boundary condition models, GEOS-Chem
produced higher $O_3$ than the other two throughout the year, and C-IFS $O_3$ is the lowest from April
to December. The three-model and eight-model ensembles are lower than the surface $O_3$
observations by <10 ppbv during high $O_3$ seasons (winter/spring), but show substantial (>10 ppbv)
positive biases during low $O_3$ seasons especially in July and August (Figure 3b), similar to the
HTAP1 model performance over Japan in 2001 (Fiore et al., 2009). During May-June 2010,
generally the models performed better at the Japanese sites than at the Korean sites (Table 2c),
with significant positive biases occurring at low $O_3$ regions (e.g., in central Japan) and negative
biases found at high $O_3$ regions, mainly owing to the uncertainty in the local and upwind emissions.
The different approaches to generate the monthly-mean modeled and the observed $O_3$ data may
have also contributed to these model-observation discrepancies. Overall $O_3$ performance by
individual models varies less significantly than at the CASTNET sites, with RMSEs ranging from
8.6 ppbv to ~13 ppbv (Table 2b). The three-model ensemble outperforms two individual models,
and the eight-model ensemble outperforms six individual models. Unlike at the CASTNET sites,
the three-model ensemble agrees better with the observations than the eight-model ensemble
(Table 2c).
3.1.3.  Evaluation of the STEM regional base simulations w/ three sets of boundary conditions
The three STEM base simulations using different boundary conditions were evaluated with
the hourly $O_3$ observations at the CASTNET sites in the four US subregions. The evaluation
included the 8 May-30 June 2010 period to exclude the results during the one-week spin-up period.
The time series plots of observed and modeled $O_3$ at the western US CASTNET sites show that
STEM was capable of capturing several high $O_3$ periods, and it produced larger biases during the
nighttime (Figure 2c), as a result of the poorer WRF performance. Figure 2c and the evaluation
statistics in Table 3a-b indicate that STEM/C-IFS $O_3$ concentrations are associated with the highest
positive bias and RMSE, while the STEM/GEOS-Chem and STEM/RAQMS predictions were
positively and negatively biased by less than 2 ppbv, respectively, with similar RMSEs and
correlations with the observations. The quality of the three STEM simulation mean is closest to
the STEM/GEOS-Chem run, with the mean bias/RMSE of ~1.6/4.9 ppbv, much better than the
three-boundary model ensemble (-5.7/10.4 ppbv). However, this good performance can be a net
effect of incorrect partitioning between the trans-boundary and local source contributions, with the
former being underestimated and offsetting the overestimation of the latter. Switching the STEM
chemical boundary conditions to the assimilated RAQMS base simulation led to increases in the
simulated surface $O_3$ concentrations by >9 ppbv in the western US (Figure S2, right), associated
with higher positive biases (due to several factors discussed in the next paragraph). Regional-scale
assimilation could further reduce uncertainties introduced from regional meteorological and
emission inputs to obtain better modeled total $O_3$ and the partitioning of trans-boundary versus US
contributions (e.g., Huang et al., 2015).
The three STEM base simulations all significantly overpredicted $O_3$ over the rest of the US
in part due to the uncertainties in $NO_x$ emissions, with the STEM/RAQMS associated with the
lowest RMSEs and mean biases, but STEM/C-IFS correlated best with the observations (Table
3b). These positive biases are higher than the global model ensembles', which can partially result
from the possible unrealistic VOC speciation of the emission inventory and the SAPRC 99
chemical mechanism: Although SAPRC mechanisms have been used in air quality modeling for
regulatory applications in some US states such as California, they usually produced higher $O_3$ than
other mechanisms such as the CB04 and the CB05 (which were used by some HTAP2 global
models, see Table 1c) over the US, and the comparisons between SAPRC 99 and SAPRC 2007
are still in progress (e.g., Luecken et al., 2008; Zhang et al., 2012; Cai et al., 2011). It is important
to timely update the chemical mechanisms in the chemistry models, and we also suggest to timely
upgrade the VOC speciation in the bottom-up emission inventories in the US to benefit the air
quality modeling. Additionally, the uncertainty from non-anthropogenic emissions, such as the
biogenic VOC emissions from WRF/MEGAN which is known to often have positive biases, can
be another cause: As Hogrefe et al. (2011) presented, the MEGAN emissions resulted in a higher
$O_3$ response to hypothetical anthropogenic $NO_x$ emission reductions compared with another set of
biogenic emission input. Huang et al. (2017) showed that MEGAN's positive biases are in part
due to the positively-biased temperature and radiation in WRF, and reducing ~2°C in WRF's
temperature biases using a different land initialization approach led to ~20% decreases in
MEGAN's isoprene emission estimates in September 2013 over some southeastern US regions.
These temperature and radiation biases, can also be important sources of uncertainty in the
modeled $O_3$ production. Quantifying the impacts of overestimated biogenic emissions and the
biased weather fields that contributed to the biases in emissions on the modeled $O_3$ is still an
ongoing work. Some existing studies also reported $O_3$ and $NO_2$ biases from other regional models
in the eastern US, due to the chemical mechanism and biases in $NO_x$ and biogenic VOC emissions
(e.g., Canty et al., 2015). We anticipate that the results from the Air Quality Model Evaluation
International Initiative (AQMEII) experiment (e.g., Schere et al., 2012; Solazzo et al., 2012;
Galmarini et al., 2015, 2017), which involves more regional model simulations over the US with
the similar set of boundary conditions but different chemical mechanisms and non-anthropogenic
emission inputs, can help better understand the causes of errors in the simulated total $O_3$.
*3.2.    The NAM surface $O_3$ sensitivity to extra-regional anthropogenic pollutants*
3.2.1.   Global model ensembles
The impact of all foreign (i.e. non-NAM) anthropogenic sources on NAM surface $O_3$ was
first explored, including the spatial distributions of the RERER metric (eq. (2)) based on various
global models' simulations (Figure 5), and the domain wide mean sensitivities R ($O_3$, non-NAM,
20%) (eq. (1d)) (Figure 6). Across the NAM, the strongest impacts were found in spring time
(March-April-May, larger than 1.5 ppbv in average over the domain) and the weakest impacts are
shown during the summertime (June-July-August, 1.0-1.3 ppbv), consistent with the existing
knowledge on the seasonal variability of the non-local pollution impacts on NAM for other years
(e.g., Fiore et al., 2009; Reidmiller et al., 2009). All global models indicate strong non-NAM
anthropogenic source impacts on the western US mainly due to the impact of its high elevation,
and also near the US-Mexico border areas, especially southern Texas, due to their vicinity to the
Mexican (not included in the NAM source regions, see Figure 1) emission sources. Over the
western states, stronger non-local impacts were reflected from the results based on higher-
horizontal resolution global models (e.g., the >0.6 RERER values from the half degree EMEP
model, corresponding to its higher R($O_3$, non-NAM, 20%) values than the other models'), similar
to the findings in previous modeling studies (Lin et al., 2010, 2012a). Although on a coarse
horizontal resolution of 2.8°, OsloCTM3 suggests stronger extra-regional source influences on the
northwestern US and the US-Canada border regions than the other models. Its largest number of
vertical layers among all global models might be a cause. Larger-than-1 RERER values are often
seen near the urban areas and large point sources due to the titration, especially evident from the
higher resolution model results. The R($O_3$, EAS, 20%) values are larger than 1/3 of the R($O_3$, non-
NAM, 20%) (0.2-0.5 ppbv from April to June), more than 3-4 times higher than R($O_3$, EUR, 20%)
and R(O$_3$, SAS, 20%). Note that all eight models contributed to the R(O$_3$, EAS, 20%) calculations,
but one or two models did not provide all necessary sensitivity runs to compute the RERER, R(O$_3$,
non-NAM, 20%), R(O$_3$, EUR, 20%), or R(O$_3$, SAS, 20%).
Comparing to the HTAP1 modeling results, the magnitudes of R(O$_3$, EUR, 20%) from this
study are smaller by a factor of 2-3; In contrast, the R(O$_3$, non-NAM, 20%) and R(O$_3$, EAS, 20%)
values are >50% higher than the HTAP1 modeling results. The different HTAP1 and HTAP2
results are possibly due to the following three reasons: 1) the substantial improvement in the
European air quality over the past decades that is shown in Crippa et al. (2016) and Pouliot et al.
(2015), which contrasts with the growing anthropogenic emissions from the East Asia and other
developing countries during 2001-2010; 2) the changes in the HTAP2 experiment setup from
HTAP1. This includes the differences in the participating models, and the different region
definitions, e.g., EUR by HTAP1's definition includes regions in Russia/Belarussia/Ukraine,
Middle East and North Africa that are excluded from the HTAP2's EUR domain. For EAS and
SAS, however, the regions not overlapped by HTAP1 and HTAP2 are mostly in the less
populated/polluted regions; 3) the stronger trans-Pacific transport in 2010 than in 2000-2001, as
first introduced in Section 2.2.1. Interannual variability of R(O$_3$, EAS, 20%) and R(O$_3$, non-NAM,
20%) is also found between 2010 and 2008-2009, based on the SNU GEOS-Chem calculations
(Figure S3). Foreign anthropogenic pollution impact on NAM was stronger in 2010 than in 2008-
2009, especially in April-May. This can be in part due to the higher O$_3$ precursors' emissions in
2010 from extra-regions including the East Asia (Table S1), as well as the spring 2010
meteorological conditions that favored the trans-Pacific pollution transport.
These monthly- and regional-mean R(O$_3$, EAS, 20%) values suggest that despite dilution
along the great transport distance, the EAS anthropogenic sources still had distinguishable impact
on the NAM surface O$_3$. Similar to the findings from the HTAP1 studies, the large intermodel
variability (as indicated in Table 4) in the estimates of intercontinental SR relationships indicates
the uncertainties of these models in representing the key atmospheric processes which needs more
investigations in the future. Figure 6b compares the R(O$_3$, EAS, 20%) estimated by individual
boundary condition models, their ensemble mean sensitivities, and the eight-global model mean.
The averaged R(O$_3$, EAS, 20%) from the boundary condition model results are smaller than the
eight-global model mean, and except for July-October 2010, GEOS-Chem gives higher R(O$_3$, EAS,
20%) than RAQMS and C-IFS, consistent with its highest O$_3$ prediction in the EAS source region
(Figure 3b). Overall, R(O$_3$, EAS, 20%) and its intermodel differences are much smaller than the
biases of the modeled total O$_3$ in NAM. Other factors can contribute more significantly to the
biases in the modeled total O$_3$, such as the stratospheric O$_3$ intrusion and the local O$_3$ formation,
and assessing the impacts from these factors would be also helpful for understanding the
uncertainties in the modeled O$_3$.
The O$_3$ sensitivities in response to the perturbations of individual species or sector
emissions in East Asia, estimated by the GEOS-Chem adjoint model, were also analyzed (Figure
S3). These sensitivities show similar seasonal variability to R(O$_3$, EAS, 20%), with the values
~twice as high in the spring than in summer, also consistent with the results on previous years
based on the 20% emission perturbation approach (e.g., Fiore et al., 2009; Brown-Steiner and Hess,
2011; Emmons et al., 2012). However, this seasonal variability is weaker than the results based on
the tagged tracer approach for earlier years: Using the CAM-Chem model, Brown-Steiner and
Hess (2011) reported that during the springtime, Asian $O_3$ created from the anthropogenic/biofuel
$NO_x$ emissions affected NAM $O_3$ ~three times as strongly as in summer. This is because the
nonlinear $O_3$ chemistry, which is stronger outside of summer, caused larger $O_3$ responses to a 100%
reduction of $NO_x$ emissions than 5 times of the $O_3$ responses to a 20% reduction of $NO_x$ emissions.
The EAS anthropogenic $NO_x$ emissions more strongly impacted the NAM surface $O_3$ than the
other major $O_3$ precursors, similar to the findings in Fiore et al. (2009) and Reidmiller et al. (2009)
using the perturbation approach, as well as the conclusions in Lapina et al. (2014) based on the
adjoint sensitivity analyses. Emissions from the power&industrial sectors are higher in East Asia
than the other sectors (Table S1), resulting in its stronger influences on the NAM surface $O_3$. As
the observed $NO_2$ columns started to drop since 2010 due to the effective denitration devices
implemented at the Chinese power and industrial plants (e.g., Liu et al., 2016), depending on the
changes in the VOC emissions, it is anticipated to see different R($O_3$, EAS, 20%) values for the
years after 2010. Therefore, continued studies to assess the East Asian anthropogenic pollution
impacts on NAM during more recent years is needed. As emissions from various source sectors
can differ by their emitted altitudes and temporal (from diurnal to seasonal) profiles, efforts should
also be placed to have the models timely update the heights and temporal profiles of the emissions
from those various sectors.
3.2.2.  Regional model sensitivities and their connections with the boundary condition models'
The monthly-mean STEM surface R($O_3$, EAS, 20%) sensitivities based on different
boundary condition models were inter-compared, and also compared with the R($O_3$, EAS, 20%)
estimated by their boundary condition models as well as the global model ensemble mean (Figure
7). For both May and June 2010, the domain-wide mean R($O_3$, EAS, 20%) values from
STEM/RAQMS were higher than the estimates from RAQMS by 0.03 ppbv; the STEM/GEOS-
Chem R($O_3$, EAS, 20%) values are lower than those of GEOS-Chem by 0.01-0.06 ppbv, and the
STEM/C-IFS R($O_3$, EAS, 20%) is 0.02 ppbv higher than C-IFS's in June but slightly (<<0.01 ppbv)
lower in May. These differences are overall smaller than the inter-global model differences, and
can be due to various factors including the uncertainties in boundary condition chemical species
mapping, and the different meteorological/terrain fields/chemistry in the global and regional model
pairs. The STEM R($O_3$, EAS, 20%) ensemble mean values, however, are less than 0.02 ppbv
different from its boundary condition model's ensemble mean for both months. The STEM R($O_3$,
EAS, 20%) ensemble mean value in June is also close to the eight-global model ensemble mean,
but is ~0.05 ppbv lower than the eight-model mean in May. Choosing other/more global model
outputs as STEM's boundary conditions may lead to different STEM ensemble mean R($O_3$, EAS,
20%) estimates. We also found that the period mean R($O_3$, EAS, 20%) of ~0.2 ppbv sampled only
at the CASTNET sites (Table 3a) are smaller than those averaged in all model grids. This indicates
that currently the sparsely distributed surface network (especially over the western US that is more
strongly affected by the extra-regional sources than the other US regions) may miss many LRT
episodes that impact the NAM. The planned geostationary satellites with ~2-5 km footprint sizes
and hourly sampling frequency (Hilsenrath and Chance, 2013; Zoogman et al., 2017) will help
better capture the high $O_3$ and LRT episodes in these regions.
The spatial patterns of the monthly-mean STEM surface R($O_3$, EAS, 20%) sensitivities
based on the three boundary condition models are notably different, but overall resemble what's
estimated by the corresponding boundary condition model, and the STEM sensitivities show more
local details in certain high elevation regions in the US west (Figure 8 shows the June 2010
conditions as an example). These different sensitivities were investigated further, by examining
the $R(O_3, EAS, 20\%)$ values near the source regions (i.e., East Asia) as well as near the receptor
regions (Figure 9). More East Asian anthropogenic $O_3$ seems to be transported at the upper
troposphere in RAQMS than in the other two models. GEOS-Chem and RAQMS $R(O_3, EAS, 20\%)$
sensitivities are similar over the EAS as well as the 500-900 hPa near the receptor in the eastern
Pacific (at ~135°W), the altitudes US surface $O_3$ are most strongly sensitive to during the
summertime as concluded from previous studies (e.g., Huang et al., 2010, 2013a; Parrish et al.,
2010). Despite the close NAM domain-wide mean values from the STEM/GEOS-Chem and
STEM/RAQMS, the spatial patterns of $R(O_3, EAS, 20\%)$ over NAM differ in these two cases,
with the latter case showing sharper gradients especially in the western US, partially due to the
impact of its higher horizontal resolution. The $R(O_3, EAS, 20\%)$ values from STEM/C-IFS are
lower than from the other two cases both near the sources and at (near) NAM. The STEM surface
(also near surface, not shown in figures) $R(O_3, EAS, 20\%)$ does not spatially correlate well with
the column $R(O_3, EAS, 20\%)$, the latter of which contributed more to the base case $O_3$ columns,
indicating that a good portion of the transported East Asian pollution did not descend to the lower
altitudes to impact the boundary layer/ground level air quality. An additional regional simulation
was performed in which the STEM boundary conditions were downscaled from a RAQMS
simulation without the East Asian anthropogenic emissions. The non-linear emission perturbation-
$O_3$ response relationships, as the larger-than-1 $S_{o3}$ metric (eq. (3)) indicate, are seen across the
domain, for both the surface and column $O_3$ (Figure 8). $S_{o3}$ for column $O_3$, ranging from 1.15-1.25
in most regions, are overall ~0.05 higher than $S_{o3}$ for the surface $O_3$. Therefore, the full source
contribution obtained by linearly scaling the receptor regional mean $O_3$ sensitivity to the 20%
reduction in the source region emissions may be underestimated by at least ~10%.
### 3.2.3. Regional model MDA8 sensitivities on all days and during the $O_3$ exceedances
The temporal variability of the STEM $R(O_3, EAS, 20\%)$ ensemble sensitivities were also
studied. For most US subregions, 3-6 LRT episodes (defined as when the sensitivities are above
the period mean) were identified during May-June. Only in certain regions, we find that the
planetary boundary layer heights (PBLHs) were higher during the LRT episodes (i.e., the daily
daytime-mean $R(O_3, EAS, 20\%)$ and PBLHs show medium/strong positive correlations ($r>0.5$)),
as these correlations may have been complicated by the relationships between the PBLHs and the
local influences. Throughout this period, the hourly $R(O_3, EAS, 20\%)$ and the observed $O_3$ at the
surface CASTNET sites are weakly correlated (Table 3a), but they display similar diurnal cycles
(e.g., Figures 2c and 2d for the western US sites), possibly because the deeper boundary layer
depth during the daytime enhanced entrainment down-mixing of the extra-regional pollutants to
the surface. The identified diurnal variability of the $R(O_3, EAS, 20\%)$ can cause differences in the
calculated MDA8 and all-hour mean $R(O_3, EAS, 20\%)$ values. Figure S4 shows that the mean
$R(MDA8, EAS, 20\%)$ values, usually at daytimes, are higher than the all-hour averaged $R(O_3,$
$EAS, 20\%)$ in most STEM model grids during both months. Therefore, it is important for more
HTAP2 participating models to save their outputs hourly in order to conveniently compute the
policy-relevant metrics for the $O_3$ sensitivities. Also, the hourly sampling frequency of the planned
geostationary satellites is anticipated to be more helpful for evaluating the impacts of the LRT
episodes.

The STEM R(MDA8, EAS, 20%) in all model grids within the four US subregions were
averaged on all days during May-June 2010 and only on the days when the simulated total MDA8
$O_3$ is over 70 ppbv (Figure 10). These sensitivities also show appreciable spatial variability: from
0.35-0.58 ppbv in the western US (also with the largest standard deviations, not shown), which is
slightly higher than the HTAP1 results reported by Reidmiller et al. (2009) for Spring 2001, to
~0.1-0.25 ppbv in the rest three subregions, which is close to the Reidmiller et al. (2009) results.
Comparing the solid bar plots in Figures 10-11, we found that on all days in the three non-
western subregions, R(MDA8, EAS, 20%) values sampled at CASTNET sites are slightly smaller
than those computed for all model grids, while in the non-western states the opposite differences
are seen. This again suggests that expanding observation network would help better capture the
high $O_3$ and LRT episodes.
Figure 10 suggests smaller R(MDA8, EAS, 20%) values during the high $O_3$ days in all
subregions. However, STEM's total $O_3$ concentrations at CASTNET sites during the $O_3$
exceedances were substantially overpredicted in non-western US regions while significantly
underpredicted in the western US (see mean biases above the bar plots in Figure 11). Therefore,
the R(MDA8, EAS, 20%) values shown in Figure 10 during the model-based periods of $O_3$
exceedances can represent the sensitivities during the actual periods of $O_3$ compliance in non-
western US regions, and may not represent the sensitivities during all actual $O_3$ exceedances in the
western US. Figures 11-12 show that if calculated only at the CASTNET sites during the
exceedances, in non-western US regions, R(MDA8, EAS, 20%) is 0.02-0.07 ppbv smaller during
the high $O_3$ total days. This is qualitatively consistent with the findings in Reidmiller et al. (2009),
and is possibly because that the LRT impacts were stronger on some days with good dispersion
conditions when the NAAQS was not exceeded, but weaker on some high $O_3$ days under stagnant
conditions. In contrast, western US R(MDA8, EAS, 20%) at CASTNET sites was ~0.05 ppbv
higher on high $O_3$ days than for all days, and this differences are larger in rural/remote areas where
local influences are less dominant. As a result, the medium/strong positive correlations are found
between modeled LRT of pollution and the total $O_3$ in these regions (Table 3a; Lin et al., 2012a).
*3.3.    Case studies of spring (9 May) and summer (10 June) LRT events mixed with stratospheric*
*$O_3$ intrusions*
Lin et al. (2012a, b) and Neuman et al. (2012) showed that the trans-Pacific pollution
transport intensely impacted the western US during 8-10 May, 2010, intermingled with a
stratospheric intrusion that contributed to at least 1/3 of the total $O_3$ in some high elevation regions.
This episode is indeed indicated by the $O_3$ and CO products from AIRS and TES at ~500 hPa over
the Eastern Pacific (Figure 13), and the observed TES and IASI $O_3$ profiles over the western US
indicated elevated $O_3$ levels (>80 ppbv) at 700-900 hPa. Huang et al. (2013b) found that the
meteorological conditions during this period (i.e., a strong jet at ~700 hPa with wind speed >20
m/s shifted southwesterly when passing the southern California and continued to travel towards
the mountain states), along with the orographic lifting, efficiently exported the southern California
anthropogenic pollution, which was chemically coupled with the extra-regional pollution and
significantly enhanced the $O_3$ levels in the US intermountain west.

We selected this episode to compare the STEM surface total $O_3$ concentrations as well as
the R($O_3$, EAS, 20%) sensitivities based on the different HTAP2 boundary condition models.
Figure 14 evaluates the simulated $O_3$ profiles in the western US from several STEM base
simulations against the TES and IASI $O_3$ retrievals, and Figures 15a-d indicate the performance of
the daily surface total MDA8 $O_3$ from these simulations. We found that the underestimated free
tropospheric $O_3$ from the STEM simulations that used any single free-running chemical boundary
conditions contributed to the underestimated STEM surface $O_3$ in the high elevation mountain
states: e.g., by 9-14 ppbv at three CASTNET sites (Grand Canyon National Park (NP), AZ;
Canyonlands NP, UT; and Rocky Mountain NP, CO) where $O_3$ exceedances were observed. The
unsatisfactory performance by free-running global models during high $O_3$ events would pose
difficulties for regional models (regardless of their resolutions and other configurations,
parameterization) to accurately estimate the SR relationships using boundary conditions
downscaled from these model runs. The STEM base simulation using the RAQMS assimilated
fields as the boundary conditions, agrees most with the observed $O_3$ at the CASTNET sites, as well
as the TES and IASI $O_3$ profiles in the western states. Similar to the conclusions drawn in Huang
et al. (2010, 2015) for summer 2008, we again demonstrated the robustness of satellite chemical
data assimilation for improving the boundary condition models' $O_3$ performance. As the
enhancement of $O_3$ due to the assimilation is much larger than the $O_3$ sensitivities to the EAS
anthropogenic emissions, the assimilation mainly improved the contributions from other sources,
possibly including the stratospheric $O_3$.
The quality of the model boundary conditions only indicates how well the total "transported
background" component is represented, and can not be directly connected with the accuracy of the
model estimated R($O_3$, EAS, 20%) sensitivities, which also show notable intermodel differences:
The estimated R(MDA8, EAS, 20%) in the different STEM cases range from <1.0 ppbv to ~1.3
ppbv, at least 40% higher than the May-June period mean in Figures 10-11. The mean R(MDA8,
EAS, 20%) at three high $O_3$ CASTNET sites range from 0.73 (STEM/GEOS-Chem) to 0.98 ppbv
(STEM/C-IFS), with the mean $S_{O_3}$ of ~1.14 at these sites based on the STEM/RAQMS runs due
to the nonlinear emission perturbation-$O_3$ response relationships (Figure 15e-h). The R(MDA8,
EAS, 100%) from the STEM/RAQMS case is as high as >7 ppbv over the high terrain regions.
These are of smaller magnitudes than the estimates in Lin et al. (2012a), possibly due to the
differences in the used models and the bottom-up emission inputs.
A stratospheric $O_3$ intrusion also affected the Northeast US on the same day, as revealed
by the satellite mid- tropospheric $O_3$ and CO observations (Figure 13). This intrusion was mixed
with LRT East Asian pollution (Figure 15 and Figure S5). However, this intrusion did not enhance
the NE boundary layer/surface $O_3$ concentrations, which were actually anomalously low
(MDA8<40 ppbv) as indicated by the model base simulations and the CASTNET observations
(Figure 15a-d). Similar characteristics during summertime stratospheric $O_3$ intrusion events
around this region have been discussed by Ott et al. (2016). The East Asian pollution less intensely
(<50%) affected the surface $O_3$ levels in these regions than in the US west, due to the greater
transport distances, stronger local emission influence on chemical production/loss, shallower
PBLHs, as well as the impact of the overall flat terrain in the US east.
A summertime LRT event on 9-10 June is analyzed to contrast with the 9 May case study.
Lin et al. (2012b) showed that >80 ppbv of ozonesonde data in northern California at 2-6 km
measured the stratospheric $O_3$ remnants during this episode, and the transported stratospheric $O_3$
contributed to as much as ~50% of the total $O_3$ in southern California based on their model
calculations. We show that on 10 June over 100 ppbv of $O_3$, as well as <90 ppbv CO, was observed
by satellites at ~500 hPa above Nevada and northern California (Figure 16), which again was
substantially underestimated by all free-running models (Figure 17), resulting in the
underpredicted total $O_3$ at two CASTNET sites in southern California (Converse Station and
Joshua Tree NP) that experienced $O_3$ exceedances on this day (Figure 18a-c). The negative biases
in the "transported background" $O_3$ and surface MDA8 $O_3$ were successfully reduced by
incorporating satellite data (Figures 17 and 18d).
Figures 18e-h show that LRT of EAS anthropogenic pollution also strongly affected
southern California and Nevada. Notable intermodel differences are again found in the estimated
R(MDA8, EAS, 20%), but they are overall lower than on 9 May (<1.0 ppbv). The mean R(MDA8,
EAS, 20%) at the two high $O_3$ CASTNET sites range from 0.54 (STEM/C-IFS) to 0.86 ppbv
(STEM/RAQMS), with the mean $S_{O3}$ of ~1.13 at these sites based on the STEM/RAQMS runs
(Figure 18e-h). The R(MDA8, EAS, 100%) from the STEM/RAQMS case is as high as >6 ppbv
over southern California and Nevada. Compared to the spring event, R(MDA8, EAS, 20%) in the
eastern US are discernable only over a limited region, due to weaker transport and stronger local
chemical production/loss.
**4.      Conclusions and suggestions on future directions**
In support of the HTAP Phase 2 experiment that involved high-resolution global models
and regional models' participation to advance the understanding of the pollutants' SR relationships
in the Northern Hemisphere, we conducted a number of regional scale STEM base and forward
sensitivity simulations over NAM during May-June 2010. STEM's top and lateral chemical
boundary conditions were downscaled from three global models' (i.e., GEOS-Chem, RAQMS,
and ECMWF C-IFS) base and sensitivity simulations (in which the East Asian anthropogenic
emissions were reduced by 20%). Despite dilution along the great transport distance, the East
Asian anthropogenic sources still had distinguishable impact on the NAM surface $O_3$, with the
period-mean NAM $O_3$ sensitivities to a 20% reduction of the East Asian anthropogenic emissions
(i.e., R($O_3$, EAS, 20%)) ranging from ~0.24 ppbv (STEM/C-IFS) to ~0.34 ppbv (STEM/RAQMS).
The spatial patterns of the STEM surface $O_3$ sensitivities over NAM overall resembled those from
its corresponding boundary condition model, with regional/period mean R($O_3$, EAS, 20%) differed
slightly (<10%) from its corresponding boundary condition model's, which are smaller than those
among its boundary condition models. The boundary condition models' two-month mean R($O_3$,
EAS, 20%) was ~8% lower than the mean sensitivity estimated by eight global models. Therefore,
choosing other global model outputs as STEM's boundary conditions may lead to different STEM
$O_3$ sensitivities. The biases and RMSEs in the simulated total $O_3$, which differed significantly
among models, can partially be due to the uncertainty in the bottom-up $NO_x$ emission inputs
according to the model comparison with the OMI $NO_2$ columns, and future work on attributing the
intermodel differences on process level is particularly important for better understanding the
sources of uncertainties in the modeled total $O_3$ and its source contribution.
The HTAP2 multi-model ensemble mean R($O_3$, EAS, 20%) values in 2010 were higher
than the HTAP1 reported 2001 conditions, due to a number of reasons including the impacts of
the growing East Asian anthropogenic emissions, the interannual variability in atmospheric
circulation (i.e., stronger trans-Pacific transport in spring 2010 following an El Niño event), and
the different experiment designs of HTAP1 and HTAP2. The GEOS-Chem $O_3$ sensitivities in 2010
were also higher than the 2008-2009 conditions due to the increasing Asian emissions and the
spring 2010 meteorological conditions that favored the trans-Pacific pollution transport. The
GEOS-Chem sensitivity calculations indicate that the East Asian anthropogenic $NO_x$ emissions
mattered more than the other East Asian $O_3$ precursors to the NAM $O_3$, qualitatively consistent
with previous adjoint sensitivity calculations. Continued research is needed on temporal changes
of emissions for different species and sectors in NAM and foreign countries as well as their impacts
on the SR relationships. As emissions from various source sectors can differ by emitted altitudes
and temporal profiles, efforts should also be placed to have the models timely update the height
and temporal profiles of the emissions from various sectors.
An additional STEM simulation was performed in which the boundary conditions were
downscaled from a RAQMS simulation without East Asian anthropogenic emissions (i.e., a 100%
emission reduction), to assess the scalability of the mean $O_3$ sensitivities to the size of the emission
perturbation. The scalability was found to be spatially varying, ranging from 1.15-1.25 for column
$O_3$ in most US regions, which were overall ~0.05 higher than the surface $O_3$'s. Therefore, the full
source contribution obtained by linearly scaling the NAM regional mean $O_3$ sensitivity to the 20%
reduction in the East Asian emissions may be underestimated by at least 10%. The underestimation
in other seasons of the HTAP2 study period may be higher and will need to be quantified in future
work. Also, motivated by Lapina et al. (2014), additional calculations will be conducted in future
to explore the scalability of different $O_3$ metrics in these cases. For future source attribution
analysis, in general it is recommended to directly choose the suitable size of the emission
perturbation based on the specific questions to address, and to avoid linearly scaling $O_3$
sensitivities that are based on other amounts of the perturbations.
The STEM $O_3$ sensitivities to the East Asian anthropogenic emissions (based on three
boundary condition models separately and averagely) were strong during 3-6 episodes in May-
June 2010, following similar diurnal cycles as the total $O_3$. Stronger East Asian anthropogenic
pollution impacts were estimated during the observed $O_3$ exceedances in the western US than on
all days, especially over the high terrain rural/remote areas; in contrast, the East Asian
anthropogenic pollution impacts were less strong during $O_3$ exceedances in other US regions. We
emphasized the importance of saving model results hourly for conveniently calculating policy-
relevant metrics, as well as the usefulness of hourly sampling frequency of the planned
geostationary satellites for better evaluating the impacts of the LRT events.
Based on model calculations, satellite $O_3$ (TES, JPL-IASI, and AIRS), CO (TES and AIRS)
and surface $O_3$ observations on 9 May 2010, we showed the different influences from stratospheric
$O_3$ intrusions along with the transported East Asian pollution on $O_3$ in the western and the eastern
US. This event was further compared with a summer event of 10 June 2010. During both events,
the unsatisfactory performance of free-running (i.e., without chemical data assimilation) global
models would pose difficulties for regional models (regardless of their resolutions and other
configurations, parameterization) to accurately simulate the surface $O_3$ and its source contribution
using boundary conditions downscaled from these model runs. Incorporating satellite (OMI and
MLS) $O_3$ data effectively improved the modeled $O_3$. As chemical data assimilation techniques
keep developing (Bocquet et al., 2015), several HTAP2 participating global models have already
been able to assimilate single- or multi- constitute satellite atmospheric composition data (e.g.,
Miyazaki et al., 2012; Parrington et al., 2008, 2009; Huang et al., 2015; Inness et al., 2015;
Flemming et al., 2017). Comparing the performance of the assimilated fields from different models,
and making the global model assimilated chemical fields in the suitable format for being used as
boundary conditions would be very beneficial for future regional modeling, as well as for better
interpreting the pollutants' distributions especially during the exceptional events. Meanwhile,
efforts should also be devoted to advancing and applying higher-resolution regional scale
modeling and chemical data assimilation. Furthermore, although satellite observations have been
applied for improving the estimated US background $O_3$ (e.g., Huang et al., 2015), using satellite
(and/or other types of) observations to improve SR relationship studies also needs to be explored.
Some of the possible methods include: 1) The combination of data assimilation and the tagging
approach; 2) Introducing observation-constrained emission estimates in the emission perturbation
analyses.
**Acknowledgements**

The global and regional modeling results used in this study have been submitted to the
AeroCom database following the HTAP2 data submission guidelines (http://iek8wikis.iek.fz-
juelich.de/HTAPWiki/HTAP-2-data-submission), or can be made available upon request.
Technical support from Anna Carlin Benedictow, Brigitte Koffi, Jan Griesfeller, and Michael
Schulz regarding formatting and submitting the data to the AeroCom is acknowledged. MH thanks
the research resources at the University of Iowa and JPL/Caltech that supported this study, as well
as the travel funding from the US EPA for attending the related HTAP2 workshops. DKH and YD
recognize support from NASA AQAST. FD Acknowledges support from the Administrative
Arrangement. Part of this research was carried out at the Jet Propulsion Laboratory, California
Institute of Technology, under contract to the National Aeronautics and Space Administration.
Reference herein to any specific commercial product, process or service by trade name, trademark,
manufacturer or otherwise does not constitute or imply its endorsement by the United States
Government or the Jet Propulsion Laboratory, California Institute of Technology. The views,
opinions, and findings contained in this report are those of the author(s) and should not be
construed as an official National Oceanic and Atmospheric Administration or U.S. Government
position, policy, or decision. We also acknowledge the feedbacks from Dr. Gail Tonnesen, two
anonymous reviewers, and Dr. Meiyun Lin on earlier versions of this paper, that helped improve
its quality.

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

Background    Ozone    White    Paper    for    Discussion,
https://www.epa.gov/sites/production/files/2016-03/documents/whitepaper-bgo3-final.pdf.
US    EPA    (2016b),    High    level    summary    of    background    ozone    workshop,
https://www.epa.gov/sites/production/files/2016-03/documents/bgo3-high-level-
summary.pdf.
van der Werf, G. R., Randerson, J. T., Giglio, L., Collatz, G. J., Mu, M., Kasibhatla, P. S., Morton,
D. C., DeFries, R. S., Jin, Y., and van Leeuwen, T. T. (2010), Global fire emissions and the
contribution of deforestation, savanna, forest, agricultural, and peat fires (1997–2009), Atmos.
Chem. Phys., 10, 11707-11735, doi:10.5194/acp-10-11707-2010.
van Noije, T. P. C., Eskes, H. J., Dentener, F. J., Stevenson, D. S., Ellingsen, K., Schultz, M. G.,
Wild, O., Amann, M., Atherton, C. S., Bergmann, D. J., Bey, I., Boersma, K. F., Butler, T.,
Cofala, J., Drevet, J., Fiore, A. M., Gauss, M., Hauglustaine, D. A., Horowitz, L. W., Isaksen,
I. S. A., Krol, M. C., Lamarque, J.-F., Lawrence, M. G., Martin, R. V., Montanaro, V., Müller,
J.-F., Pitari, G., Prather, M. J., Pyle, J. A., Richter, A., Rodriguez, J. M., Savage, N. H., Strahan,
S. E., Sudo, K., Szopa, S., and van Roozendael, M. (2006), Multi-model ensemble simulations
of tropospheric $NO_2$ compared with GOME retrievals for the year 2000, Atmos. Chem. Phys.,
6, 2943-2979, doi:10.5194/acp-6-2943-2006.
Verstraeten, W. W., K. F. Boersma, J. Zörner, M. A. F. Allaart, K. W. Bowman, and J. R. Worden
(2013), Validation of six years of TES tropospheric ozone retrievals with ozonesonde
measurements: Implications for spatial patterns and temporal stability in the bias, Atmos. Meas.
Tech., 6, 1413–1423.
Verstraeten, W.W., J. L. Neu, J. E. Williams, K. W. Bowman, J. R. Worden, and K. F. Boersma
(2015), Rapid increases in tropospheric ozone production and export from China, Nature
Geoscience, 8, 690–695, doi:10.1038/ngeo2493.
Wang, H., D. J. Jacob, P. L. Sager, D. G. Streets, R. J. Park, A. B. Gilliland, and A. van Donkelaar
(2009), Surface ozone background in the United States: Canadian and Mexican pollution
influences, Atmos. Environ., 43(6), 1310–1319, doi:10.1016/j.atmosenv.2008.11.036.
Wang, Y., Konopka, P., Liu, Y., Chen, H., Müller, R., Plöger, F., Riese, M., Cai, Z., and Lü, D.
(2012), Tropospheric ozone trend over Beijing from 2002–2010: ozonesonde measurements
and modeling analysis, Atmos. Chem. Phys., 12, 8389-8399, doi:10.5194/acp-12-8389-2012.
Warneke, C., J. A. deGouw, J. S. Holloway, J. Peischl, T. B. Ryerson, E. Atlas, D. Blake, M.
Trainer, and D. D. Parrish (2012), Multiyear trends in volatile organic compounds in Los
Angeles, California: Five decades of decreasing emissions, J. Geophys. Res., 117, D00V17,
doi:10.1029/2012JD017899.
Warner, J. X., McCourt Comer, M., Barnet, C. D., McMillan, W. W., Wolf, W., Maddy, E., and
Sachse, G. (2007), A comparison of satellite tropospheric carbon monoxide measurements
from AIRS and MOPITT during INTEX-A, J. Geophys. Res., 112, D12S17,
doi:10.1029/2006JD007925, 2007.
Wiedinmyer, C., Akagi, S. K., Yokelson, R. J., Emmons, L. K., Al-Saadi, J. A., Orlando, J. J., and
Soja, A. J. (2011), The Fire INventory from NCAR (FINN): a high resolution global model to
estimate the emissions from open burning, Geosci. Model Dev., 4, 625-641, doi:10.5194/gmd-
4-625-2011.
Wigder, N.L., Jaffe, D.A., Herron-Thorpe, F.L., and Vaughan, J.K. (2013), Influence of daily
variations in baseline ozone on urban air quality in the United States Pacific Northwest, J.
Geophys. Res., 118, 3343–3354, doi: 10.1029/2012JD018738.
Wild, O., Fiore, A. M., Shindell, D. T., Doherty, R. M., Collins, W. J., Dentener, F. J., Schultz, M.
G., Gong, S., MacKenzie, I. A., Zeng, G., Hess, P., Duncan, B. N., Bergmann, D. J., Szopa,
S., Jonson, J. E., Keating, T. J., and Zuber, A. (2012), Modelling future changes in surface
ozone: a parameterized approach, Atmos. Chem. Phys., 12, 2037-2054, doi:10.5194/acp-12-
2037-2012.
Wu, S., B. N. Duncan, D. J. Jacob, A. M. Fiore, and O. Wild (2009), Chemical nonlinearities in
relating intercontinental ozone pollution to anthropogenic emissions, Geophys. Res. Lett., 36,
L05806, doi:10.1029/2008GL036607.
Yarwood, G., Rao, S., Yocke, M., and Whitten, G. (2005), Updates to the carbon bond chemical
mechanism: CB05. Final report to the US EPA, EPA Report Number: RT-0400675.
Zhang, L., Jacob, D. J., Boersma, K. F., Jaffe, D. A., Olson, J. R., Bowman, K. W., Worden, J. R.,
Thompson, A. M., Avery, M. A., Cohen, R. C., Dibb, J. E., Flock, F. M., Fuelberg, H. E.,
Huey, L. G., McMillan, W. W., Singh, H. B., and Weinheimer, A. J. (2008), Transpacific
transport of ozone pollution and the effect of recent Asian emission increases on air quality in
North America: an integrated analysis using satellite, aircraft, ozonesonde, and surface
observations, Atmos. Chem. Phys., 8, 6117-6136, doi:10.5194/acp-8-6117-2008.
Zhang, L., Jacob, D. J., Kopacz, M., Henze, D. K., Singh, K., and Jaffe, D. A. (2009),
Intercontinental source attribution of ozone pollution at western U.S. sites using an adjoint
method, Geophys. Res. Lett., 36, L11810, doi: 10.1029/2009GL037950.
Zhang, L., D. J. Jacob, N. V. Downey, D. A. Wood, D. Blewitt, C. C. Carouge, A. van Donkelaar,
D. B. A. Jones, L. T. Murray, and Y. Wang (2011), Improved estimate of the policy-relevant
background ozone in the United States using the GEOS-Chem global model with $1/2°×2/3°$
horizontal resolution over North America, Atmos. Environ., 45, 6769–6776, doi:
10.1016/j.atmosenv.2011.07.054.
Zhang, Q., Yuan, B., Shao, M., Wang, X., Lu, S., Lu, K., Wang, M., Chen, L., Chang, C.-C., and
Liu, S. C. (2014), Variations of ground-level $O_3$ and its precursors in Beijing in summertime
between 2005 and 2011, Atmos. Chem. Phys., 14, 6089-6101, doi:10.5194/acp-14-6089-2014.
Zhang, Y., Y. Chen, G. Sarwar, and K. Schere (2012), Impact of gas-phase mechanisms on
Weather Research Forecasting Model with Chemistry (WRF/Chem) predictions: Mechanism
implementation and comparative evaluation, J. Geophys. Res., 117, D01301,
doi:10.1029/2011JD015775.
Zoogman, P., X. Liu, R.M. Suleiman, W.F. Pennington, D.E. Flittner, J.A. Al-Saadi, B.B. Hilton,
D.K. Nicks, M.J. Newchurch, J.L. Carr, S.J. Janz, M.R. Andraschko, A. Arola, B.D. Baker,
B.P. Canova, C. Chan Miller, R.C. Cohen, J.E. Davis, M.E. Dussault, D.P. Edwards, J.
Fishman, A. Ghulam, G. González Abad, M. Grutter, J.R. Herman, J. Houck, D.J. Jacob, J.
Joiner, B.J. Kerridge, J. Kim, N.A. Krotkov, L. Lamsal, C. Li, A. Lindfors, R.V. Martin, C.T.
McElroy, C. McLinden, V. Natraj, D.O. Neil, C.R. Nowlan, E.J. O'Sullivan, P.I. Palmer, R.B.
Pierce, M.R. Pippin, A. Saiz-Lopez, R.J.D. Spurr, J.J. Szykman, O. Torres, J.P. Veefkind, B.
Veihelmann, H. Wang, J. Wang, and K. Chance (2017), Tropospheric emissions: Monitoring
of pollution (TEMPO), Journal of Quantitative Spectroscopy and Radiative Transfer, 186, 17-
39, doi: 10.1016/j.jqsrt.2016.05.008.

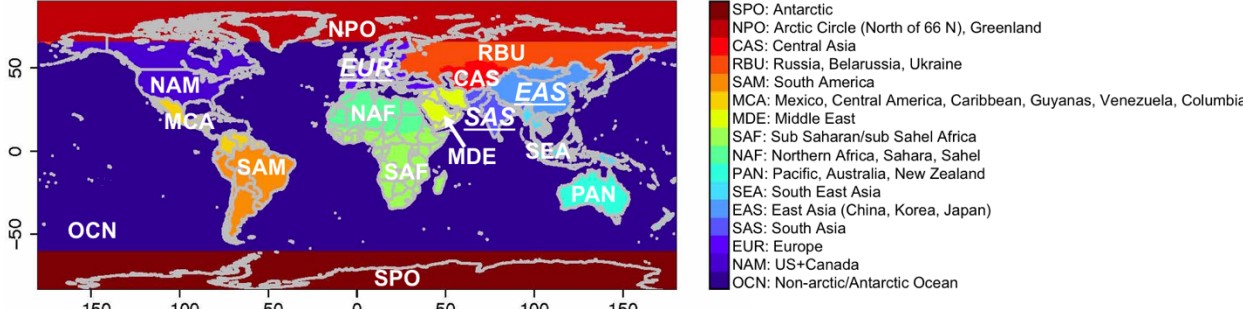

**Figure 1.** Definitions of the 16 source regions used in HTAP2 SR relationship study (More details
in Koffi et al., 2016). The map is plotted based on data on a 0.1°×0.1° resolution grid. We focus
in this study on the impact of anthropogenic pollution from selected non-North American source
regions (i.e., EAS, SAS, and EUR), whose names are underlined and in italic.

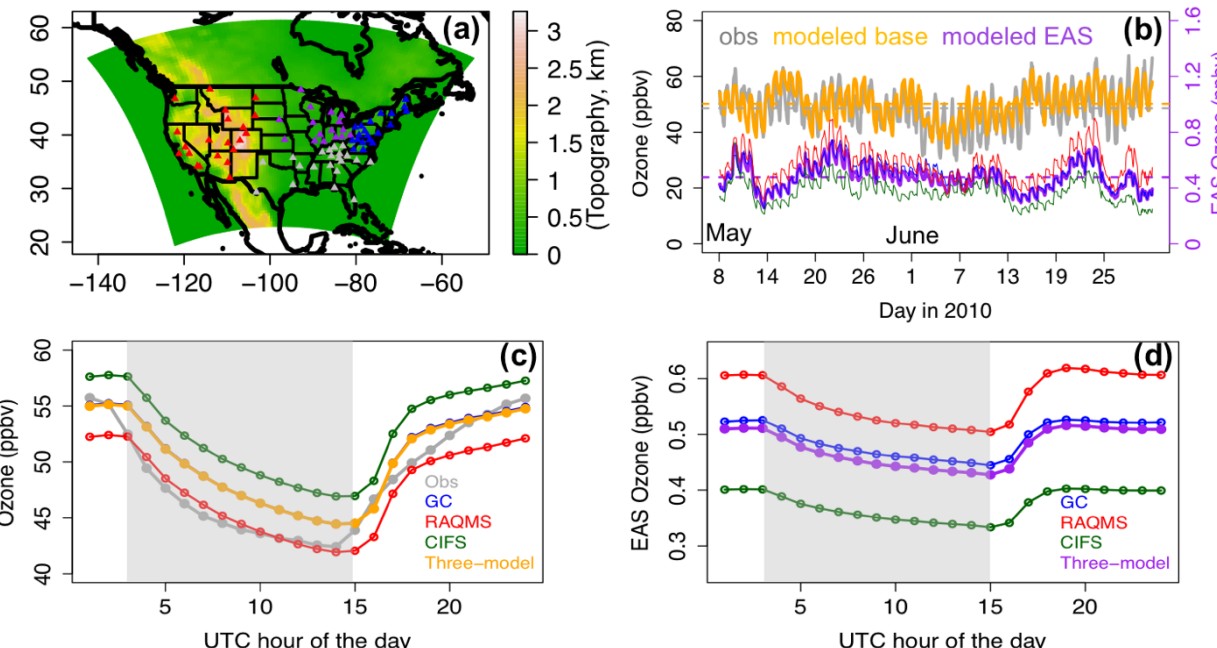

**Figure 2. (a)** The 60 km STEM NAM domain, colored by the model topography. The CASTNET
sites used in the STEM base $O_3$ evaluation are marked as triangles in different colors that identify
the subregions they belong to (red: western US; grey: southern US; purple: Midwest; blue:
northeastern US). **(b)** Evaluation of the STEM modeled (averaged from the three base simulations
using the GEOS-Chem, ECMWF C-IFS, and RAQMS base runs as the chemical boundary
conditions) hourly $O_3$ at the western US (i.e., EPA regions 8, 9, and 10) CASTNET sites.
Observations, modeled base $O_3$ and the modeled R($O_3$, EAS, 20%) are in grey, orange, and purple
lines, respectively. The horizontal dashed lines indicate the period mean values. The R($O_3$, EAS,
20%) values from STEM calculations using three different chemical boundary conditions are
shown separately in thin lines (blue: GEOS-Chem; red: RAQMS; green: C-IFS). The period-mean
diurnal variability of the STEM modeled **(c)** base and **(d)** R($O_3$, EAS, 20%) at the western US
CASTNET sites. The STEM calculations using three different chemical boundary conditions are
shown separately as well as averagely. Light grey-shaded areas indicate the local standard
nighttime (from 6/7 pm to 7/8 am).

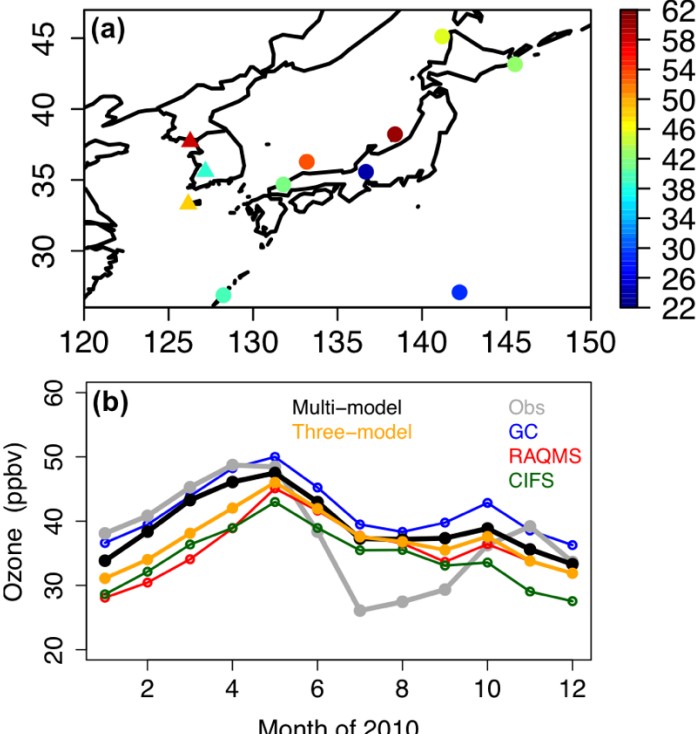

**Figure 3. (a)** May-June 2010 period mean surface $O_3$ observations in ppbv at eight Japanese (filled
circles) and three Korean (filled triangles) EANET sites. **(b)** Observed and modeled monthly-mean
surface $O_3$ in 2010 at all eleven EANET sites. The "Multi-model" and "Three-model" in the legend
indicate the mean values of all eight global models and only of the three boundary condition
models, respectively.

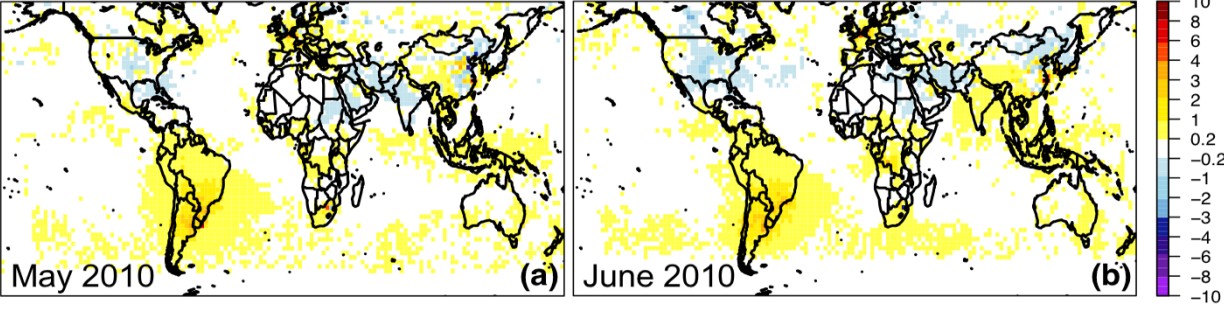

**Figure 4.** Evaluation of the GEOS-Chem adjoint base $NO_2$ product (recorded at near the satellite
overpassing time) with the OMI $NO_2$ columns. The differences between OMI and GEOS-Chem
(OMI-modeled) tropospheric $NO_2$ columns ($\times 10^{15}$ molec./cm$^2$) are shown for **(a)** May and **(b)** June
2010. Details of the comparison are included in Section 2.3.2.

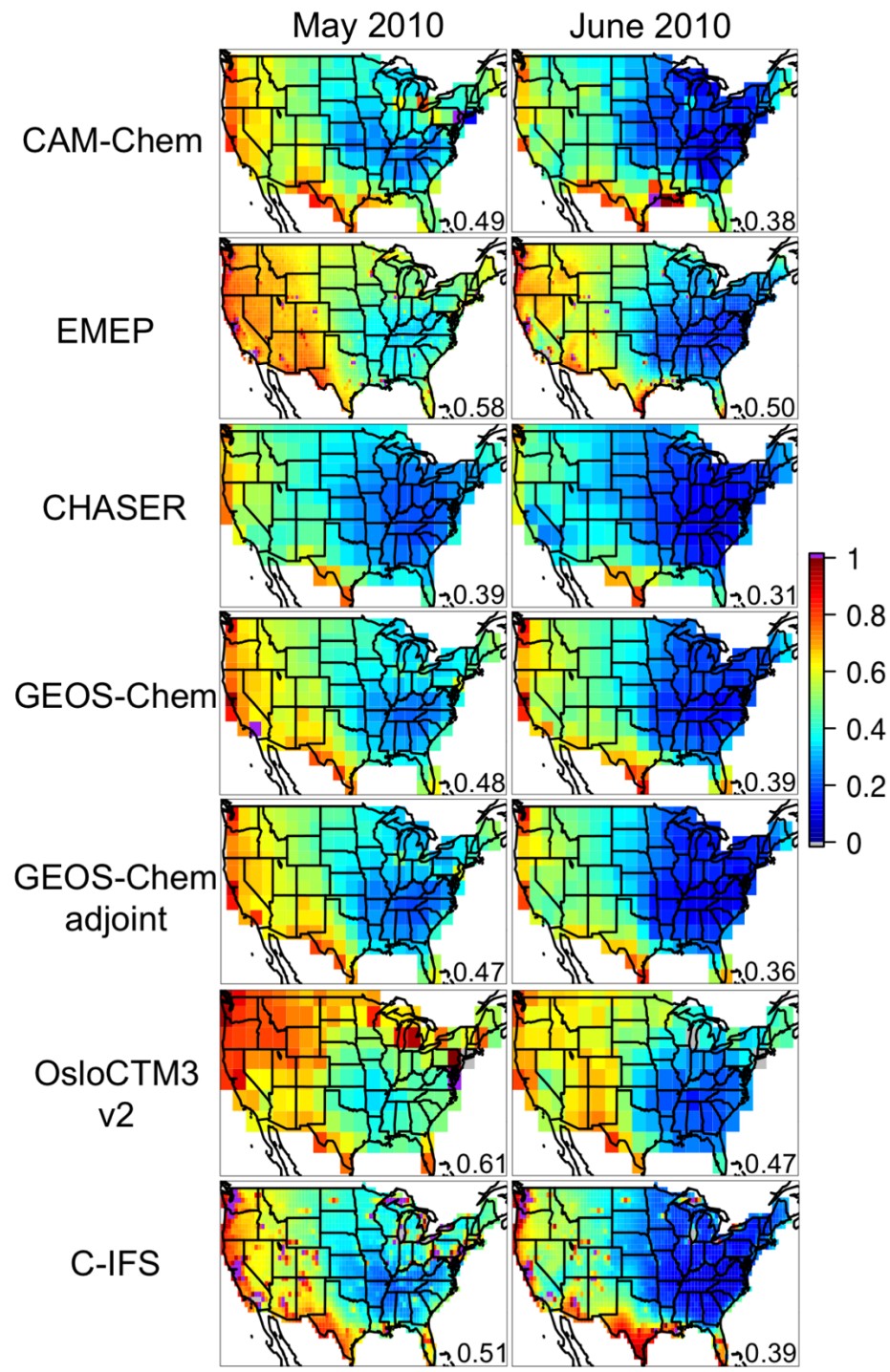

**Figure 5.** The RERER maps in May (left) and June (right) 2010 over the continental US, calculated
based on the monthly mean $O_3$ from multiple global models' base and emission sensitivity
simulations. The RERER metric (unitless) was defined in eq. (2) in the text. Values larger than 1
and smaller than 0 are shown in purple and grey, respectively. The US (including continental US
as well as Hawaii which is not shown in the plots) mean values are indicated for each panel at the
lower right corner. All models show declining RERER values from May to June, and the 7-model
mean RERER values for May and June 2010 are ~0.5 and ~0.4, respectively.

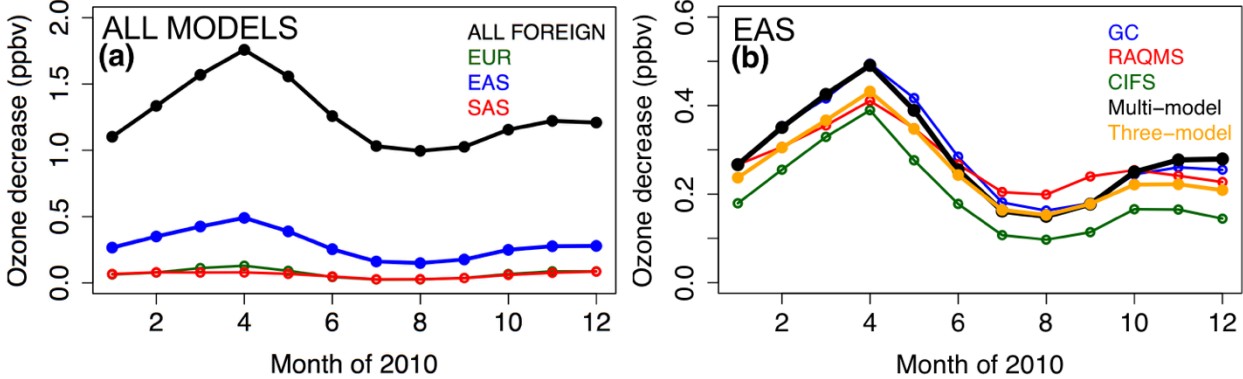

**Figure 6. (a)** North American (130-65°W; 20-50°N) mean $O_3$ sensitivity to 20% anthropogenic
emission reductions in various non-North American regions, averaged from multiple (six-eight,
see details in text) global models. **(b)** North American surface $R(O_3, EAS, 20\%)$ values, as
estimated by single (the three STEM boundary condition models) or multi- global model means.
The "Multi-model" and "Three-model" in the legend indicate the mean sensitivities of all eight
global models and only of the three boundary condition models, respectively.

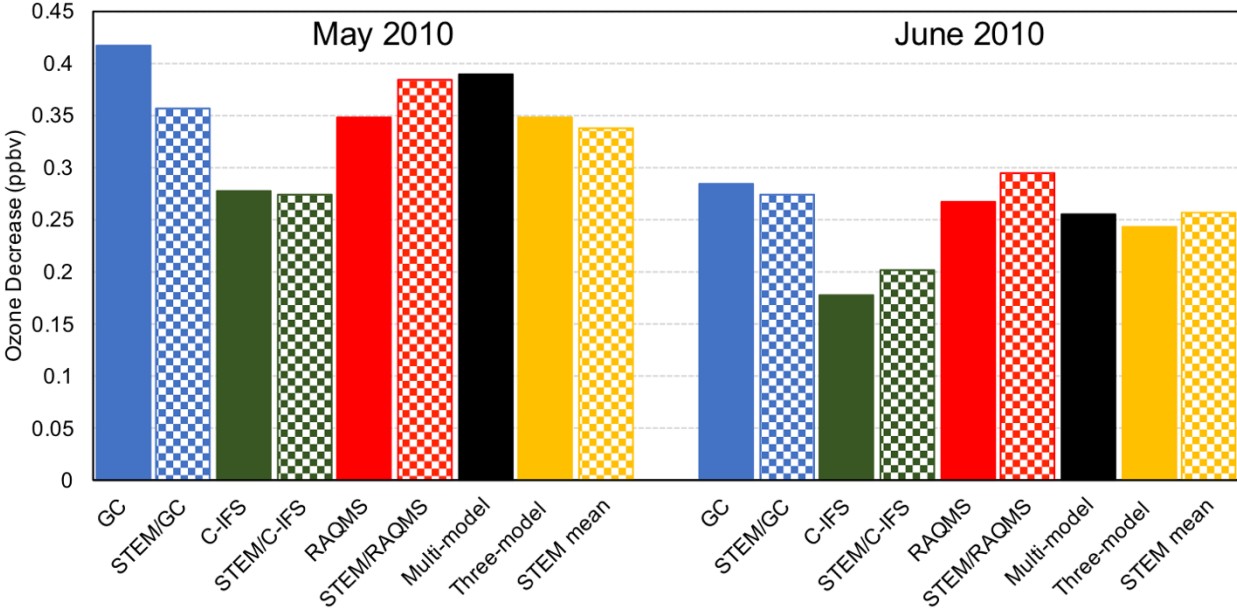

**Figure 7.** Monthly-mean North American (130-65°W; 20-50°N) surface $R(O_3, EAS, 20\%)$ values
from multiple global and regional model simulations for May (left) and June (right) 2010. STEM
model mean values were calculated from its hourly output from 8 May and on. The "Multi-model"
and "Three-model" in the legend indicate the mean sensitivities of all eight global models and only
of the three boundary condition models, respectively.

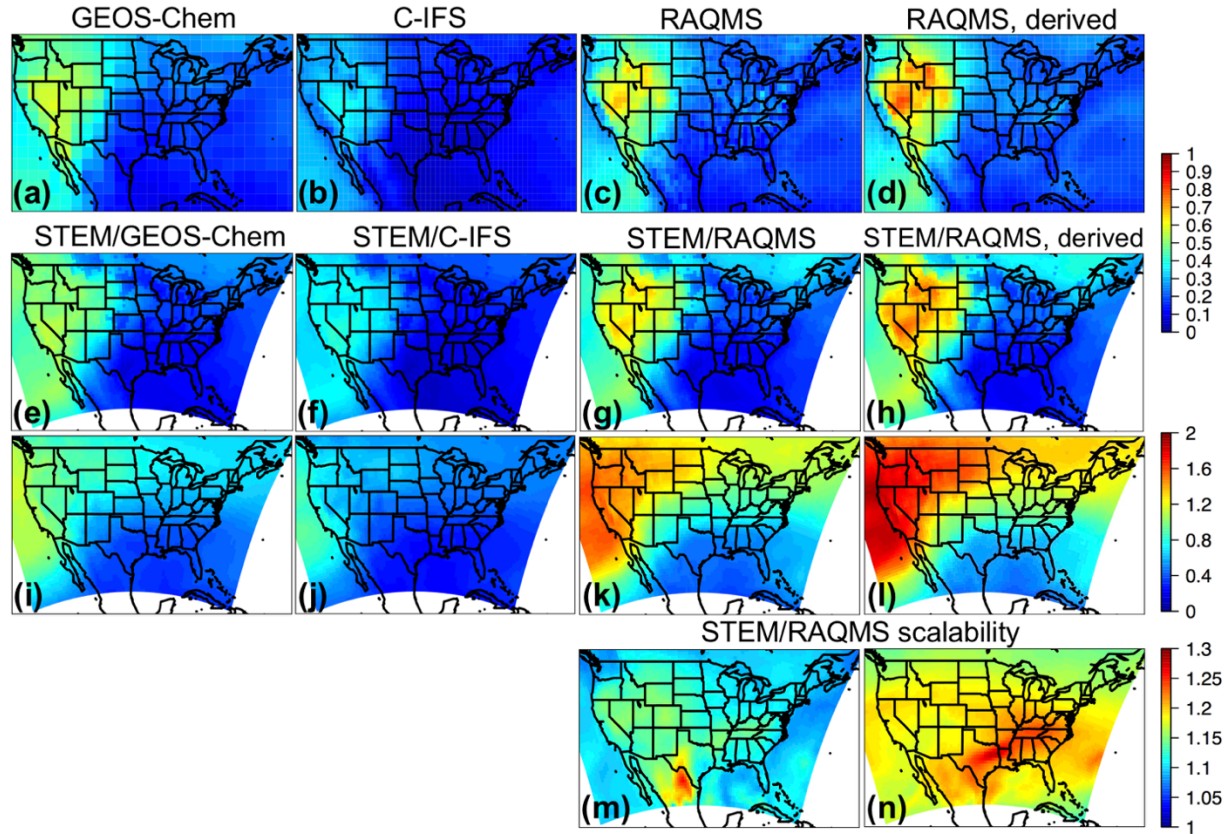


**Figure 8.** The monthly-mean R(O$_3$, EAS, 20%) in June 2010 for: **(a-d)** surface O$_3$ (ppbv) from the three boundary condition models, **(e-h)** STEM surface O$_3$ (ppbv), and **(i-l)** STEM column O$_3$ ($\times 10^{16}$ molecules/cm$^2$). R(O$_3$, EAS, 20%) values from the simulations associated with GEOS-Chem, ECMWF C-IFS, and RAQMS are shown in **(a;e;i)**, **(b;f;j)** and **(c;g;k)**, respectively. **(d;h;l)** show 1/5 of the R(O$_3$, EAS, 100%) from the simulations related to RAQMS. STEM/RAQMS-based "Scalability" S$_{O3}$ (eq. (3)) values over the NAM are shown for **(m)** surface and **(n)** column O$_3$.

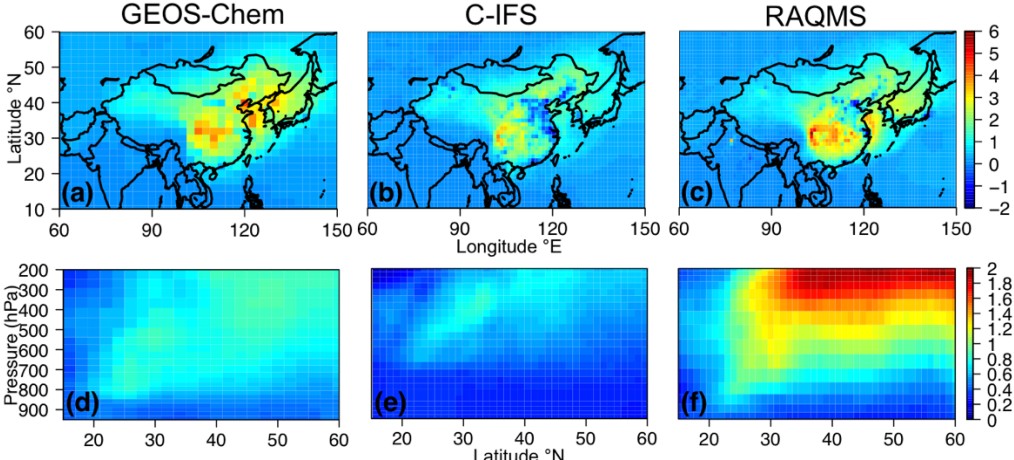

1547

**Figure 9.** The monthly-mean R(O$_3$, EAS, 20%) in ppbv in June 2010 from the three boundary condition models at the source and near the receptor regions: **(a-c)** surface O$_3$ in the East Asia; and **(d)** O$_x$ (GEOS-Chem) or **(e-f)** O$_3$ (ECMWF C-IFS and RAQMS) along the cross section of 135°W (near the west boundary of the STEM model domain as defined in Figure 2a).

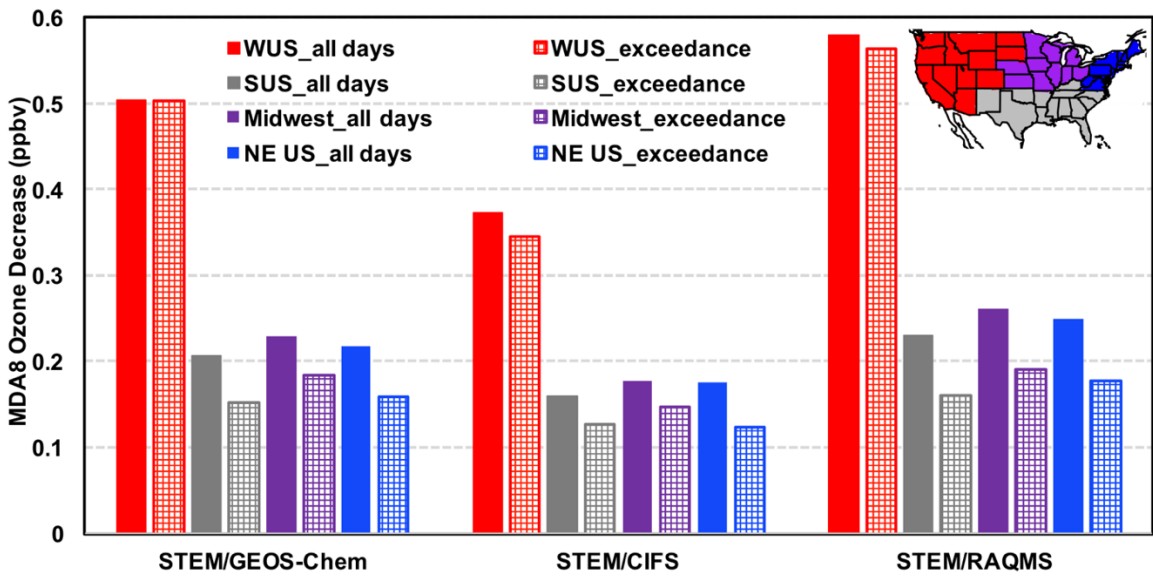

**Figure 10.** STEM R(MDA8, EAS, 20%) for May-June 2010 in four US subregions (defined in the
inset panel, also consistent with the definitions in Figures 2/S4 and Tables 2-3), averaged on all
days (bars with solid fill) and only on the days when the simulated total MDA8 $O_3$ concentrations
were over 70 ppbv (bars with grid pattern fill). The results from the STEM runs using GEOS-
Chem, ECMWF C-IFS and RAQMS boundary conditions are shown separately.

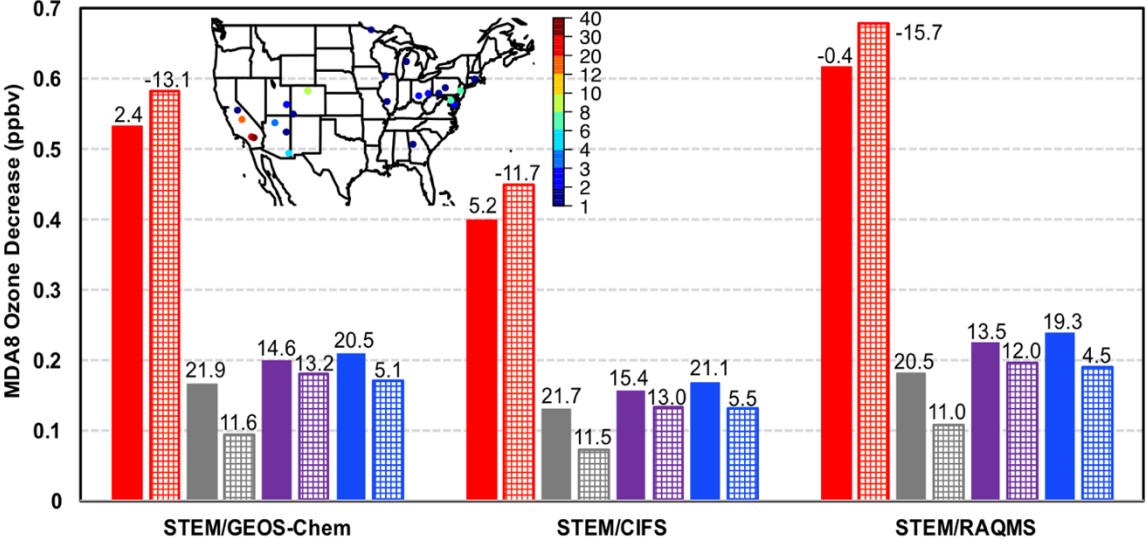

**Figure 11.** STEM R(MDA8, EAS, 20%) for May-June 2010 at the CASTNET sites in four US
subregions (same definition as in Figure 10 inset), averaged on all days (bars with solid fill) and
only on the days when the observed MDA8 $O_3$ concentrations were over 70 ppbv (bars with grid
pattern fill). The results from the STEM runs using GEOS-Chem, ECMWF C-IFS and RAQMS
boundary conditions are shown separately. Biases for the corresponding model base runs are
shown above the bar plots. Inset shows at various CASTNET sites the number of days when the
observed MDA8 $O_3$ concentrations were over 70 ppbv.

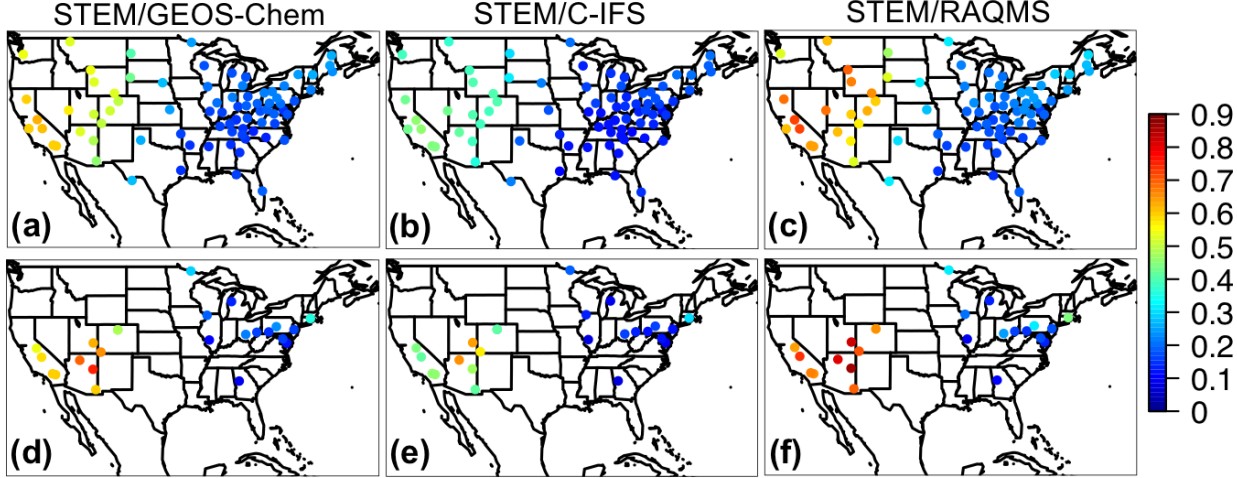

**Figure 12.** STEM R(MDA8, EAS, 20%) in ppbv for May-June 2010 at the CASTNET sites on **(a-**
**c)** all days and **(d-f)** the days when the observed MDA8 O₃ concentrations were over 70 ppbv. The
results from the STEM runs using **(a;d)** GEOS-Chem, **(b;e)** ECMWF C-IFS and **(c;f)** RAQMS
boundary conditions are shown separately.

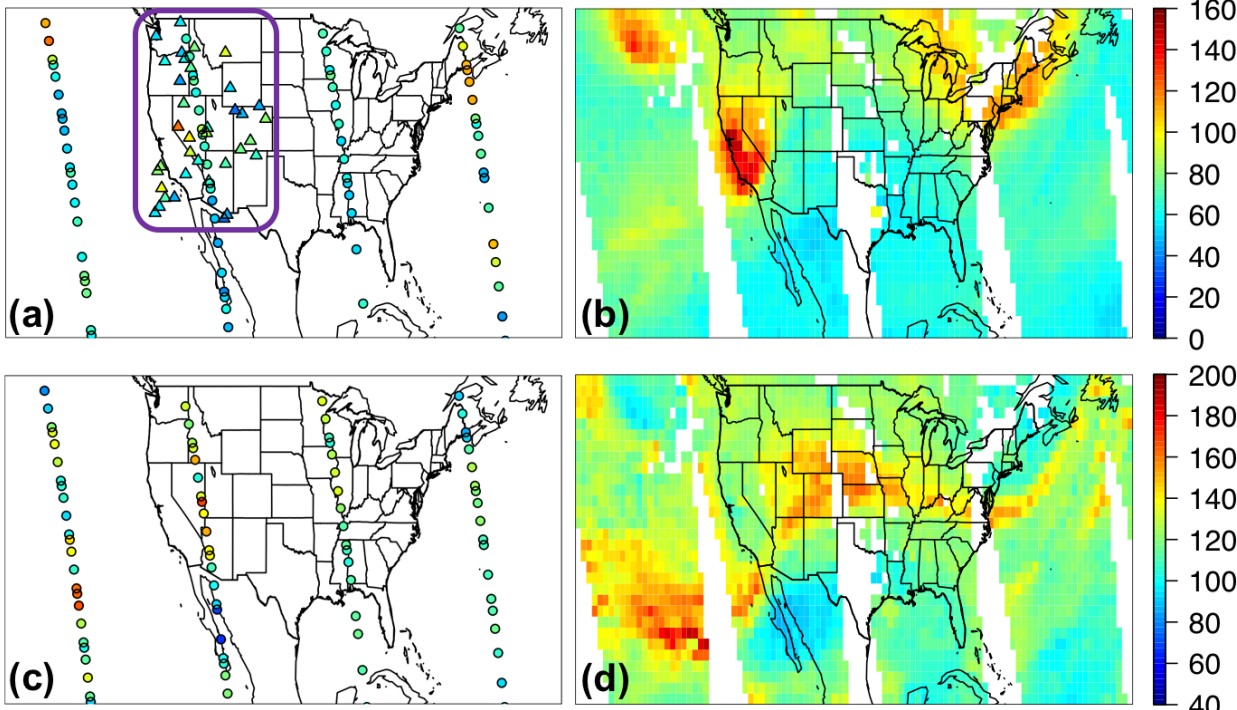

**Figure 13.** Case study of 9 May 2010: **(a-b)** Ozone (ppbv) and **(c-d)** CO (ppbv) at ~500 hPa from
the L2 **(a;c)** TES retrievals (circles) and **(b;d)** L3 AIRS products at early afternoon local time. The
L2 IASI O₃ (ppbv) at ~500 hPa retrieved using the TES algorithm (details in Section 2.3.2) at the
mid- morning local times is shown on panel (b) as triangles. The O₃ profiles within the purple box
in panel (a) were used in the model evaluation shown in Figure 14.

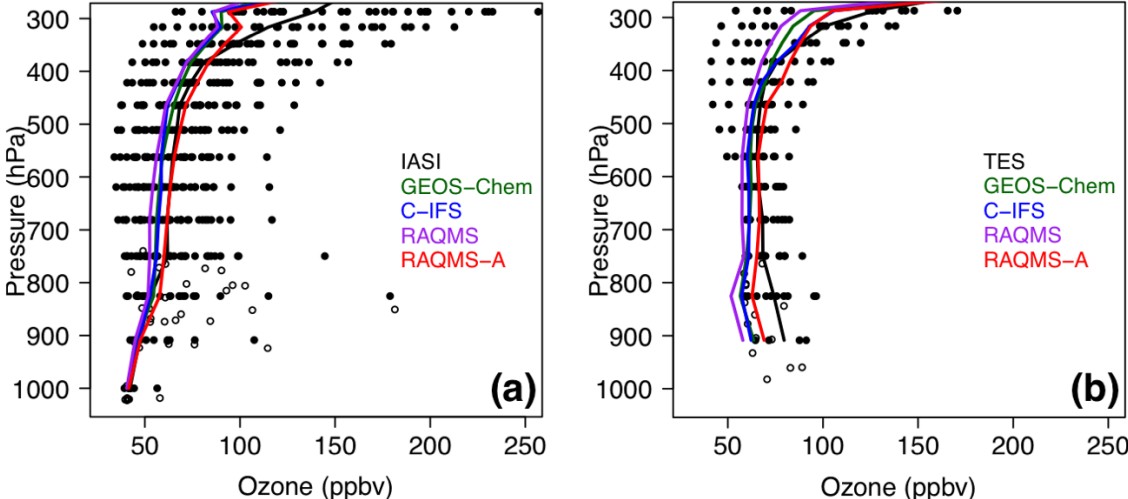

**Figure 14.** Case study of 9 May 2010: The comparisons between **(a)** IASI and **(b)** TES O$_3$ in the
western US with the simulated O$_3$ in the STEM runs using the GEOS-Chem (green), C-IFS (blue),
RAQMS (purple), and assimilated RAQMS (red) boundary conditions. The O$_3$ profiles within the
purple box in Figure 10a were used in the evaluation. Observation operators were applied in the
comparisons (details in Section 2.3.2). Solid and open dots are TES/IASI data at the TES retrieval
reporting levels and at the variable surface pressure levels, respectively. Solid lines are median O$_3$
profiles from the satellite observations and the different STEM simulations, calculated only on the
TES retrieval reporting levels.

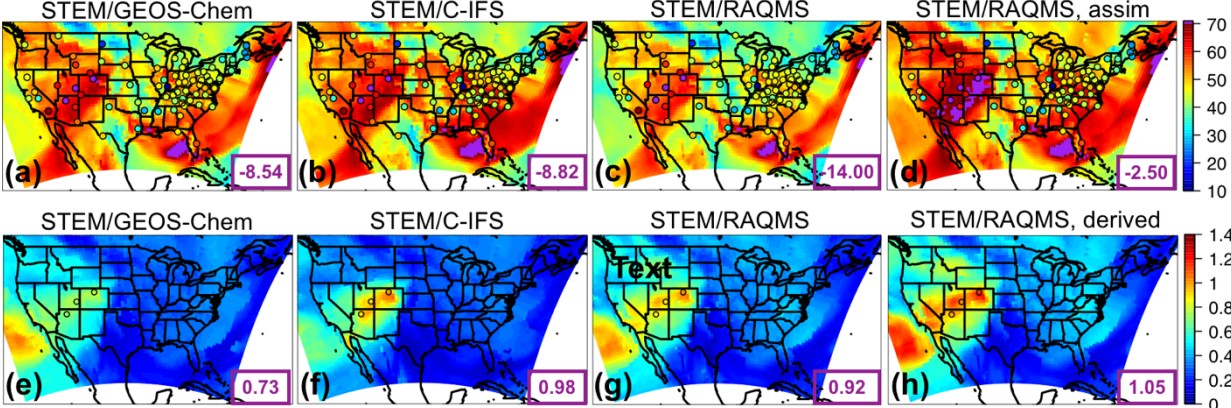

**Figure 15.** Case study of 9 May 2010: **(a-d)** Surface MDA8 total O$_3$ and **(e-h)** surface R(MDA8,
EAS, 20%) from the STEM simulations using the **(a;e)** GEOS-Chem, **(b;f)** ECMWF C-IFS, and
**(c;g)** RAQMS free run as the boundary conditions. **(d)** Surface MDA8 total O$_3$ in a STEM base
simulation using the RAQMS assimilation run as the boundary conditions. CASTNET
observations are overlaid in filled circles in panels (a-d). **(h)** 1/5 of the surface R(MDA8, EAS,
100%) from STEM/RAQMS simulations. The conditions at ~400-500 hPa are shown in Figure S5.
Purple numbers at the lower right corners of **(a-d)** and **(e-h)** are mean model biases and mean
R(MDA8, EAS, 20%) values in ppbv at the three mountain sites (Grand Canyon NP, AZ;
Canyonlands NP, UT; and Rocky Mountain NP, CO) where O$_3$ exceedances were observed on this
day. The locations of these sites are shown in panel **(e-h)** as open circles.

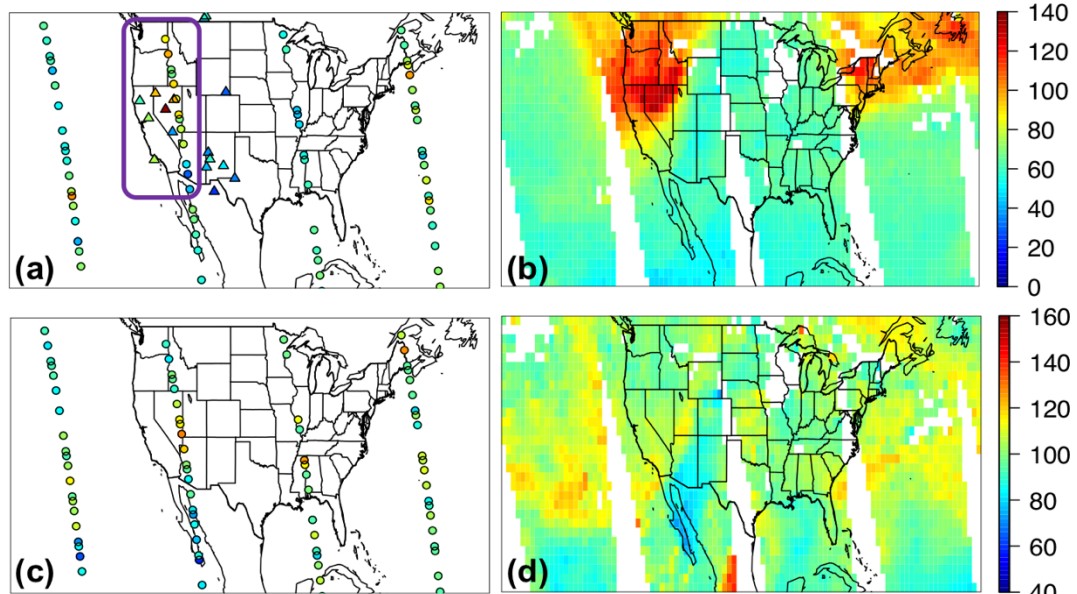

**Figure 16.** Same as Figure 13, but for a case study of 10 June 2010.

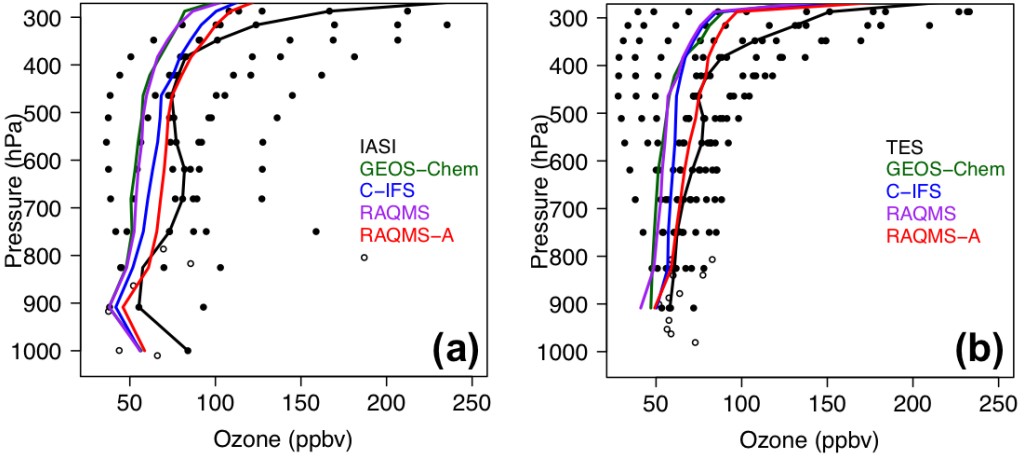

**Figure 17.** Same as Figure 14, but for a case study of 10 June 2010.

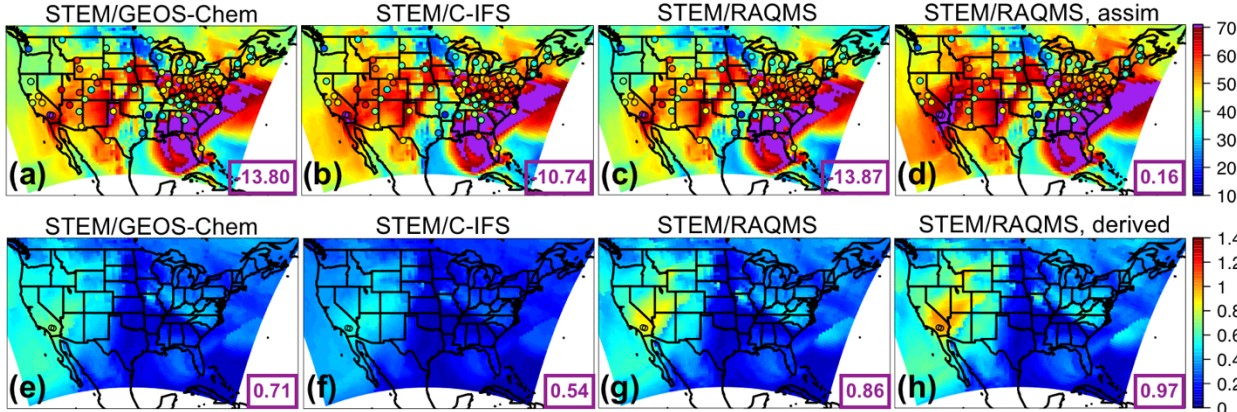

**Figure 18.** Same as Figure 15, but for a case study of 10 June 2010. The CASTNET sites with $O_3$
exceedances on this day are Converse Station and Joshua Tree NP in southern California.
**Table 1a.** HTAP2 base and sensitivity simulations by various global models. The STEM boundary
condition models are highlighted in bold.

| Global model, Resolution: lon×lat×vertical layer, (References) | BASE | EASALL (-20%) | EASALL (-100%) | GLOALL (-20%) | NAMALL (-20%) | EURALL (-20%) | SASALL (-20%) |
|---|---|---|---|---|---|---|---|
| CAM-Chem, 2.5°×1.9°×56 (Tilmes et al., 2016) | ✓ | ✓ | | ✓ | ✓ | ✓ | ✓ |
| CHASER T42, ~2.8°×2.8°×32 (Sudo et al., 2002) | ✓ | ✓ | | ✓ | ✓ | ✓ | ✓ |
| EMEP rv48, 0.5°×0.5°×20 (Simpson et al., 2012) | ✓ | ✓ | | ✓ | ✓ | ✓ | ✓ |
| **SNU GEOS-Chem v9-01-03, 2.5°×2°×47** (Park et al., 2004; http://iek8wikis.iek.fz-juelich.de/HTAPWiki/WP 2.3?action=AttachFile&do =view&target=_README _GEOS-Chem.pdf) | ✓ | ✓ | | ✓ | ✓ | | |
| CU-Boulder GEOS-Chem adjoint v35f, 2.5°×2°×47 (Henze et al., 2007) | ✓ | ✓ | | ✓ | ✓ | ✓ | ✓ |
| **RAQMS, 1°×1°×35, free running** (Pierce et al., 2007, 2009) | ✓ | ✓ | ✓ | | | | |
| **RAQMS, 1°×1°×35, with satellite assimilation** (Pierce et al., 2007, 2009) | ✓ | | | | | | |
| OsloCTM3 v2, ~2.8°×2.8°×60 (Søvde et al., 2012) | ✓ | ✓ | | ✓ | ✓ | ✓ | ✓ |
| **ECMWF C-IFS,** ~0.7°×0.7°×54/**1.125°×1.1 25°×54, as the STEM chemical boundary conditions** (Flemming et al., 2015) | ✓ | ✓ | | ✓ | ✓ | ✓ | ✓ |

Acronyms:
CAM-Chem: Community Atmosphere Model with Chemistry
C-IFS: Composition-Integrated Forecasting System
ECMWF: European Center for Medium range Weather Forecasting
EMEP: European Monitoring and Evaluation Programme
GEOS-Chem: Goddard Earth Observing System with Chemistry
RAQMS: Realtime Air Quality Modeling System
SNU: Seoul National University
**Table 1b.** STEM regional simulations for HTAP2

| Boundary condition model, Resolution: lon×lat×vertical layer | BASE | EASALL (-20%) | EASALL (-100%) |
|---|---|---|---|
| SNU GEOS-Chem v9-01-03, 2.5°×2°×47 | ✓ | ✓ | |
| RAQMS, 1°×1°×35, free running | ✓ | ✓ | ✓ |
| RAQMS, 1°×1°×35, with satellite assimilation | ✓ | | |
| ECMWF C-IFS, 1.125°×1.125°×54 | ✓ | ✓ | |


**Table 1c.** STEM and its boundary condition models' key inputs and chemical mechanisms, with
references. More details on the models can be found in Table 1a and the text.

| Model | Meteorology | Biogenic VOCs; $NO_x$ | Lightning | Biomass Burning | Chemical Mechanism |
|---|---|---|---|---|---|
| GEOS-Chem | GEOS-5 | MEGAN v2.1 (Guenther et al., 2012); Wang et al., 2009 | based on GEOS-5 deep convective cloud top heights and climatological observations (Murray et al., 2012) | GFED v3.0 (van der Werf et al., 2010) | GEOS-Chem standard $NO_x$-$O_x$-hydrocarbon-aerosol (http://acmg.seas.harvard.edu/geos/doc/archive/man.v9-01-03/appendix_1.html) |
| RAQMS | Online (Pierce et al., 2007) | | | | CB-IV (Gery et al., 1989) with adjustments |
| ECMWF C-IFS | IFS | MEGAN-MACC, (Sindelarova et al., 2014); POET database for 2000 (Granier et al., 2005) | based on IFS convective precipitation (Meijer et al., 2001) | GFAS v1.0 (Kaiser et al., 2012) | CB05 (Yarwood et al., 2005) |
| STEM | WRF-ARW v3.3.1 | WRF-MEGAN v2.1 | based on scaled WRF convective precipitation | FINN v1.0 (Wiedinmyer et al., 2011) | SAPRC99 (Carter, 2000) |

Acronyms:
CB: Carbon Bond
FINN: Fire INventory from NCAR
GFAS: Global Fire Assimilation System
GFED: Global Fire Emissions Database
IFS: Integrated Forecasting System
MACC: Monitoring Atmospheric Composition and Climate
MEGAN: Model of Emissions of Gases and Aerosols from Nature
POET: Precursors of Ozone and their Effects in the Troposphere
WRF-ARW: Advanced Research Weather Research and Forecasting Model
**Table 2a.** Evaluation of the period mean (1 May-30 June, 2010) multi- global model free
simulations against the CASTNET observations, only at the sites where 95% of the hourly $O_3$
observations are available. Evaluation of the individual models is summarized in Table 2b.

| Subregion | US EPA regions contained | Number of sites | Mean bias (ppbv) | | RMSE (ppbv) | |
|---|---|---|---|---|---|---|
| | | | 3 BC[a] models | 8 global models | 3 BC models | 8 global models |
| Western US | 8, 9, 10 | 19 | -5.68 | -2.52 | 10.37 | 7.05 |
| Southern US | 4, 6 | 18 | 11.61 | 10.24 | 13.62 | 11.96 |
| Midwest | 5, 7 | 13 | 8.03 | 7.66 | 9.16 | 8.67 |
| Northeast | 1, 2, 3 | 17 | 9.55 | 10.63 | 10.28 | 11.24 |
| All | 1-10 | 67 | 5.49 | 6.22 | 11.11 | 9.96 |

[a]BC: Boundary Conditions

**Table 2b.** Evaluation of the period mean (May-June 2010) global model free simulations against
the EANET and CASTNET observations. The STEM boundary condition models are highlighted
in bold.

| Network | Number of sites | RMSE (ppbv) | | | | | | | |
|---|---|---|---|---|---|---|---|---|---|
| | | CAM-Chem | EMEP | CHASER | **SNU GEOS-Chem** | GEOS-Chem adjoint | **RAQMS** | OsloCTM3 v2 | **C-IFS** |
| CASTNET | 67 | 13.30 | 11.61 | 15.43 | **15.55** | 13.48 | **9.32** | 11.05 | **11.00** |
| EANET | 11 | 10.38 | 9.96 | 11.39 | **9.18** | 11.04 | **8.60** | 12.97 | **10.86** |


**Table 2c.** Evaluation of the period mean (May-June 2010) multi- global model free simulations
against the EANET observations in Japan and Korea. Evaluation of the individual models is
summarized in Table 2b.

| Country | Number of sites | Mean bias (ppbv) | | RMSE (ppbv) | |
|---|---|---|---|---|---|
| | | 3 BC[a] models | 8 global models | 3 BC models | 8 global models |
| Japan | 8 | 0.36 | 1.01 | 8.77 | 9.25 |
| Korea | 3 | 1.14 | 3.98 | 8.37 | 10.51 |
| All | 11 | 0.57 | 1.82 | 8.66 | 9.61 |

[a]BC: Boundary Conditions
**Table 3a.** Evaluation of the hourly STEM simulated total $O_3$ (averaged from the three base
simulations that used the different free-running boundary conditions) against the CASTNET
surface observations for 8 May-30 June, 2010. The subregional mean R($O_3$, EAS, 100%) and its
correlation coefficient with the observed $O_3$ are also shown.

| Subregion | US EPA regions contained | Number of sites | Mean elevation (km): actual/model | Mean bias (ppbv) | RMSE (ppbv) | Correlation (model base; obs) | Correlation (obs; modeled EAS) | Mean EAS sensitivity (ppbv) |
|---|---|---|---|---|---|---|---|---|
| Western US | 8, 9, 10 | 22 | 1.75/ 1.71 | 1.60 | 4.86 | 0.76 | 0.34 | 0.48 |
| Southern US | 4, 6 | 22 | 0.38/ 0.31 | 20.33 | 22.13 | 0.58 | 0.27 | 0.15 |
| Midwest | 5, 7 | 16 | 0.29/ 0.28 | 15.64 | 17.97 | 0.70 | 0.15 | 0.17 |
| Northeast | 1, 2, 3 | 20 | 0.36/ 0.26 | 20.94 | 24.16 | 0.47 | 0.17 | 0.21 |
| All | 1-10 | 80 | 0.73/ 0.68 | 16.17 | 18.30 | 0.66 | 0.13 | 0.20 |


**Table 3b.** Evaluation of the hourly STEM simulated total $O_3$ (separately for three base simulations
that used the different free-running boundary conditions) against the CASTNET surface
observations for 8 May-30 June, 2010.

| Subregion | US EPA regions contained | Number of sites | Mean bias (ppbv)/RMSE (ppbv)/Correlation (model base; obs) | | |
|---|---|---|---|---|---|
| | | | SNU GEOS-Chem | C-IFS | RAQMS |
| Western US | 8, 9, 10 | 22 | 1.68/4.83/0.77 | 4.16/6.63/0.70 | -1.03/4.81/0.76 |
| Southern US | 4, 6 | 22 | 21.18/22.94/0.57 | 20.34/22.07/0.60 | 19.48/21.45/0.56 |
| Midwest | 5, 7 | 16 | 15.77/18.17/0.70 | 16.41/18.46/0.72 | 14.73/17.35/0.69 |
| Northeast | 1, 2, 3 | 20 | 21.25/24.36/0.47 | 21.86/24.80/0.48 | 19.71/23.40/0.45 |
| All | 1-10 | 80 | 16.57/18.62/0.66 | 16.89/18.84/0.67 | 15.03/17.52/0.64 |


**Table 4.** The ranges and standard deviations (ppbv, separated by ";") of R(O$_3$, *source region*, 20%)
by 6-8 global models (defined in eq. (1a-d)), summarized by months in 2010. The monthly multi-
model mean values are shown in Figures 5-6.

| Month/ Source region | All Foreign/ Non-NAM (ppbv) | EUR (ppbv) | EAS (ppbv) | SAS (ppbv) |
|---|---|---|---|---|
| Jan | 0.38-1.69; 0.41 | 0.002-0.12; 0.05 | 0.02-0.72; 0.24 | 0.001-0.11; 0.04 |
| Feb | 0.92-2.07; 0.37 | 0.02-0.15; 0.05 | 0.16-0.91; 0.28 | 0.02-0.12; 0.04 |
| Mar | 1.30-2.37; 0.38 | 0.07-0.21; 0.06 | 0.24-1.03; 0.30 | 0.03-0.12; 0.03 |
| Apr | 1.42-2.46; 0.33 | 0.09-0.23; 0.05 | 0.33-1.07; 0.28 | 0.04-0.12; 0.03 |
| May | 1.24-1.91; 0.21 | 0.06-0.17; 0.04 | 0.24-0.75; 0.19 | 0.05-0.11; 0.02 |
| Jun | 1.03-1.41; 0.13 | 0.03-0.07; 0.02 | 0.14-0.39; 0.09 | 0.04-0.07; 0.01 |
| Jul | 0.86-1.18; 0.13 | 0.02-0.04; 0.01 | 0.08-0.22; 0.06 | 0.01-0.04; 0.01 |
| Aug | 0.80-1.19; 0.13 | 0.01-0.04; 0.01 | 0.07-0.20; 0.05 | 0.02-0.04; 0.01 |
| Sep | 0.85-1.18; 0.13 | 0.03-0.05; 0.01 | 0.10-0.25; 0.06 | 0.02-0.06; 0.01 |
| Oct | 0.96-1.31; 0.14 | 0.04-0.10; 0.02 | 0.17-0.42; 0.09 | 0.03-0.08; 0.02 |
| Nov | 0.90-1.48; 0.19 | 0.05-0.15; 0.04 | 0.17-0.54; 0.14 | 0.04-0.10; 0.02 |
| Dec | 0.73-1.67; 0.29 | 0.03-0.18; 0.05 | 0.14-0.66; 0.19 | 0.04-0.12; 0.03 |
