# Peer review of "Impact of Intercontinental Pollution Transport on North American Ozone Air Pollution: An HTAP Phase 2 Multi-model Study"

_Atmospheric Chemistry and Physics, 2016_

## Short Comment (SC1) · 9 Dec 2016

1. Lines 93-100, Page 3:

Regarding Asian influence on US ozone trends, please consider citing the following papers and discuss their findings:

Lin, M., L.W. Horowitz, O.R. Cooper, D. Tarasick, S. Conley, L.T. Iraci, B. Johnson, T. Leblanc, I. Petropavlovskikh, E.L. Yates (2015): Revisiting the evidence of increasing springtime ozone mixing ratios in the free troposphere over western North America, Geophysical Research Letter, 42, doi:10.1002/2015GL065311

Lin, M., W. Horowitz, R. Payton, A.M. Fiore, G. Tonnesen. US surface ozone trends and

extremes over 1980-2014: Quantifying the roles of rising Asian emissions, domestic controls, wildfires, and climate. Atmos. Chem. Phys. Discuss., doi:10.5194/acp-2016-1093, 2016

You cited Cooper et al. (2010, Nature). But Lin et al. (2015 GRL) investigated the representativeness of ozone trends derived from sparse measurements reported by Cooper et al. They found that sampling biases can substantially influence calculated ozone trends.

2. The multi-model results presented in this article are based on the spring of 2010 following strong El Nino conditions. I think it would be useful to the readers if you can discuss the representativeness of your results on inter-annual context. There are studies showing that long-transport transport of Asian pollution is stronger during El Nino springs due to the eastward extension and equator-ward shift of the subtropical jet stream (e.g., Lin et al., 2014, Nature Geoscience).

---

## Referee Comment (RC1) · G.S.T Tonnesen (Referee) · 12 Dec 2016

General comments

Most of the paper is focused on comparison of monthly mean model results with very limited evaluation of model performance and no analysis of the causes of the differences among the global model simulations. This analysis is not substantially different from previous HTAP studies and is not informative. I suggest moving most of the text and plots that discuss monthly mean model results to the supplement, and instead, the authors should evaluate and compare model performance on several short term episodes that are more relevant to ozone transport and air quality planning.

[Figure]

The most interesting aspect of the paper is the section that addresses the May 9, 2010 O3 episode. I suggest including a more detailed discussion of this event, including an assessment of the relative contributions of stratospheric O3 and international transport of O3 for this event. Given that all of the global models performed poorly for this event (with the exception of RAQMS with data assimilation), a key finding could be that currently available global models do not perform well for some high ozone events. I also suggest performing additional analysis for at least one other high O3 event during summer 2010 to contrast with the May 9 event. By performing a more detailed evaluation and comparison of the different global models (and the couple STEM/Global model simulations) for specific episodes, the authors can more directly evaluate model performance and the suitability of the individual global models for use as boundary condition data in higher resolution regional models.

I suggest deleting the text that asserts that the use of an ensemble of global models is a preferred approach. The citation (U.S.EPA 2016) is summary of comments at a public meeting and should not be used as citation because the comments were not peer reviewed and do not reflect the consensus of the meeting participants. A better citation would be the EPA whitepaper on background ozone which was reviewed within EPA and is available at https://www.epa.gov/ozone-pollution/background-ozone-workshop-and-information. The whitepaper does not recommend the use of multi-model means to reduce uncertainty. The Li et al. 2016 citation is an analysis of visibility trends and does not evaluate multi-model results. Moreover, there is no valid theoretical basis to assume that the average of poorly performing models will be more accurate than the best performing individual model for key atmospheric processes. While it is possible that, by chance, the average of several poorly performing models will better match observations, the average may still inaccurately represent the individual processes that contribute to O3 and the sensitivity of O3 to emissions reductions. While it might be true that positive and negative bias errors cancel when averaging multiple model results for monthly or seasonal means, this does not necessarily indicate that the multi-model average represents O3 more accurately for episodic events that are of interest to the

air quality planning community. A better approach would be to evaluate and compare models at the process level and specifically for high O3 episodes, and then select the best performing individual model.

Detailed comments

Line 63: Opening sentence is awkward. There is no clear link of the uneven distribution to the health/ecosystem impacts of O3. Also, the uneven distribution of O3 is mostly caused by strong concentration gradients in precursor emissions, but this sentence only identifies the O3 lifetime as a cause of the distribution. Suggest rewriting with a focus on the high mixing ratios, not just the distribution.

Line 73: "to control the emissions of its precursors from these various sources". Not clear what "these various sources" refers to here. The previous sentence identified the stratosphere and local to distant emissions sources, so presumably this sentence is suggesting that there will be benefits of control of both local and international emissions sources, but this sentence then goes on to list precursors categories (VOC, NOx, CO) without reference to local vs international or biogenic vs. anthropogenic. I can infer what the authors mean, but the introductory paragraphs are awkwardly written and potentially confusing to a reader who is not an expert.

Line 75: Also include methane in this list

Line 80: background and baseline are not the same. See Cooper et al. for their definition of baseline, and EPA white paper (link below) for definition of U.S. background ozone. Briefly, baseline O3 (as defined by Cooper et al.) can include contributions from upwind U.S. anthropogenic precursor emissions while U.S. background ozone excludes all U.S. anthropogenic emissions.

Line 82: Given how the authors defined baseline/background O3, this statement is problematic: "below which the air quality standard is not recommended to be set". Baseline O3 can be elevated in some areas because of transport of anthropogenic

precursors and O3 from upwind U.S. states. It is appropriate to set the NAAQS below the baseline O3 level in these areas because the elevated baseline O3 is being addressed by emissions reductions in upwind states. I recommend breaking this very long sentence into several sentences that describe each of the points identified more clearly and more accurately.

Line 90: "It has been revealed" is awkward – "revealed" has other connotations. Suggest "It has been found".

Line 95: "A better understanding of the processes that determine the O3 distributions" Note that the authors have not yet clearly and comprehensively described the processes. They should describe the roles of stratospheric (both routine contributions and discrete intrusion events), biogenic precursors, wildfires, and anthropogenic precursors. We are especially concerned with conditions in which the mixing ratio exceeds the NAAQS, so it is not only the distribution but also the mixing ratio that is important.

Line 96: delete "for recent years". This will be useful for all past years and for future predictions.

Lines 110-112: "Large intermodel diversity was found in the simulated total O3 and the intercontinentally transported pollution for the chosen SR pairs in the northern midlatitudes, indicating the challenges with simulations by any individual model to accurately represent the key atmospheric processes." The conclusion that no individual model performs well is not supported by a finding of inter-model diversity. For example, it is possible that one model performs well while other models do not. The authors need to cite results of the individual model performance evaluations to support the statement that no model perform well.

Lines 113-116: The citation (U.S.EPA 2016) is summary of comments at a public meeting and should not be used as citation because the comments were not peer reviewed and do not reflect the consensus of the meeting participants. A better citation would be the EPA whitepaper on background ozone which did receive review within EPA and

is available at https://www.epa.gov/ozone-pollution/background-ozone-workshop-and-information. The whitepaper does not recommend the use of multi-model means to reduce uncertainty. The Li et al. 2016 citation is an analysis of visibility trends and does not evaluate multi-model results. Moreover, there is no valid theoretical basis to assume that the average of poorly performing models will be more accurate than the best performing individual model for key atmospheric processes. While it is possible that, by chance, the average of several poorly performing models will better match observations, the average may still inaccurately represent the individual processes that contribute to O3 and the sensitivity of O3 to emissions reductions. While it might be true that positive and negative bias errors cancel when averaging multiple model results for seasonal or annual means, this does not necessarily indicate that the multi-model average represents O3 more accurately for episodic events that are of interesting to the air quality planning community. A better approach would be to evaluate and compare models at the process level and for high O3 episodes, and then select the best performing individual model.

Lines 123-125: Note that in certain VOC/NOx chemical regimes the model response to NOx emissions can be strongly non-linear for smaller NOx changes, so the statement that 20% emissions were selected to be "small enough in the assumed near-linear atmospheric chemistry regime" is not consistent with how models respond to NOx emissions and does not provide an explanation for using 20% emissions reductions. A 100% reduction in emissions from a source sector or region is a better approach to evaluate source attribution. Lines 126-130 identify problems with the use of a 20% reduction and this should also be noted in the conclusions. For future work, I recommend 100% reductions when evaluating source contributions.

Line 143: "the necessity of evaluating the extra-regional source impacts on event scale [have] has been emphasized" This is a key point – check to see if this addressed in discussion and conclusions.

Lines 214-216: Biogenic emissions of VOC are larger than anthropogenic VOC globally, and biogenic and geogenic emissions of NOx, SO2, CO and CH4 are also large and can have a substantial impact on model results. It would have been best to harmonize the natural emissions in addition to anthropogenic emissions, and this approach should be used in future work. For this manuscript, the natural emissions used for each model should be summarized and compared, and, if the natural emissions are significantly different between models, the possible effects on model results should be discussed.

Line 254: Equation 2 is confusing because the labels for the scenarios are confusing. It is not clear what RERER(O3,NAM) represents. Does this represent a percent contribution from local versus non-local sources?

Lines 227-240: The description of the model scenarios and the naming convention is complicated and difficult to understand. In line 231, why is "all" enclosed parentheses? Why is a 20% sensitivity simulation described as "*source region*ALL". It is not clear what "ALL" means, and generally, the approach used to label the scenarios is not intuitive.

Line 266-270: Why would lower than normal temperatures in the western U.S. favor decomposition of transported PAN? Lower temperatures would make PAN more stable.

Lines 287-291: The discussion/conclusions should address the uncertainty introduced by using monthly mean emissions.

Lines 293-294: I doubt that the speciation of VOC emissions in 2005 is substantially different compared to 2010, but if the authors' statement that it is "highly unrealistic" to approximate 2010 using 2005 VOC speciation, this seems to be a significant problem for interpreting the model results.

Line 404: Table 2. The model performance evaluation results in Table 2 are not adequate to evaluate the models. In addition to showing the mean bias for multiple models, the model evaluation should also show the bias and error for each model, and the bias

and error for the highest observed O3 days because these are the days that are most relevant for air quality planning.

Lines 430-432: "Except in the northeastern US, the eight-model ensembles show better agreement with the CASTNET O3 observations than the three boundary condition-model ensemble, suggesting that using a larger number of models in the ensemble calculations may result in better overall model performance." Given that the goal of this study is to evaluate the contributions of international emissions to O3 transport in different regions of the world, it is critically important to understand the individual performance of each global model. If there are substantial difference among models in the contributions of stratospheric O3, chemical production of O3 from precursors, or transport and dispersion of O3, the effect of averaging multiple models may be to introduce additional error into the analysis. A better approach is to compare each global model at the process level, and select the best performing models. If it is uncertain which model performs best, source response relationship should be evaluated using simulation with each BC from each of the global models to estimate the uncertainty in the SR relationships.

Lines 461-463: Recommend showing the individual model performance results using each global model BC instead of averaging the results for all three simulations.

Table 4: These results are interesting, but to be policy relevant, we need estimates of the contributions on days that exceed the O3 NAAQS. For example, international transport contributions might be highest on days with good dispersion conditions that do not exceed the NAAQS, and lower for days with stagnant dispersion conditions that are more likely to exceed the NAAQS in urban areas. Alternatively, if might be possible that NAAQS exceedances are more likely to occur in rural areas as a result of international transport because of strong mixing from the troposphere to the surface. It is very difficult to interpret the significance of results that are presented as the mean for all days.

[Figure]

Lines 467-469: This is a key uncertainty that the study does not address. If the modeling systems is biased low for international transport and biased high for local O3 production, the results of the SR analysis may not be reliable.

Lines 476-479: Speciation in SAPRC99 is unlikely to be the cause of model overestimates for O3. SAPRC99 underestimates VOC reactivity in chamber experiments, and the most recent updates to SAPRC are more reactive for urban than SAPRC99. For the rural CASTNet sites in this study, it is more likely that overestimates of biogenic VOC in MEGAN and uncertainty in NOx emissions and fate contribute to the positive bias for O3.

Lines 505-510: Note that larger-than-1 RERER values will be less likely to occur if the model results are analyzed for high O3 days. It is not informative to present results for low O3 days on which NO titration is more likely to occur because these days are not relevant to air quality attainment planning.

Lines 516-519: "Comparing to the HTAP I modeling results, the magnitudes of R(O3, EUR, 20%) are smaller by a factor of 2-3, as a result of the substantial improvement in the European air quality over the past decades" The modeling for HTAP II is for 2010 versus 2001 for HTAP I, so any O3 reductions should reflect emissions reductions for 9 years, not for decades. Have European emissions been reduced by a factor of 2 to 3 from 2001 to 2010, or is it possible that other changes in the HTAP II modeling platform are the cause of this change?

Lines 541-545: This text seems to inappropriately discount the significance of international transport and also the possible importance of differences among the global models. For interstate transport EPA uses 1% of the NAAQS as a significant contribution. Thus, differences among global model much less than 5% of the total model O3 can be very important, especially given that the values discussed in the text are based on a 20% emissions sensitivity and that results are reported as the monthly mean. While local emissions will have a larger contribution, it may not be true that local

emissions control programs alone are the most effective way to attain the NAAQS, as the text seems to suggest.

Lines 562-571: It is surprising that the couple STEM/global model predicts large transport contributions than some global models and smaller transport contributions than other global models. The authors provide a list of factors that contribute to model uncertainty as a possible explanation, but it seems like these uncertainties (e.g., terrain, chemistry) should affect in similar ways each of the coupled STEM/global model simulations. More investigation is needed to explain why STEM sometimes shows higher or lower transport contributions compared to the global model.

Lines 607-609: This is an important finding that should be highlighted in the conclusions and abstract.

Lines 620-622: "Therefore, it is important for more HTAP2 participating models to save their outputs hourly in order to conveniently compute the policy-relevant metrics for the O3 sensitivities." I agree with this statement, and moreover, I do not think you can do a meaningful analysis of any models that do not save the hourly outputs (or 3-hour if that is the finest time resolution used), and I would recommend excluding them from this study.

Lines 612-624 and Figure 9: It is obvious that day time O3 is greater than nighttime O3 at surface sites because O3 deposits to surfaces and is destroyed by chemical reactions at night. So the findings in this text that the maximum daily 8-hour average O3 is greater the 24-hour average O3 is self-evident. I suggest deleting this text. I also recommend focusing the analysis on maximum daily 8 hour averages, especially for the highest O3 days, and not showing results for monthly mean O3.

Line 633: "R(MDA8, EAS, 20%) is smaller during the high O3 total days in all subregions." For GEOS-Chem the contribution appears to be the same on high O3 days compared to all days, and the results are very similar for RAQMS. It would be helpful to show more details for this analysis. Is this the mean O3 for all sites for days in which

any monitor was > 70 ppb, or does it only include data for the monitor that was greater than 70 ppb? I suggest performing a more detailed analysis, e.g., show EAS contribution on each day for a few key sites that frequently have high O3, e.g., Great Basin and Canyonlands sites.

Line 655: "We found that the underestimated free tropospheric O3 from the STEM simulations that used any single free-running chemical boundary conditions contributed to the underestimated STEM surface O3 in the high elevation mountain states." Need to edit and clarify meaning of the above sentence. Was this because the global models underestimated stratospheric O3 or international transport?

---

## Referee Comment (RC2) · Anonymous Referee #2 · 24 Dec 2016

This paper represents a huge task, assembling and comparing the results from the multi-model HTAP2 study. It is a brave undertaking.

However, I do have some concerns about what was learned in the process. I believe the stated goals of the paper are not well met or met in a cursory fashion. There are a number of inferences stated as fact but not in fact proved. In some cases more analysis seems to be needed. In other cases a clearer explication of what has been learned would be helpful. Recommendations about future work are succinctly summarized, but the paper needs to be stronger in detailing what was learned and in justifying the methodology used.

Major Comments: I) The stated paper goals are to address: "1) the differences in O3

sensitivities generated from the HTAP2 and HTAP1 experiments to help address how the LRT impacts on NAM changed through time; 2) how the multi-model approach, as well as the refined model experiment design in HTAP2 can help advance our understanding of the LRT impacts, especially the benefits of increasing the global models' resolutions and involving the regional models; 3) the usefulness of satellite observations for better understanding the sources of uncertainties in the modeled total O3 (e.g., from the emission and regional models' boundary condition inputs) as well as for reducing the uncertainties in some of these model inputs via chemical data assimilation."

As the paper stands it is not clear if it achieved its goals. The answers to these questions should be clearly articulated in the conclusions and in the body of the paper itself. In particular:

1) Between HTAP1 and HTAP2 models have changed, emissions have changed and the transport has changed. So it is not really clear how the sensitivity changed through time. The authors suggest many of the changes are due to the changes in emissions, but this remains to be proven. The authors could determine if the changes in the sensitivities are consistent with the change in emissions by using the HTAP1 emissions and the current sensitivities (delta O3/delta emissions) to determine if most of the changes from HTAP1 are consistent with emission changes. However, as it stands the first goal of the paper cannot be met without substantially more analysis.

2) It is not clear how this study enhanced our understanding of LRT nor is it very clear how changes in model resolution impacted the solutions. The STEM model resolution is 60x60 km, actually rather comparable to a global model of 1o resolution (about 85 km at 40o N). While there is a wide range of different resolutions in the global models it is unclear how this paper really explored the impact of resolution on the results. What aspects of LRT did the paper enhance? This should be clear in the paper.

3) The usefulness of satellite data is essentially a "motherhood" statement. It is somewhat unclear how this paper further showed this usefulness. This is especially true

since the case study using satellite data was presented in a rather cursory manner.

II) It is not clear what the goal of using the STEM model is here. As pointed out above the resolution is not that much higher than some of the global models that give the boundary conditions. Differences between the STEM results and the boundary condition model could be due to the different chemistry in the two models or due to the differences in transport. Driving the models with different meteorological datasets also risks an inconsistency in the boundary conditions (e.g., chemical plumes transported in the jet in the parent model might be mismatched with the jet in STEM). At any rate the rationale for the use of the STEM model should be clearly articulated. What did we learn by coupling the global models with the STEM model?

III) The case study is rather thin. What are the goals of this section? This section should either be expanded or dropped.

Specific Comments: 1. L42. The sentence beginning is rather awkward. Consider rewording.

2. L48-49, "This indicates…..". This has to be proven. As is well known interannual variability of the atmosphere is substantial.

3. L175. Starting here the manuscript goes into considerable detail about how the simulatons are set up. This does not work well in the introduction, but belongs in the methodology section.

4. L202, Section 2.1. The manuscript parses the emissions between East Asia, MICS Asian regions and south Asian countries. The domains of each these regions is not clear.

5. Table 1. All abbreviations should be defined. Also the table headings need to be reformatted.

6. L250-253. This notation is should be improved: the left hand side of the equation has a percentage sign, but not the right. I would suggest something like EASALL(-

20%) on the right to distinguish this from the R(O3, EAS, 100%) where presumably all EAS emissions are reduced by 100%.

7. L290 and following paragraph. A long discussion is presented concerning STEM lightning emissions, biogenic emissions and VOC speciation. How were these emissions parameterized in the other models, the same as STEM or differently? Please specify more thoroughly differences in emissions between STEM and other models.

8. L394. "less sensitive" – less sensitive to what?

9. L420. "de-stripped" – the meaning is unclear.

10. L459. "suggesting that using". This seems rather speculative. There are many possible explanations.

11. L471-472. "overall there does appear to be a positive bias". This seems to be a rather strong statement considering the previous sentence. It would be better to say satellite is consistent with a positive bias.

12. L478-479. Can you provide a reference why co-emitted species are likely to be biased in the same way as NOx. It is not at all clear to me that emission factors would be all biased in one direction.

13. L509-510. "mainly due to". Maybe. It would be better to say consistent with.

14. L556-557. Did you show this? Probably better to say "consistent with".

15. L567-568. This is an interesting result: that R in HTAP2 is larger than in HTAP1. However, the reasons for this have not been clearly shown. Certainly the difference is consistent with emission trends but the authors need to establish that this is the case (see general comments above)

16. Figure 9. I think this is a scatter plot of R(MDA8,EAS,20%) and R(O3, EAS, 20%). Please address the notation.

17. The point of section 3.3 is not clear. Some of the figures panels in this section seem to be referred to in a very cursory manner or not at all (e.g., Figure 11). This section needs to be much better developed or not presented.

18. Figure 7 caption. I assume (a), (b), and (c) refer to the first three rows. Better to say row 1, row2 and row 3 or label all panels with letters.

---

## Referee Comment (RC3) · Anonymous Referee #3 · 3 Jan 2017

General Comments

This manuscript presents the first HTAP Phase II findings, expanding on HTAP Phase I by incorporating regional models to estimate the impact of international anthropogenic emissions on U.S. surface ozone. The authors use boundary conditions from three different global models to drive the regional STEM model, and compare the sensitivities of surface ozone in North America to international anthropogenic emissions with those determined from 8 global models. They further compare with an adjoint version of one model, use boundary conditions from a model that assimilated satellite ozone products, and conduct a case study using multiple satellite and ground-based products. This is a major undertaking, as noted by another reviewer. I agree with the other reviewers,

however, that the paper suffers from some shortcomings. Several of the issues I was planning to cover were discussed at length in the earlier reviews, so I focus below on additional points.

I'd like to see the abstract/conclusions clarify and quantify (e.g., within 10%, 30%, factors of 2?) the conclusions regarding how different the global and regional model estimates are, and how much the RER sensitivity estimates have changed from those reported in the 2010 HTAP report. I agree with Dr. Tonnesen that more emphasis on episodic events would enhance the policy relevance of this work.

Throughout the text, more quantitative and specific language should be used wherever possible, and the paper should be edited carefully for clarity (e.g., incomplete sentence L768). The introduction is quite long and could state earlier on what the point of this study is to provide a context before going into all the details of past work.

Specific Comments L42-45. Elaborate on what this means for drawing conclusions regarding the role of hemispheric transport of air pollution.

L48 'Tagged tracer approach' is mentioned here and elsewhere (e.g. L564); a brief explanation is needed as approaches can involve tagging ozone itself or tagging precursors. I'm not convinced that this study cleanly isolated the role of rising East Asian anthropogenic emissions; see also RC2 comments.

L51 Are the adjoint sensitivities compared to all the global models or just the forward version of GEOS-Chem? Is this the same version as used to provide boundary conditions? (see also L591)

L54-56 Try to quantify this statement: is it off by 20%? Factor of 2?

L57-59 This appears to be a general statement rather than a conclusion drawn from this work and thus does not seem appropriate to include in the abstract.

L96 The first paper to show this was Jacob et al., GRL, 1999: http://onlinelibrary.wiley.com/doi/10.1029/1999GL900450/abstract

L148. Region-dependent, but also time-dependent?

L220-227 Seems relevant to provide BVOC emissions over Asia and North America. How much do North American anthropogenic emissions contribute to global totals?

L233 References could be included in Table 1

L238 Why are boxes shown in Figure 1 if the regions are actually following the political boundaries as indicated in L258?

L276. Given that Lin et al. 2012 estimated Asian ozone pollution transport to the western U.S. using a global model about this resolution, a case needs to be made for why it's appropriate to use a regional model (e.g., allows testing of multiple boundary conditions, and regulatory applications would presumably run at finer scales).

L283. This may be true for the Asian pollution transport, but Lin et al. 2015 indicate that 2010 isn't a particularly high year for stratospheric intrusions reaching surface air over the WUS. http://www.nature.com/articles/ncomms8105 See their figure 2.

L287-88 Is this just reflecting the warming trend over the 81-00 period? Temperatures and ozone production were even higher in 2011 and 2012 in the eastern US.

L315 How was this downscaling done?

L442 perhaps needs a reference for the HTAP1 work unless this was done as part of this study?

L445-447. It's not clear what the take-away point is here. Are the models underestimating Asian pollution influence or can we not tell because it could be regional transport? Presumably even though the data assimilation fixes this problem, it does not help us to distinguish between these possible sources of error?

L472. How did this study determine that the bias is likely due to overestimated anthropogenic NOx emissions? May doesn't look like it has a clear bias whereas July does. How do we know this is associated with anthropogenic sources rather than seasonally varying sources like soil NOx for example? Are there seasonal variations in the anthropogenic NOx emissions?

L481-483. Doesn't this interpretation depend on where the photochemical regime is at in terms of ozone production with respect to NOx emissions?

L520 An estimate of how large these biases are and how much bias they introduce into ozone would be useful here.

L541-544. There seems to be model disagreement near the Canadian border, with Oslo for example suggesting high cross-border influence but CHASER suggesting much less.

L544-547. Is Oslo also higher resolution as it looks similar to EMEP in terms of higher influence.

L585-588 Where is this shown?

L612-616. Can you provide estimates of how the ozone lifetimes in the boundary layer differ in the different simulations?

L627-628. There seems to be an assumption that LRT is obvious from satellite data. This isn't the case for ozone. How will LRT be convincingly separated out from other ozone sources?

L658-659. Did all models capture the same events in terms of their timing and approximate regional location?

L661. It would be more convincing to show this as a monthly mean diurnal cycle rather than rely on Figure 2a.

L679-681. I didn't follow this point.

L741-742. Be more specific here.

L744-747. Is there a relationship between the bias and the Asian transport events?

L747-749. It's not clear how better quantifying stratospheric o3 intrusion helps reduce North American pollution levels and model uncertainties. This statement also implies that stratospheric intrusions are as important as local ozone formation.

L750. How frequent are these episodic sensitivities to East Asian emissions? Are they occurring when measured ozone is highest?

L800-801. These suggestions seem to neglect the important caveat that these approaches assume that model transport is perfect.

---

## Author Comment (AC1) · 4 Jan 2017

We thank the careful reviews by Dr. Tonnesen and two Anonymous Referees, as well as the suggestions on additional references by Dr. Lin.

We are glad that all three reviewers recognized highlights of the study, and are interested in seeing additional analyses related to the questions we address for HTAP2. The revised paper will articulate the purpose of the regional modeling, include more details on model setup and evaluation, and introduce a summer event. These will address the reviewers' major concerns.

Some of the reviews published online referred to the version we initially submitted to

[Figure]

ACPD in Oct 2016. Although not required at the technical correction stage for ACPD, they had been addressed before the paper was published online. In the point-by-point response to the reviews that we will be submitting during the revision phase, the corresponding changes made to address these reviews will be clearly marked.
* * *

---

## Author Comment (AC2) · 1 Mar 2017

**Response to Reviewer #1 (Dr. Tonnesen)'s comments**

We thank the careful review by Dr. Tonnesen. Please see below our response (in blue) to her general and specific comments (in black). As a majority of her comments were also received during the ACPD reviewing phase, some changes have been made to the original manuscript to address a number of these comments. The revised ACPD manuscript with tracked changes (submitted in late Nov 2016 to ACP together with a clean version) shows these changes more clearly.

General comments

Most of the paper is focused on comparison of monthly mean model results with very limited evaluation of model performance and no analysis of the causes of the differences among the global model simulations. This analysis is not substantially different from previous HTAP studies and is not informative. I suggest moving most of the text and plots that discuss monthly mean model results to the supplement, and instead, the authors should evaluate and compare model performance on several short term episodes that are more relevant to ozone transport and air quality planning.

The EPA White Paper that you suggested in the later comments summarizes some analyses falling into two categories: 1) the monthly, seasonal, or annual mean analyses that provide a broad characterization perspective. Many published and ongoing analyses with focus on conditions in the past decades are done on such large scales, including HTAP1 and some of the HTAP2 analysis; and 2) those focusing on specific polluted events, which are more important to US air quality management. It is mentioned in the White Paper that as long as the averaging time of the results is clarified, both kinds of analysis would be considered.

A uniqueness of this paper is that it includes analyses on both large and small scales, as now highlighted in the abstract: "In addition to the analyses on large spatial/temporal scales relative to the HTAP1, we also show results on subcontinental- and event-scale that are more relevant to the US air quality management." To meet the objective of this study of connecting results from the past studies, including the HTAP1 and other HTAP2 works, we performed analyses using the multi-model mean approach over large spatial and temporal scales. Moreover, model evaluation over some non-NAM regions would be only possible on a monthly basis (e.g., over East Asia) using the available sparse/infrequent in-situ measurements there. More detailed model evaluation has been added to the paper.

To be more relevant to the US air quality management, we also conducted event-based analysis over the US in May and June 2010, and reported model performance and modeled SR relationships on polluted sites/days. A June event was newly added per your following suggestion. The weight of the $O_3$ exceedance based analyses in the revised paper significantly increased. See Figures 10-18 and related text.

The most interesting aspect of the paper is the section that addresses the May 9, 2010 O3 episode. I suggest including a more detailed discussion of this event, including an assessment of the relative contributions of stratospheric O3 and international transport of O3 for this event. Given that all of the global models performed poorly for this event (with the exception of RAQMS with data assimilation), a key finding could be that currently available global models do not perform well

for some high ozone events. I also suggest performing additional analysis for at least one other high O3 event during summer 2010 to contrast with the May 9 event. By performing a more detailed evaluation and comparison of the different global models (and the couple STEM/Global model simulations) for specific episodes, the authors can more directly evaluate model performance and the suitability of the individual global models for use as boundary condition data in higher resolution regional models.

Significant changes to the paper have been made to address this:

- We added a summer event (~10 June, 2010) leading to similar conclusions to the existing 9 May case study. See Figures 16-18 and related text.

- Model performance and modeled SR relationships on polluted sites/days are now reported (Figures 10, 11, 12 for May-June 2010; Figures 14/15a-d and 17/18a-d panels for two exceptional events) and the conditions for spring and summer times are compared.

- The impacts of stratospheric O$_3$ intrusion reported by Lin et al. (2012a, b) for these two events were added to Section 3.3 (i.e., ~1/3 and ~50% of the total at where exceedences occurred based on their model estimates).

We extended the event-based analysis and discussions to highlight the findings from these case studies, for example, as you said, that all of the global models performed poorly for some high O$_3$ events (with the exception of RAQMS with data assimilation). We believe such uncertainty in the chemical boundary conditions poses difficulties for regional models (regardless of their resolutions and other configurations, parameterizations) to accurately simulate the total O$_3$ and estimate the SR relationships using boundary conditions downscaled from these global models.

I suggest deleting the text that asserts that the use of an ensemble of global models is a preferred approach. The citation (U.S.EPA 2016) is summary of comments at a public meeting and should not be used as citation because the comments were not peer reviewed and do not reflect the consensus of the meeting participants. A better citation would be the EPA whitepaper on background ozone which was reviewed within EPA and is available at https://www.epa.gov/ozone-pollution/background-ozone-workshopand-information. The whitepaper does not recommend the use of multi-model means to reduce uncertainty. The Li et al. 2016 citation is an analysis of visibility trends and does not evaluate multi-model results. Moreover, there is no valid theoretical basis to assume that the average of poorly performing models will be more accurate than the best performing individual model for key atmospheric processes. While it is possible that, by chance, the average of several poorly performing models will better match observations, the average may still inaccurately represent the individual processes that contribute to O3 and the sensitivity of O3 to emissions reductions. While it might be true that positive and negative bias errors cancel when averaging multiple model results for monthly or seasonal means, this does not necessarily indicate that the multi-model average represents O3 more accurately for episodic events that are of interest to the air quality planning community. A better approach would be to evaluate and compare models at the process level and specifically for high O3 episodes, and then select the best performing individual model.

The EPA White Paper is now cited as "US EPA, 2016a" in the introduction. The Li et al. (2016) citation was removed. Thanks for the comments on the use of multi-model approach, as well as the suggestions on evaluating/comparing the models on process level. A sentence has been added to introduce the multi-model approach: "'Ensemble' model analyses have been suggested by some US stakeholders as one of the methods for helping with the characterization of the background O$_3$ components (US EPA, 2016b)." The multi-model approach in this paper was mainly used to

connect the findings in HTAP1. We now show individual model's performance in Table 1, Figure 11, 15a-d, 18a-d, and the event-based analysis has been extended in which individual model's performance was shown. The language in the discussions related to the multi-model mean results has been modified. For example, over the US, "This reflects that averaging the results from a larger number of models in this case more effectively cancelled out the positive or negative biases from the individual models.", but for the East Asia, "Unlike at the CASTNET sites, the three-model ensemble agrees better with the observations than the eight-model ensemble". We listed in this study possible sources of uncertainty for some model and pointed out "Future work should emphasize on evaluating and comparing all models on process level to better understand their performance", which would be good materials for follow-on papers.

Detailed comments

Line 63: Opening sentence is awkward. There is no clear link of the uneven distribution to the health/ecosystem impacts of O3. Also, the uneven distribution of O3 is mostly caused by strong concentration gradients in precursor emissions, but this sentence only identifies the O3 lifetime as a cause of the distribution. Suggest rewriting with a focus on the high mixing ratios, not just the distribution.
The opening sentence was rewritten.

Line 73: "to control the emissions of its precursors from these various sources". Not clear what "these various sources" refers to here. The previous sentence identified the stratosphere and local to distant emissions sources, so presumably this sentence is suggesting that there will be benefits of control of both local and international emissions sources, but this sentence then goes on to list precursors categories (VOC, NOx, CO) without reference to local vs international or biogenic vs. anthropogenic. I can infer what the authors mean, but the introductory paragraphs are awkwardly written and potentially confusing to a reader who is not an expert.
This sentence was rewritten.

Line 75: Also include methane in this list
The original "VOCs" has been split to methane and non-methane VOCs in this sentence.

Line 80: background and baseline are not the same. See Cooper et al. for their definition of baseline, and EPA white paper (link below) for definition of U.S. background ozone. Briefly, baseline O3 (as defined by Cooper et al.) can include contributions from upwind U.S. anthropogenic precursor emissions while U.S. background ozone excludes all U.S. anthropogenic emissions. Line 82: Given how the authors defined baseline/background O3, this statement is problematic: "below which the air quality standard is not recommended to be set". Baseline O3 can be elevated in some areas because of transport of anthropogenic precursors and O3 from upwind U.S. states. It is appropriate to set the NAAQS below the baseline O3 level in these areas because the elevated baseline O3 is being addressed by emissions reductions in upwind states. I recommend breaking this very long sentence into several sentences that describe each of the points identified more clearly and more accurately.
This part has been modified and now reads as: "Issues regarding making accurate estimates of the total $O_3$ as well as the background $O_3$ level (defined as the concentration that is not affected by recent locally-emitted or produced anthropogenic pollution) (e.g., McDonald-Buller et al., 2011;

Zhang et al., 2011; Fiore et al., 2014; Huang et al., 2015), have been recently discussed as part of the implementation of the new US $O_3$ standard (US EPA, 2016a, b)."

Line 90: "It has been revealed" is awkward – "revealed" has other connotations. Suggest "It has been found".
Done.

Line 95: "A better understanding of the processes that determine the O3 distributions" Note that the authors have not yet clearly and comprehensively described the processes. They should describe the roles of stratospheric (both routine contributions and discrete intrusion events), biogenic precursors, wildfires, and anthropogenic precursors. We are especially concerned with conditions in which the mixing ratio exceeds the NAAQS, so it is not only the distribution but also the mixing ratio that is important.
Changed "$O_3$ distributions" to "$O_3$ pollution levels…". While those multiple sources contribute to the total $O_3$ and its exceedances, the component this study mainly focuses on is the LRT of non-NAM anthropogenic pollution, particularly those from the East Asia.

Line 96: delete "for recent years". This will be useful for all past years and for future predictions.
Done.

Lines 110-112: "Large intermodel diversity was found in the simulated total O3 and the intercontinentally transported pollution for the chosen SR pairs in the northern midlatitudes, indicating the challenges with simulations by any individual model to accurately represent the key atmospheric processes." The conclusion that no individual model performs well is not supported by a finding of inter-model diversity. For example, it is possible that one model performs well while other models do not. The authors need to cite results of the individual model performance evaluations to support the statement that no model perform well.
We changed "any individual model" to "model simulations". Now the global models, particularly the three boundary condition models, are evaluated individually in places. Model evaluation at the receptor side (western US) is performed against both the surface in-situ observations and satellite vertical profiles; Evaluation at a focused source region (East Asia) has been added. The model comparison with OMI column data provides the uncertainty introduced by the bottom-up emission inventory. These all help better understand the different models' performance.

Lines 113-116: The citation (U.S.EPA 2016) is summary of comments at a public meeting and should not be used as citation because the comments were not peer reviewed and do not reflect the consensus of the meeting participants. A better citation would be the EPA whitepaper on background ozone which did receive review within EPA and is available at https://www.epa.gov/ozone-pollution/background-ozone-workshop-andinformation. The whitepaper does not recommend the use of multi-model means to reduce uncertainty. The Li et al. 2016 citation is an analysis of visibility trends and does not evaluate multi-model results. Moreover, there is no valid theoretical basis to assume that the average of poorly performing models will be more accurate than the best performing individual model for key atmospheric processes. While it is possible that, by chance, the average of several poorly performing models will better match observations, the average may still inaccurately represent the individual processes that contribute to O3 and the sensitivity of O3 to emissions reductions. While it might be true that positive and

negative bias errors cancel when averaging multiple model results for seasonal or annual means, this does not necessarily indicate that the multi-model average represents O3 more accurately for episodic events that are of interesting to the air quality planning community. A better approach would be to evaluate and compare models at the process level and for high O3 episodes, and then select the best performing individual model.

Same as our response to the last general comment: The EPA White Paper is now cited as "US EPA, 2016a" in the introduction. The Li et al. (2016) citation was removed. Thanks for the comments on the use of multi-model approach, as well as the suggestions on evaluating/comparing the models on process level. A sentence has been added to introduce the multi-model approach: "'Ensemble' model analyses have been suggested by some US stakeholders as one of the methods for helping with the characterization of the background O$_3$ components (US EPA, 2016b)." The multi-model approach in this paper was mainly used to connect the findings in HTAP1. We now show individual model's performance in Table 1, Figure 11, 15a-d, 18a-d, and the event-based analysis has been extended in which individual model's performance was shown. The language in the discussions related to the multi-model mean results has been modified. For example, over the US, "This reflects that averaging the results from a larger number of models in this case more effectively cancelled out the positive or negative biases from the individual models.", but for the East Asia, "Unlike at the CASTNET sites, the three-model ensemble agrees better with the observations than the eight-model ensemble". We listed in this study possible sources of uncertainty for some model and pointed out "Future work should emphasize on evaluating and comparing all models on process level to better understand their performance", which would be good materials for follow-on papers.

Lines 123-125: Note that in certain VOC/NOx chemical regimes the model response to NOx emissions can be strongly non-linear for smaller NOx changes, so the statement that 20% emissions were selected to be "small enough in the assumed near-linear atmospheric chemistry regime" is not consistent with how models respond to NOx emissions and does not provide an explanation for using 20% emissions reductions. A 100% reduction in emissions from a source sector or region is a better approach to evaluate source attribution. Lines 126-130 identify problems with the use of a 20% reduction and this should also be noted in the conclusions. For future work, I recommend 100% reductions when evaluating source contributions.

We chose a 20% reduction to be consistent with HTAP1 and HTAP2's experiment design. We cited papers here and also in places in Section 3 comparing the sensitivities in response to different sizes of perturbations and the suitability of each choice for address different questions. We also included a couple of sentences in the conclusion related to the scalability: "…The underestimation in other seasons of the HTAP2 study period may be higher and will need to be quantified in future work. Motivated by Lapina et al. (2014), additional calculations will be conducted in future to explore the scalability of different O$_3$ metrics in these cases. For future source attribution analysis, in general it is recommended to directly choose the suitable size of the emission perturbation based on the specific questions to address, and to avoid linearly scaling O$_3$ sensitivities that are based on other amounts of the perturbations."

Line 143: "the necessity of evaluating the extra-regional source impacts on event scale [have] has been emphasized" This is a key point – check to see if this addressed in discussion and conclusions. We have added a summer event case study for comparison. Model performance and modeled SR relationships on polluted sites/days are now reported.

Lines 214-216: Biogenic emissions of VOC are larger than anthropogenic VOC globally, and biogenic and geogenic emissions of NOx, SO2, CO and CH4 are also large and can have a substantial impact on model results. It would have been best to harmonize the natural emissions in addition to anthropogenic emissions, and this approach should be used in future work. For this manuscript, the natural emissions used for each model should be summarized and compared, and, if the natural emissions are significantly different between models, the possible effects on model results should be discussed.

The non-anthropogenic emissions do differ by models, which impact the background $O_3$ estimation, but these have only been compared in detail between GEOS-Chem and STEM. What's shown in the following plots (also included in the paper discussion and SI) are June 2010 comparisons for soil, lightning, biomass burning and NEI05 anthropogenic $NO_x$ emissions (in molec./cm$^2$/s) used for the Lapina study, and the numbers at lower-left corners indicate the domain integrated amounts (note that GEOS-Chem emissions were plotted/integrated over a slightly larger domain). The same set of non-anthropogenic emissions was used for our HTAP2 simulations.

GEOS-Chem:

STEM:

[Figure]

Comparing this study's GEOS-Chem emissions with previous studies on summer 2005 (Choi et al., 2009: Soil: 0.05 Tg, lightning: 0.19 Tg; biomass burning: 0.005 Tg; anthropogenic: 0.46 Tg), it seems that non-anthropogenic emissions contributed more to the total CONUS $NO_x$ emissions in June 2010.

GEOS-Chem, C-IFS and WRF/STEM BVOC emissions were all calculated using MEGAN, but the meteorological inputs for their calculations are different (listed in Table 1c), which could lead to notable differences. Wolfe et al. (2015) showed that GEOS-Chem isoprene emissions are ~40% higher than aircraft flux observations in some US regions, and a detail quantification of WRF/MEGAN's biases is included in Huang et al. (2017).

We agree and suggest that for future activities the non-anthropogenic emissions should be formally reported for all models by region, sector, and species. In this section, we now added: "Non-anthropogenic emission inputs used in different models' simulations may differ, and their impacts on the modeled total $O_3$ and the SR relationships will be compared in detail in future studies." And for STEM and its boundary condition models, we added: "Note that non-anthropogenic emission inputs used in STEM and its boundary condition models differed, as summarized in Table 1c. Figure S1 shows detailed comparisons between STEM and GEOS-Chem's non-anthropogenic (i.e., soil, lightning, biomass burning) $NO_x$ emission inputs, and their impacts on the modeled NAM background $O_3$ were included in Lapina et al. (2014). Such quantitative comparisons will also be carried out between STEM and its other boundary condition models in future studies."

Line 254: Equation 2 is confusing because the labels for the scenarios are confusing. It is not clear what RERER(O3,NAM) represents. Does this represent a percent contribution from local versus non-local sources?
Equation 2 has been rewritten. For further explanation, a sentence was added following the equation: "The denominator and numerator terms of RERER represent the impacts of global and non-NAM anthropogenic emissions on NAM $O_3$, respectively."

Lines 227-240: The description of the model scenarios and the naming convention is complicated and difficult to understand. In line 231, why is "all" enclosed parentheses? Why is a 20% sensitivity simulation described as "*source region*ALL". It is not clear what "ALL" means, and generally, the approach used to label the scenarios is not intuitive.
A sentence has been added: "where "ALL" refers to "all species and sectors", consistent with HTAP1 and HTAP2's naming convention."

Line 266-270: Why would lower than normal temperatures in the western U.S. favor decomposition of transported PAN? Lower temperatures would make PAN more stable.
The sentence now reads as: "The mean near-surface air temperatures in the western US in this spring were lower than the climatology, with larger anomalies in the mountain states, which may have led to weaker local $O_3$ production and decomposition of the transported peroxyacyl nitrates (PAN)."

Lines 287-291: The discussion/conclusions should address the uncertainty introduced by using monthly mean emissions.

Following this sentence, we added a sentence: "This change can introduce uncertainty for some US regions where weekday-weekend variability of some $O_3$ precursors' emissions was notable during the studied period (e.g., weekend $NO_x$ emissions in southern California during spring/summer 2010 were 0.6-0.7 of the weekday emissions as reported by Kim et al. (2016) and Brioude et al. (2013)), but this was done to ensure consistency with the HTAP2 global model simulations, that also didn't use daily variable emissions for any regions in the world." In Section 3.1.1, we added another sentence: "Also, the use of monthly-mean anthropogenic emissions as well as the overall rough treatment of emission height and temporal profiles can be sources of uncertainty." In conclusion, we now have: "..efforts should also be placed to have the models timely update the height and temporal profiles of the emissions from various sectors". This includes both diurnal and weekly cycles.

Lines 293-294: I doubt that the speciation of VOC emissions in 2005 is substantially different compared to 2010, but if the authors' statement that it is "highly unrealistic" to approximate 2010 using 2005 VOC speciation, this seems to be a significant problem for interpreting the model results.
We agree that the uncertainty of VOC speciation may be high for its base year of 2005 as well. This sentence has been changed to: "The VOC speciation based on the year of 2005 can be unrealistic for 2005 as well as 2010…".

Line 404: Table 2. The model performance evaluation results in Table 2 are not adequate to evaluate the models. In addition to showing the mean bias for multiple models, the model evaluation should also show the bias and error for each model, and the bias and error for the highest observed O3 days because these are the days that are most relevant for air quality planning.
We have done extra work to evaluate the boundary condition model in greater detail. Model performance and modeled SR relationships on polluted sites/days are now reported for the entire study period and during two case studies. In addition to the evaluation over the US, we added the evaluation over the East Asia with the EANET surface observations.

Lines 430-432: "Except in the northeastern US, the eight-model ensembles show better agreement with the CASTNET O3 observations than the three boundary condition model ensemble, suggesting that using a larger number of models in the ensemble calculations may result in better overall model performance." Given that the goal of this study is to evaluate the contributions of international emissions to O3 transport in different regions of the world, it is critically important to understand the individual performance of each global model. If there are substantial difference among models in the contributions of stratospheric O3, chemical production of O3 from precursors, or transport and dispersion of O3, the effect of averaging multiple models may be to introduce additional error into the analysis. A better approach is to compare each global model at the process level, and select the best performing models. If it is uncertain which model performs best, source response relationship should be evaluated using simulation with each BC from each of the global models to estimate the uncertainty in the SR relationships.
This part has been modified to: "As reported in the literature (e.g., Geddes et al., 2016; Travis et al., 2016), the representation of land use/land cover, boundary layer mixing and chemistry can be sources of uncertainty for certain global model (i.e., GEOS-Chem), but how serious these issues were in the other models need to be investigated further. Some other possible reasons include the variation of these models' non-anthropogenic emission inputs and chemical mechanisms (Table

1c). Future work should emphasize on evaluating and comparing all models on process level to better understand their performance. Except in the northeastern US, the eight-model ensembles show better agreement with the CASTNET O$_3$ observations than the three boundary condition-model ensemble. Overall the three-model ensemble only outperforms one model but the eight-model ensemble outperforms seven. This reflects that averaging the results from a larger number of models in this case more effectively cancelled out the positive or negative biases from the individual models." We now evaluate each of the global models individually, with strong focus on the three boundary condition models.

Lines 461-463: Recommend showing the individual model performance results using each global model BC instead of averaging the results for all three simulations.
Done.

Table 4: These results are interesting, but to be policy relevant, we need estimates of the contributions on days that exceed the O3 NAAQS. For example, international transport contributions might be highest on days with good dispersion conditions that do not exceed the NAAQS, and lower for days with stagnant dispersion conditions that are more likely to exceed the NAAQS in urban areas. Alternatively, if might be possible that NAAQS exceedances are more likely to occur in rural areas as a result of international transport because of strong mixing from the troposphere to the surface. It is very difficult to interpret the significance of results that are presented as the mean for all days.
As mentioned previously, the monthly-based analyses in this paper were mainly used to connect the findings in HTAP1, and the analysis focusing on polluted sites/days has been extended.

Lines 467-469: This is a key uncertainty that the study does not address. If the modeling systems is biased low for international transport and biased high for local O3 production, the results of the SR analysis may not be reliable.
We point out the uncertainty from the free running HTAP2 simulations in these sentences, but do also suggest the methods to reduce this uncertainty. In the following sentences: "Switching the STEM chemical boundary conditions to the assimilated RAQMS base simulation led to increases in the simulated surface O$_3$ concentrations by >9 ppbv in the western US (Figure S2, right), associated with higher positive biases (due to several factors discussed in the next paragraph). Regional-scale assimilation could further reduce uncertainties introduced from regional meteorological and emission inputs to obtain better modeled total O$_3$ and the partitioning of trans-boundary versus US contributions (e.g., Huang et al., 2015)." Additionally, in the last paragraph of this paper, we proposed the possible approaches to improve source attribution estimates by incorporating observations.

Also, the quality of the model boundary conditions only indicates how well the total "transported background" component is represented, and can not be directly connected with the accuracy of the model estimated LRT pollutants. This is emphasized in Section 3.3.

Lines 476-479: Speciation in SAPRC99 is unlikely to be the cause of model overestimates for O3. SAPRC99 underestimates VOC reactivity in chamber experiments, and the most recent updates to SAPRC are more reactive for urban than SAPRC99. For the rural CASTNet sites in this study, it

is more likely that overestimates of biogenic VOC in MEGAN and uncertainty in NOx emissions and fate contribute to the positive bias for O3.

This study does not cover any investigation on more recent updates in SAPRC, like SAPRC 07 or 11. In the text that we describe the amplified biases in STEM compared with the global models, here we just list some references that showed SAPRC99 produced much higher $O_3$ than other mechanisms which were used by certain HTAP2 global models. CASTNET $O_3$ is subject to the US urban $O_3$ pollution due to the regional-scale transport (See Huang et al., 2013b). In addition to the uncertainty from $NO_x$ and BVOC emissions, we now extended the discussions to address a comment by Reviewer #3's: "Huang et al. (2017) showed that MEGAN's positive biases are in part due to the positively-biased temperature and radiation in WRF, and reducing ~2°C in WRF's temperature biases using a different land initialization approach led to ~20% decreases in MEGAN's isoprene emission estimates in September 2013 over some southeastern US regions…Quantifying the impacts of overestimated biogenic emissions and the biased weather fields that contributed to the biases in emissions on the modeled $O_3$ is still an ongoing work."

We also cited other regional model studies that attributed modeled biases to chemical mechanisms and emission biases: "Some existing studies also reported $O_3$ and $NO_2$ biases from other regional models in the eastern US, due to the chemical mechanism and biases in $NO_x$ and biogenic VOC emissions (e.g., Canty et al., 2015)." And pointed out the need for future investigation on these from the AQMEII in the following sentence.

Lines 505-510: Note that larger-than-1 RERER values will be less likely to occur if the model results are analyzed for high O3 days. It is not informative to present results for low O3 days on which NO titration is more likely to occur because these days are not relevant to air quality attainment planning.

Again the monthly mean based analysis are shown to connect with the HTAP1 and other HTAP 2 modeling results (e.g., Surendran et al., AE, 2016 showed seasonal RERERs for HTAP2 in SAS) that are done on a monthly or seasonal basis. Although not directly relevant to the air quality management, these provide a broad characterization perspective (also considered in the 2016 EPA White Paper). We included a separate section (3.3) in the paper showing event based analysis.

Lines 516-519: "Comparing to the HTAP I modeling results, the magnitudes of R(O3, EUR, 20%) are smaller by a factor of 2-3, as a result of the substantial improvement in the European air quality over the past decades" The modeling for HTAP II is for 2010 versus 2001 for HTAP I, so any O3 reductions should reflect emissions reductions for 9 years, not for decades. Have European emissions been reduced by a factor of 2 to 3 from 2001 to 2010, or is it possible that other changes in the HTAP II modeling platform are the cause of this change?

In Section 2, we now have descriptions of "North Africa that are included in HTAP1's EUR domain. The impact of emissions over these regions on comparing the NAM R(O3, EUR, 20%) values in HTAP1 and HTAP2 will be discussed in Section 3.2.1." And in the results section, following this commented sentence, we added: "… and also possibly due to the changes in the HTAP2 experiment setup from HTAP1 (e.g., EUR by HTAP1's definition includes regions in Russia/Belarussia/Ukraine, Middle East and North Africa that are excluded from the HTAP2's EUR domain)." We also believe this difference is in part due to the different HTAP1 and HTAP2 participating models and their configurations.

Lines 541-545: This text seems to inappropriately discount the significance of international transport and also the possible importance of differences among the global models. For interstate transport EPA uses 1% of the NAAQS as a significant contribution. Thus, differences among global model much less than 5% of the total model O3 can be very important, especially given that the values discussed in the text are based on a 20% emissions sensitivity and that results are reported as the monthly mean. While local emissions will have a larger contribution, it may not be true that local emissions control programs alone are the most effective way to attain the NAAQS, as the text seems to suggest.

The text in this paragraph has been modified to: "These monthly- and regional-mean R(O$_3$, EAS, 20%) values suggest that despite dilution along the great transport distance, the EAS anthropogenic sources still had distinguishable impact on the NAM surface O$_3$…. Also, similar to the findings from the HTAP1 studies, the large intermodel variability (as indicated in Table 4) in the estimates of intercontinental SR relationships indicates the uncertainties of these models in representing the key atmospheric processes which needs more investigations in the future. Overall, R(O$_3$, EAS, 20%) and its intermodel differences are much smaller than the biases of the modeled total O$_3$ in NAM. Other factors can contribute more significantly to the biases in the modeled total O$_3$, such as the stratospheric O$_3$ intrusion and the local O$_3$ formation, and assessing the impacts from these factors would be also helpful for understanding the uncertainties in the modeled O$_3$." Related sentences in Section 4 and the abstract were also revised.

Lines 562-571: It is surprising that the couple STEM/global model predicts large transport contributions than some global models and smaller transport contributions than other global models. The authors provide a list of factors that contribute to model uncertainty as a possible explanation, but it seems like these uncertainties (e.g., terrain, chemistry) should affect in similar ways each of the coupled STEM/global model simulations. More investigation is needed to explain why STEM sometimes shows higher or lower transport contributions compared to the global model.

This part now has been extended. The differences between regional and global models' results are due to the different terrain, met fields, transport and chemical production/loss between STEM and its boundary condition model. The differences of STEM/GC, STEM/CIFS and STEM/RAQMS pairs are different.

Lines 607-609: This is an important finding that should be highlighted in the conclusions and abstract.

This is now emphasized in both the abstract and the conclusions.

Lines 620-622: "Therefore, it is important for more HTAP2 participating models to save their outputs hourly in order to conveniently compute the policy-relevant metrics for the O3 sensitivities." I agree with this statement, and moreover, I do not think you can do a meaningful analysis of any models that do not save the hourly outputs (or 3-hour if that is the finest time resolution used), and I would recommend excluding them from this study.

Again the monthly mean based analysis are shown to connect with the HTAP1 and other HTAP 2 modeling results that are done on a monthly basis. Although not directly relevant to the air quality management, these provide a broad characterization perspective (also considered in the 2016 EPA White Paper). We included a separate section (3.3) in the paper showing event based analysis.

Lines 612-624 and Figure 9: It is obvious that day time O3 is greater than nighttime O3 at surface sites because O3 deposits to surfaces and is destroyed by chemical reactions at night. So the findings in this text that the maximum daily 8-hour average O3 is greater the 24-hour average O3 is self-evident. I suggest deleting this text. I also recommend focusing the analysis on maximum daily 8 hour averages, especially for the highest O3 days, and not showing results for monthly mean O3.

The original Figure 9 has been moved to the supplement, and the MDA8-based analyses have been extended. Figure 2c-d also show the diurnal cycles of the total $O_3$ and R ($O_3$, EAS, 20%) values.

Line 633: "R(MDA8, EAS, 20%) is smaller during the high O3 total days in all subregions." For GEOS-Chem the contribution appears to be the same on high O3 days compared to all days, and the results are very similar for RAQMS. It would be helpful to show more details for this analysis. Is this the mean O3 for all sites for days in which any monitor was > 70 ppb, or does it only include data for the monitor that was greater than 70 ppb? I suggest performing a more detailed analysis, e.g., show EAS contribution on each day for a few key sites that frequently have high O3, e.g., Great Basin and Canyonlands sites.

In the earlier version, regionally averaged (not only at CASTNET sites), and for each location/grid, only when the predicted total $O_3$ was over 70 ppbv. We now show averaged calculations and spatial plots at all CASTNET sites for all days and during the observed $O_3$ exceedances (Figures 11-12) and extended the discussions in text. These included Canyonlands and Great Basin. Canyonlands NP is also one of the sites that experienced $O_3$ exceedances on 9 May (Section 3.3).

Line 655: "We found that the underestimated free tropospheric O3 from the STEM simulations that used any single free-running chemical boundary conditions contributed to the underestimated STEM surface O3 in the high elevation mountain states." Need to edit and clarify meaning of the above sentence. Was this because the global models underestimated stratospheric O3 or international transport?

It could be a result of the underestimation of both. The possible uncertainty in LRT of Asian pollution was determined with the help of evaluation at the source side. We also added a sentence in this paragraph about the limitation of models representing the stratospheric intrusion: "As the enhancement of $O_3$ due to the assimilation is much larger than the $O_3$ sensitivities to the EAS anthropogenic emissions, the assimilation mainly improved the contributions from other sources, such as the stratospheric $O_3$."

**References (those not cited in the text but in this response file)**

Choi, Y., J. Kim, A. Eldering, G. Osterman, Y. L. Yung, Y. Gu, and K. N. Liou (2009), Lightning and anthropogenic $NO_x$ sources over the United States and the western North Atlantic Ocean: Impact on OLR and radiative effects, Geophys. Res. Lett., 36, L17806, doi:10.1029/2009GL039381.

Surendran, D. E., S. D. Ghude, G. Beig, C. Jena, D.M. Chate (2016), Quantifying the sectoral contribution of pollution transport from South Asia during summer and winter monsoon seasons in support of HTAP-2 experiment, Atmos. Environ., 145, 60-71, doi:10.1016/j.atmosenv.2016.09.011.

Wolfe, G. M., T. F. Hanisco, H. L. Arkinson, T. P. Bui, J. D. Crounse, J. Dean-Day, A. Goldstein, A. Guenther, S. R. Hall, G. Huey, et al. (2015), Quantifying sources and sinks of reactive gases in the lower atmosphere using airborne flux observations, Geophys. Res. Lett., 42, 8231–8240, doi:10.1002/2015GL065839.

---

## Author Comment (AC3) · 1 Mar 2017

**Response to Reviewer #2's comments**

This paper represents a huge task, assembling and comparing the results from the multi-model HTAP2 study. It is a brave undertaking. However, I do have some concerns about what was learned in the process. I believe the stated goals of the paper are not well met or met in a cursory fashion. There are a number of inferences stated as fact but not in fact proved. In some cases more analysis seems to be needed. In other cases a clearer explication of what has been learned would be helpful. Recommendations about future work are succinctly summarized, but the paper needs to be stronger in detailing what was learned and in justifying the methodology used.

We thank the careful review by Reviewer #2. Please see below our response (in blue) to the general and specific comments (in black). Additional results and discussions have been added to the text to clarify the methodology and help strengthen the key points.

Major Comments:

I) The stated paper goals are to address: "1) the differences in O3 sensitivities generated from the HTAP2 and HTAP1 experiments to help address how the LRT impacts on NAM changed through time; 2) how the multi-model approach, as well as the refined model experiment design in HTAP2 can help advance our understanding of the LRT impacts, especially the benefits of increasing the global models' resolutions and involving the regional models; 3) the usefulness of satellite observations for better understanding the sources of uncertainties in the modeled total O3 (e.g., from the emission and regional models' boundary condition inputs) as well as for reducing the uncertainties in some of these model inputs via chemical data assimilation." As the paper stands it is not clear if it achieved its goals. The answers to these questions should be clearly articulated in the conclusions and in the body of the paper itself. In particular: 1) Between HTAP1 and HTAP2 models have changed, emissions have changed and the transport has changed. So it is not really clear how the sensitivity changed through time. The authors suggest many of the changes are due to the changes in emissions, but this remains to be proven. The authors could determine if the changes in the sensitivities are consistent with the change in emissions by using the HTAP1 emissions and the current sensitivities (delta O3/delta emissions) to determine if most of the changes from HTAP1 are consistent with emission changes. However, as it stands the first goal of the paper cannot be met without substantially more analysis.

The comparisons of HTAP1 and HTAP2 findings over larger spatial/temporal scales in this paper are limited to the total sensitivities themselves, and disentangling the cause of these changes is beyond the scope of this study. However, rather than simply reporting these differences, we now do have extended discussions to point out that these different sensitivities can be attributed to the following factors:
1) changes in anthropogenic emissions from 2001 to 2010 (HTAP1 to HTAP2)
2) climate variability driven interannual variability of LRT. We now cited the Lin et al. (2014) work as she suggested, in which stronger LRT impact is suggested in 2010.
3) the experimental design, including the different participating models (and even for the models that participated both HTAP1 and HTAP2, different versions and configurations were implemented), SR domain definitions

The objective 2) of this work has been modified to: "how the refined modeling experiment design in HTAP2 can help advance our understanding of the LRT impacts on NAM, particularly the

involvement of regional models and the inclusion of small spatial/temporal scale analysis during high $O_3$ episodes that are more relevant to air quality management." These also help address your following general comments.

2) It is not clear how this study enhanced our understanding of LRT nor is it very clear how changes in model resolution impacted the solutions. The STEM model resolution is 60x60 km, actually rather comparable to a global model of 1o resolution (about 85 km at 40N). While there is a wide range of different resolutions in the global models it is unclear how this paper really explored the impact of resolution on the results. What aspects of LRT did the paper enhance? This should be clear in the paper.
Please see our response to your general comment II.

3) The usefulness of satellite data is essentially a "motherhood" statement. It is somewhat unclear how this paper further showed this usefulness. This is especially true since the case study using satellite data was presented in a rather cursory manner.
Please see our response to your general comment III regarding the use of satellite data in the case study. The text in the introduction, Section 2 and Section 3.1 explain the purpose, methods, and the findings of using OMI $NO_2$ data to evaluate the bottom-up emissions. These were also explicitly mentioned in Section 2.3.2, and in the abstract and Section 4 as a highlight in HTAP2.

We also make the readers aware of the uncertainty of the satellite products. For example, for the use of OMI $NO_2$: "It is important to note that uncertainty in satellite retrievals can prevent us from producing accurate assessment on emissions (e.g., van Noije et al., 2006), and this comparison does not account for the biases in the used OMI data, and would be further validated by using other OMI $NO_2$ products as well as the bias-corrected (if applicable) in-situ $NO_2$ measurements." For TES and IASI $O_3$, "TES $O_3$ is generally positively biased by <15% relative to high accuracy/precision reference datasets (e.g., Verstraeten et al., 2013). Although IASI is in general less sensitive than TES due to its coarse spectral resolution, the 681–316 hPa partial column-averaged $O_3$ mixing ratios in the JPL product agree well with TES $O_3$ for the 2008–2011 period with a -3.9 ppbv offset (Oetjen et al., 2016)."

II) It is not clear what the goal of using the STEM model is here. As pointed out above the resolution is not that much higher than some of the global models that give the boundary conditions. Differences between the STEM results and the boundary condition model could be due to the different chemistry in the two models or due to the differences in transport. Driving the models with different meteorological datasets also risks an inconsistency in the boundary conditions (e.g., chemical plumes transported in the jet in the parent model might be mismatched with the jet in STEM). At any rate the rationale for the use of the STEM model should be clearly articulated. What did we learn by coupling the global models with the STEM model?
As been pointed out in the text and recognized by Reviewers #1 and #3, the use of STEM model here is to test the global-regional model couplings. In Sections 1 and 2, we introduced that "For regional simulations over the North America and Europe, boundary conditions were mostly taken from a single model such as the ECMWF C-IFS or GEOS-Chem.", while in this study we "Extending the HTAP2 regional simulations' basic setup, the STEM top and lateral chemical boundary conditions were downscaled from three global models' (i.e., the Seoul National University (SNU) GEOS-Chem, RAQMS, and the ECMWF C-IFS)". As a key finding of this

work, which is also relevant to your next comment, we did show in case studies that all of the global models performed poorly for for some high $O_3$ events (except RAQMS with data assimilation). We believe such uncertainty poses difficulties for regional models (regardless of their resolutions and other configurations, parameterization) to accurately estimate total $O_3$ and the SR relationships using boundary conditions downscaled from these models. This finding provides important information for future regional modeling works on higher resolutions and this point has been sharpened in the revised paper.

Please note that all three global models used to be coupled with STEM are known to have satellite chemical data assimilation capability. Given that satellite assimilation can improve the modeled $O_3$ performance (as demonstrated in this paper for STEM/RAQMS and in a previous study for STEM/GEOS-Chem), near the end of the paper, we suggested directions for future multi-scale modeling works: "As chemical data assimilation techniques keep developing (Bocquet et al., 2015), several HTAP2 participating global models have already been able to assimilate single- or multi-constitute satellite atmospheric composition data (e.g., Miyazaki et al., 2012; Parrington et al., 2008, 2009; Huang et al., 2015; Inness et al., 2015; Flemming et al., 2017). Comparing the performance of the assimilated fields from different models, and making the global model assimilated chemical fields in the suitable format for being used as boundary conditions would be very beneficial for future regional modeling, as well as for better interpreting the pollutants' distributions especially during the exceptional events…."

We used STEM calculations also because we saved STEM $O_3$ calculations hourly everywhere within the regional domain, while most of the HTAP2 global models did not do so in all model grids. Using hourly observations is important to generating more accurate MDA8 based analysis and comparing the model fields with satellite observations, which are more policy relevant and are favored components by other reviewers.

While we agree that ideally it'd be better to perform all STEM simulations on a finer resolution grid, that has been determined to be not so practical due to the limitations in time and computational resources, especially that the STEM modeling work shown here is a voluntary/unfunded activity. However, 12 km STEM/RAQMS test simulations were indeed performed and the results have been presented at previous HTAP workshops (e.g., http://www.htap.org/meetings/2015/2015_May_11-15/Powerpoint%20Presentations/Monday/Huang%20HTAP_05112015.pdf). These simulations were not updated to account for the later updates in the HTAP2 emission inventory and are therefore not suitable to be included in this manuscript. However, the findings are overall qualitatively similar to the results based on the 60 km simulations in this paper, for example, STEM/RAQMS and RAQMS show similar spatial patterns and domain-mean values of the sensitivities; STEM/RAQMS free run and RAQMS free run show negative biases (relative to CalNex ozonesonde and aircraft in-situ measurements) in free tropospheric $O_3$ during high $O_3$ episodes, which were reduced by satellite data assimilation.

Yes, it is understood that "Driving the models with different meteorological datasets also risks an inconsistency in the boundary conditions (e.g., chemical plumes transported in the jet in the parent model might be mismatched with the jet in STEM)." This does not seem to be a big issue in this study. However, we do think that it is worth carrying out additional experiments in the future to

determine if such inconsistency can be resolved by using the boundary condition models' meteorological fields as WRF's initial and boundary conditions.

III) The case study is rather thin. What are the goals of this section? This section should either be expanded or dropped.

Section 3.3 includes event-based analysis that is more relevant to air quality management than the larger scale results, which is favored by other reviewers, and as a result is an important part of this paper. We have expanded this section and added a summertime case study for comparison as other reviewers suggested. Figures 14/17 evaluate the modeled $O_3$ vertical distributions during LRT events, showing that all of the global models performed poorly for for these high $O_3$ events (with the exception of RAQMS with data assimilation). The underestimated "transported background" $O_3$ levels were connected with the underpredicted surface $O_3$ exceedances in the western US shown in Figures 15/18. We believe such uncertainty poses difficulties for regional models (regardless of its resolution and other configurations, parameterization) to accurately estimate the total $O_3$ and the SR relationships using boundary conditions downscaled from these global models.

As other types of observed $O_3$ vertical profiles, such as ozonesonde data, are not available during one of the events we show and are only available in limited regions (only in California) during another event, we believe that evaluating the boundary condition models using satellite $O_3$ vertical profiles during the selected high $O_3$ episodes is new and very informative.

Specific Comments:

1. L42. The sentence beginning is rather awkward. Consider rewording.

Reworded.

2. L48-49, "This indicates. . ...". This has to be proven. As is well known interannual variability of the atmosphere is substantial.

Interannual variability has been included in the discussions, which also addresses the comments by Dr. Lin and Reviewer #3.

3. L175. Starting here the manuscript goes into considerable detail about how the simulatons are set up. This does not work well in the introduction, but belongs in the methodology section.

This paragraph has been substantially modified, with specific goals of the study stated first (also accounting for Reviewer #3's suggestions), and some details of the methods were moved to Section 2.

4. L202, Section 2.1. The manuscript parses the emissions between East Asia, MICS Asian regions and south Asian countries. The domains of each these regions is not clear.

MICS Asia is defined in text as: "MICS-Asia regions (south, southeast, and east Asia, based on country inventory for China and from the Clean Air Policy Support System and the Regional Emission inventory in ASia 2.1, more information also in Li et al., 2017)…" Figure 1 defines the different part of the Asian regions for HTAP2's SR relationship study.

5. Table 1. All abbreviations should be defined. Also the table headings need to be reformatted.

Done.

6. L250-253. This notation is should be improved: the left hand side of the equation has a percentage sign, but not the right. I would suggest something like EASALL(-20%) on the right to distinguish this from the R(O3, EAS, 100%) where presumably all EAS emissions are reduced by 100%.
Done.

7. L290 and following paragraph. A long discussion is presented concerning STEM lightning emissions, biogenic emissions and VOC speciation. How were these emissions parameterized in the other models, the same as STEM or differently? Please specify more thoroughly differences in emissions between STEM and other models.
The non-anthropogenic emissions do differ by models, which impact the background $O_3$ estimation. See Table 1c, Figure S1 for detailed comparisons between GEOS-Chem and STEM, as well as summary for the boundary condition models. We agree and suggest that for future activities the non-anthropogenic emissions should be formally reported for all models by region and species. We now added in Section 2.1: "Non-anthropogenic emission inputs used in different models' simulations may differ, and their impacts on the modeled total $O_3$ and the SR relationships will be compared in detail in future studies." And for STEM and its BC models at near L290, we added: "Note that non-anthropogenic emission inputs used in STEM and its boundary condition models differed, as summarized in Table 1c. Figure S1 shows detailed comparisons between STEM and GEOS-Chem's non-anthropogenic (i.e., soil, lightning, biomass burning) $NO_x$ emission inputs, and their impacts on the modeled NAM background $O_3$ were included in Lapina et al. (2014). Such quantitative comparisons will also be carried out between STEM and its other boundary condition models in future studies."

8. L394. "less sensitive" – less sensitive to what?
To the changes in the "true" state, which can be measured by the averaging kernels. This is introduced by a sentence in the following paragraphs: "$A_{TES}$ is the averaging kernel matrix reflecting the sensitivity of retrieval to changes in the true state (Rodgers, 2000)." Comparison of the TES and IASI sensitivities can be found in Oetjen et al. (2016).

9. L420. "de-stripped" – the meaning is unclear.
Corrected to "de-striped". This is described in Boersma et al. (2011a) which we cited.

10. L459. "suggesting that using". This seems rather speculative. There are many possible explanations.
The discussion has been changed to: As reported in the literature (e.g., Geddes et al., 2016; Travis et al., 2016), the representation of land use/land cover, boundary layer mixing and chemistry can be sources of uncertainty for certain global model (i.e., GEOS-Chem), but how serious these issues were in the other models need to be investigated further. Some other possible reasons include the variation of these models' non-anthropogenic emission inputs and chemical mechanisms (Table 1c). Future work should emphasize on evaluating and comparing all models on process level to better understand their performance. Except in the northeastern US, the eight-model ensembles show better agreement with the CASTNET $O_3$ observations than the three boundary condition-model ensemble. Overall the three-model ensemble only outperforms one model but the eight-model ensemble outperforms seven. This reflects that averaging the results from a larger number

of models in this case more effectively cancelled out the positive or negative biases from the individual models."

11. L471-472. "overall there does appear to be a positive bias". This seems to be a rather strong statement considering the previous sentence. It would be better to say satellite is consistent with a positive bias.
This sentence has been reworded to: "While grid-scale differences in $NO_2$ columns may not be directly indicative of emissions biases (Qu et al., 2016), these discrepancies are possibly due to a positive bias in the bottom-up emissions, mainly from the anthropogenic sources, which have also been pointed out by Anderson et al. (2014) and Travis et al. (2016)."

12. L478-479. Can you provide a reference why co-emitted species are likely to be biased in the same way as NOx. It is not at all clear to me that emission factors would be all biased in one direction.
Janssens-Maenhout et al. (2015) and Li et al. (2017) summarized that generally the uncertainty ranges are relatively small for species whose emissions are dominated by large-scale combustion sources but larger for those from small-scale and scattered sources. Based on such information, this part of discussion has been modified to reflect the sector and species dependent uncertainty ranges. Additional text was added to Section 2.1 as well.

13. L509-510. "mainly due to". Maybe. It would be better to say consistent with.
"mainly" was changed to "in part".

14. L556-557. Did you show this? Probably better to say "consistent with".
The literature we cited showed this. This sentence has been changed to "the substantial improvement in the European air quality over the past decades that is shown in Crippa et al. (2016) and Pouliot et al. (2015), which contrasts with the growing anthropogenic emissions from the East Asia and other developing countries during 2001-2010". Discussions were also extended to other reasons causing the differences between HTAP1 and HTAP2.

15. L567-568. This is an interesting result: that R in HTAP2 is larger than in HTAP1. However, the reasons for this have not been clearly shown. Certainly the difference is consistent with emission trends but the authors need to establish that this is the case (see general comments above)
Please see our response to your first general comment.

16. Figure 9. I think this is a scatter plot of R(MDA8,EAS,20%) and R(O3, EAS, 20%). Please address the notation.
Figure 9 in the original submission to ACPD in Oct 2016 was moved to the supplement per Reviewer #1's suggestion.

17. The point of section 3.3 is not clear. Some of the figures panels in this section seem to be referred to in a very cursory manner or not at all (e.g., Figure 11). This section needs to be much better developed or not presented.
Same as the response to your general comment (III): Section 3.3 includes event-based analysis that is more relevant to air quality management than the larger scale results, which is favored by other reviewers, and as a result is an important part of this paper. We have expanded this section and

added a summertime case study for comparison as other reviewers suggested. Figures 14/17 evaluate the modeled $O_3$ vertical distributions during a LRT event, showing that all of the global models performed poorly for for some high ozone events (with the exception of RAQMS with data assimilation). The underestimated "transported background" $O_3$ levels were connected with the underpredicted surface $O_3$ exceedances in the western US shown in Figures 15/18. We believe such uncertainty poses difficulties for regional models (regardless of its resolution and other configurations, parameterization) to accurately estimate the total $O_3$ and the SR relationships using boundary conditions downscaled from these global models. As other types of observed $O_3$ vertical profiles, such as ozonesonde data, are not available during one of the events we show and are only available in limited regions (only in California) during another event, we believe that evaluating the boundary condition models using satellite $O_3$ vertical profiles during selected high $O_3$ episodes is new and very informative.

18. Figure 7 caption. I assume (a), (b), and (c) refer to the first three rows. Better to say row 1, row2 and row 3 or label all panels with letters.
We have labelled all panels of this figure with letters.

---

## Author Comment (AC4) · 1 Mar 2017

**Response to Reviewer #3's comments**

We thank the careful review by Reviewer #3. Please see below our response (in blue) to the general and specific comments (in black).

General Comments

This manuscript presents the first HTAP Phase II findings, expanding on HTAP Phase I by incorporating regional models to estimate the impact of international anthropogenic emissions on U.S. surface ozone. The authors use boundary conditions from three different global models to drive the regional STEM model, and compare the sensitivities of surface ozone in North America to international anthropogenic emissions with those determined from 8 global models. They further compare with an adjoint version of one model, use boundary conditions from a model that assimilated satellite ozone products, and conduct a case study using multiple satellite and ground-based products. This is a major undertaking, as noted by another reviewer. I agree with the other reviewers, however, that the paper suffers from some shortcomings. Several of the issues I was planning to cover were discussed at length in the earlier reviews, so I focus below on additional points. I'd like to see the abstract/conclusions clarify and quantify (e.g., within 10%, 30%, factors of 2?) the conclusions regarding how different the global and regional model estimates are, and how much the RER sensitivity estimates have changed from those reported in the 2010 HTAP report. I agree with Dr. Tonnesen that more emphasis on episodic events would enhance the policy relevance of this work. Throughout the text, more quantitative and specific language should be used wherever possible, and the paper should be edited carefully for clarity (e.g., incomplete sentence L768). The introduction is quite long and could state earlier on what the point of this study is to provide a context before going into all the details of past work.

Quantitative language is now used in places as you suggested, especially in the abstract and the conclusions. The R values were compared with the HTAP1 results in the 2010 report, while the RERER calculation is a new element in HTAP2. Taking Dr. Tonnesen's suggestion, we added a summer event case study, drawing some similar conclusions to the 9 May event. We also show averaged calculations and spatial plots at all CASTNET sites for all days and during the observed $O_3$ exceedances (Figures 11-12).

Manuscript has been carefully and extensively edited. The sentence starting at L768 now reads as: "As emissions from various source sectors can differ by emitted altitudes and temporal profiles, efforts should also be placed to have the models timely update the height and temporal profiles of the emissions from various sectors." The last paragraph of the introduction section was substantially modified, with the specific goals of this study stated first and some details of the methods moved to later sections.

Specific Comments L42-45. Elaborate on what this means for drawing conclusions regarding the role of hemispheric transport of air pollution.

We found that the differences between STEM surface $O_3$ sensitivities and its corresponding boundary condition model's are often smaller than those among its boundary condition models. We also reported the differences between boundary models and all global models. We agree that these are key findings of this paper, indicating that accurately attributing pollution in the global model(s), which still appeared to be difficult, is a critical first step for any follow-on estimates

based on regional models using the boundary conditions downscaled from these global models. We have rewritten this (and later) sentences.

L48 'Tagged tracer approach' is mentioned here and elsewhere (e.g. L564); a brief explanation is needed as approaches can involve tagging ozone itself or tagging precursors.
This is a good point. The Asian $O_3$ in Brown-Steiner and Hess (2011) means any $O_3$ created as a result of anthropogenic+biofuel $NO_x$ emissions (with no interannual variability) over the East Asia. They should be compared with EAS $NO_x$ emission perturbation runs. However, we here only used sensitivities to EAS $NO_x$ emission perturbation from GEOS-Chem, so the direct comparisons in the abstract and multiple place in the text were removed. The seasonality based on tagging and 20% $NO_x$ emission perturbation was compared in Section 3.2.1 instead.

I'm not convinced that this study cleanly isolated the role of rising East Asian anthropogenic emissions; see also RC2 comments.
Same as the response to RC2's comments:
The comparisons of HTAP1 and HTAP2 findings over larger spatial/temporal scales in this study are limited to the total sensitivities themselves, and disentangling the cause of these changes is beyond the scope of this study. However, rather than simply reporting these differences, we now do have extended discussions to point out that these different sensitivities can be attributed to the following factors:
1) changes in anthropogenic emissions from 2001 to 2010 (HTAP1 to HTAP2)
2) climate variability driven interannual variability of LRT. We now cited the Lin et al. (2014) work as she suggested, in which stronger LRT impact is suggested in 2010.
3) the experimental design, including the different participating models (and even for the models that participated both HTAP1 and HTAP2, different versions and configurations were implemented), SR domain definitions

L51 Are the adjoint sensitivities compared to all the global models or just the forward version of GEOS-Chem? Is this the same version as used to provide boundary conditions? (see also L591)
Just GEOS-Chem's. The CU and SNU GEOS-Chem are different.

L54-56 Try to quantify this statement: is it off by 20%? Factor of 2?
Done.

L57-59 This appears to be a general statement rather than a conclusion drawn from this work and thus does not seem appropriate to include in the abstract.
More conclusive language is now used to describe the findings from satellite data related work in this paper.

L96 The first paper to show this was Jacob et al., GRL, 1999: http://onlinelibrary.wiley.com/doi/10.1029/1999GL900450/abstract
Cited.

L148. Region-dependent, but also time-dependent?
Added "time-".

L220-227 Seems relevant to provide BVOC emissions over Asia and North America. How much do North American anthropogenic emissions contribute to global totals?

North American anthropogenic emissions contribute to global totals can be calculated by numbers in Table S1. A sentence is added to Section 2.1: "In 2008, NAM $NO_x$, NMVOC and CO contributed to 18.0%, 11.7% and 11.9% of the global total, respectively, and in 2010, these contributions became 15.8%, 10.5% and 10.2%.".

The non-anthropogenic emissions do differ by models, which impact the background $O_3$ estimation. See Table 1c, Figure S1 for detailed comparisons between GEOS-Chem and STEM, as well as summary for the boundary condition models. We agree and suggest that for future activities the non-anthropogenic emissions should be formally reported for all models by region and species. We now added in Section 2.1: "Non-anthropogenic emission inputs used in different models' simulations may differ, and their impacts on the modeled total $O_3$ and the SR relationships will be compared in detail in future studies." And for STEM and its BC models at near L290, we added: "Note that non-anthropogenic emission inputs used in STEM and its boundary condition models differed, as summarized in Table 1c. Figure S1 shows detailed comparisons between STEM and GEOS-Chem's non-anthropogenic (i.e., soil, lightning, biomass burning) $NO_x$ emission inputs, and their impacts on the modeled NAM background $O_3$ were included in Lapina et al. (2014). Such quantitative comparisons will also be carried out between STEM and its other boundary condition models in future studies."

L233 References could be included in Table 1
Done. Related text and Table 1 caption was modified accordingly.

L238 Why are boxes shown in Figure 1 if the regions are actually following the political boundaries as indicated in L258?

The boxes were used to highlight the three focused source regions (EAS, SAS, EUR) rather than defining the boundaries of these regions, as mentioned in the figure caption. To avoid the confusion, we instead highlighted these three regions by underlining the region names in the map.

L276. Given that Lin et al. 2012 estimated Asian ozone pollution transport to the western U.S. using a global model about this resolution, a case needs to be made for why it's appropriate to use a regional model (e.g., allows testing of multiple boundary conditions, and regulatory applications would presumably run at finer scales).

Lin et al. (2012a) used a different model (parameterizations are different) with different configurations (e.g., the emission input). They mainly focused on the western US, and the impact of data assimilation on the modeled $O_3$ was not addressed in that study. As included in the discussions, all R values during the exceptional events are smaller than 1/5 of their reported sensitivities, due to the differences in model parameterizations and configurations. Some related discussions can be found in Section 3.3.

In terms of the use of regional models, we agree with your suggestion that it allowed us to test the multiple boundary conditions. And, same as our response to Reviewer #2's comment: In Section 1 and 2, we introduced that "For regional simulations over the North America and Europe, boundary conditions were mostly taken from a single model such as the ECMWF C-IFS or GEOS-Chem.", while in this study we "Extending the HTAP2 regional simulations' basic setup, the

STEM top and lateral chemical boundary conditions were downscaled from three global models' (i.e., the Seoul National University (SNU) GEOS-Chem, RAQMS, and the ECMWF C-IFS)". As a key finding of this work, we did show in case studies that all of the global models performed poorly for for some high $O_3$ events (except RAQMS with data assimilation). We believe such uncertainty poses difficulties for regional models (regardless of its resolution and other configurations, parameterization) to accurately estimate the total $O_3$ and the SR relationships using boundary conditions downscaled from these models. This finding provides important information for future regional modeling works on higher resolutions and this point has been sharpened in the revised paper.

We also make the readers be aware that all three global models used to be coupled with STEM are known to have satellite chemical data assimilation capability. Given that satellite assimilation can improve the modeled $O_3$ performance (as demonstrated in this paper for STEM/RAQMS and in a previous study for STEM/GEOS-Chem), near the end of the paper, we suggested directions for future multi-scale modeling works: "As chemical data assimilation techniques keep developing (Bocquet et al., 2015), several HTAP2 participating global models have already been able to assimilate single- or multi- constitute satellite atmospheric composition data (e.g., Miyazaki et al., 2012; Parrington et al., 2008, 2009; Huang et al., 2015; Inness et al., 2015; Flemming et al., 2017). Comparing the performance of the assimilated fields from different models, and making the global model assimilated chemical fields in the suitable format for being used as boundary conditions would be very beneficial for future regional modeling, as well as for better interpreting the pollutants' distributions especially during the exceptional events…."

L283. This may be true for the Asian pollution transport, but Lin et al. 2015 indicate that 2010 isn't a particularly high year for stratospheric intrusions reaching surface air over the WUS. http://www.nature.com/articles/ncomms8105 See their figure 2.
This part has been modified to focus on the interannual variability of LRT of Asian pollution.

L287-88 Is this just reflecting the warming trend over the 81-00 period? Temperatures and ozone production were even higher in 2011 and 2012 in the eastern US.
Yes. As it's based on "the climatology from the NCEP/NCAR reanalysis data for the 1981-2010" described earlier in this paragraph. This paper does not cover the periods after 2010.

L315 How was this downscaling done?
Standard downscaling approach: spatial/temporal interpolation and species mapping.

L442 perhaps needs a reference for the HTAP1 work unless this was done as part of this study?
Added Fiore et al. (2009).

L445-447. It's not clear what the take-away point is here. Are the models underestimating Asian pollution influence or can we not tell because it could be regional transport? Presumably even though the data assimilation fixes this problem, it does not help us to distinguish between these possible sources of error?
This sentence just lists the possible sources of error, including both trans-boundary (see case study for details) and regional transport, but it does not distinguish/quantify the impact from each factor.

L472. How did this study determine that the bias is likely due to overestimated anthropogenic NOx emissions? May doesn't look like it has a clear bias whereas July does. How do we know this is associated with anthropogenic sources rather than seasonally varying sources like soil NOx for example? Are there seasonal variations in the anthropogenic NOx emissions?

Both anthropogenic and non-anthropogenic emissions are time-varying. Anthropogenic emissions differ by month (Section 2.1) while many non-anthropogenic emissions are weather dependent and display stronger temporal varibility. Overall anthropogenic $NO_x$ emissions contribute most to the total $NO_x$ emissions, but the uncertainty can definitely be due to those from other emission sources. We now added to the SI the natural emissions from GEOS-Chem and STEM in June 2010, and mentioned about the possible overprediction in soil/lightning $NO_x$ in the central/eastern US near L472: "Larger OMI-model disagreement was found over the central/eastern US during June 2010, likely also due to the uncertainty in GEOS-Chem's soil or lightning $NO_x$ emissions, which appear to be high over these regions (Figure S1)".

L481-483. Doesn't this interpretation depend on where the photochemical regime is at in terms of ozone production with respect to NOx emissions?

We added "Under different chemical regimes," before "this statement would also rely on the quality of other $O_3$ precursors in the HTAP2 emission inventory..".

L520 An estimate of how large these biases are and how much bias they introduce into ozone would be useful here.

The biases are time- and region- dependent and in part depend on the quality of the WRF inputs. We added the findings from Huang et al. (2017) on the Sep 2013 conditions for MO and TX regions. Quantifying the impacts of overestimated biogenic emissions and the biased weather fields that contributed to the biases in emissions on the modeled $O_3$ is still an ongoing work.

L541-544. There seems to be model disagreement near the Canadian border, with Oslo for example suggesting high cross-border influence but CHASER suggesting much less. L544-547. Is Oslo also higher resolution as it looks similar to EMEP in terms of higher influence.

OsloCTM3's horizontal resolution is 2.8°×2.8° (Table 1a), but we noticed that the number of its vertical layers, which affects the export and import of pollution, are larger than the rest of the models'. The number of vertical layers for each model are now added to Table 1. We added: "Although on a coarse horizontal resolution of 2.8°, OsloCTM3 suggests stronger extra-regional source influences on the northwestern US and the US-Canada border regions than the other models. Its largest number of vertical layers among all global models might be a cause."

L585-588 Where is this shown?

This is a general statement pointing out the other key sources.

L612-616. Can you provide estimates of how the ozone lifetimes in the boundary layer differ in the different simulations?

This is a good suggestion that we did not prepare for HTAP2 and is in need for future analysis.

L627-628. There seems to be an assumption that LRT is obvious from satellite data. This isn't the case for ozone. How will LRT be convincingly separated out from other ozone sources?

All observations, not only the satellite observations, represent the total $O_3$. The use of satellite $O_3$ and CO can distinguish anthropogenic/biomass burning sources from the stratospheric intrusions, and additional tools and data will also be helpful. However, this sentence is to say that the broader coverage provided by the future satellites (than the CASTNET network) would better help capture polluted events. As Dr. Lin also pointed out in their 2015 GRL paper, the sampling strategy does affect the calculated pollution trends and source attribution, and in the paper we compared sensitivities in all grids v.s. only at CASTNET sites.

L658-659. Did all models capture the same events in terms of their timing and approximate regional location?
Qualitatively similar. We added "(based on three boundary condition models separately and averagely)". See revised Figure 2b (the thin lines show individual models' EAS sensitivity for the western US) and the case studies for the detailed comparisons (e.g., Figures 15/18).

L661. It would be more convincing to show this as a monthly mean diurnal cycle rather than rely on Figure 2a.
Time series in Figure 2b (previously Figure 2a) shows the 3-6 LRT events during May-June 2010. Period-mean diurnal cycles are now also shown in Figures 2c-d for total $O_3$ and the EAS sensitivities, respectively.

L679-681. I didn't follow this point.
STEM base simulations overall substantially overpredicted the total $O_3$ in non-western US regions based on our evaluation at the CASTNET sites, as described in the previous sections. So the R(MDA8, EAS, 20%) calculated during the days of $O_3$ exceedances (based on the STEM-estimated total $O_3$ in all model grids) can actually represent the sensitivities during some days when total $O_3$ actually did not exceed 70 ppbv. We now also added that some of the exceedances in the western US were not correctly captured which also affected conclusions from this figure.

L741-742. Be more specific here.
Quantitative results are summarized here and in the abstract, which also addressed your general comment.

L744-747. Is there a relationship between the bias and the Asian transport events?
The biases in modeled total $O_3$ are attributed to those in the modeled LRT Asian pollution as well as other factors. But the model that predicted the higher $O_3$ does not always gave higher estimates of the EAS contribution, as shown in the case study.

L747-749. It's not clear how better quantifying stratospheric o3 intrusion helps reduce North American pollution levels and model uncertainties. This statement also implies that stratospheric intrusions are as important as local ozone formation.
This paragraph has been rewritten to suggest impact from bottom-up emission input and future work on attributing the intermodel differences and model biases.

L750. How frequent are these episodic sensitivities to East Asian emissions? Are they occurring when measured ozone is highest?

This paragraph has been rewritten based on the additional analyses we performed for high $O_3$ days: "The STEM $O_3$ sensitivities to the East Asian anthropogenic emissions (based on three boundary condition models separately and averagely) were strong during 3-6 episodes in May-June 2010, following similar diurnal cycles as the total $O_3$. Stronger-than-normal East Asian anthropogenic pollution impacts were estimated during $O_3$ exceedances in the western US, especially over the high terrain rural/remote areas; in contrast, non-local pollution impacts were less strong during $O_3$ exceedances in other US regions."

L800-801. These suggestions seem to neglect the important caveat that these approaches assume that model transport is perfect.
This depends on what kind of model(s) to be used. For online models, weather fields may be modified together with the chemical fields; For offline models, you are right, but these suggested methods still incrementally improve the source attribution and should be encouraged.

---

## Author Comment (AC5) · 1 Mar 2017

**Response to Dr. Lin**

We thank the additional references suggested by Dr. Lin and have made changes to the manuscript accordingly. Please see below our responses in blue.

1. Lines 93-100, Page 3: Regarding Asian influence on US ozone trends, please consider citing the following papers and discuss their findings:

Lin, M., L.W. Horowitz, O.R. Cooper, D. Tarasick, S. Conley, L.T. Iraci, B. Johnson, T. Leblanc, I. Petropavlovskikh, E.L. Yates (2015): Revisiting the evidence of increasing springtime ozone mixing ratios in the free troposphere over western North America, Geophysical Research Letter, 42, doi:10.1002/2015GL065311

Lin, M., W. Horowitz, R. Payton, A.M. Fiore, G. Tonnesen. US surface ozone trends and extremes over 1980-2014: Quantifying the roles of rising Asian emissions, domestic controls, wildfires, and climate. Atmos. Chem. Phys. Discuss., doi:10.5194/acp-2016- 1093, 2016

You cited Cooper et al. (2010, Nature). But Lin et al. (2015 GRL) investigated the representativeness of ozone trends derived from sparse measurements reported by Cooper et al. They found that sampling biases can substantially influence calculated ozone trends.

Both papers are now cited in this paragraph. We do agree that the observation sampling methods affect the calculated pollution trends and SR relationships, and related discussions have been included in several places in Section 3, e.g., comparing R values averaged over all grids and only sampled at the CASTNET sites.

2. The multi-model results presented in this article are based on the spring of 2010 following strong El Nino conditions. I think it would be useful to the readers if you can discuss the representativeness of your results on inter-annual context. There are studies showing that long-transport transport of Asian pollution is stronger during El Nino springs due to the eastward extension and equator-ward shift of the subtropical jet stream (e.g., Lin et al., 2014, Nature Geoscience).

We agree that discussing the results on inter-annual context is important. Interannual differences of LRT of Asian pollution due to the impact of atmospheric circulation v.s. anthropogenic emission changes are briefly discussed in Sections 2 and 3.2, and are highlighted in the abstract and conclusions. A sentence has been added to Section 2.1 mentioning the findings in Lin et al. (2014): "This is consistent with the findings by Lin et al. (2014) that the El Niño conditions during the 09/10 winter strengthened the trans-Pacific transport of Asian pollution in spring 2010."

---

## Author Response (AR2)

**Response to Reviewers' comments**

We appreciate the re-reviews by Anonymous Reviewers #2 and #3. Please see below our response (in blue) to their comments (in black).

**Reviewer #2:**

I would suggest accepting contingent on the following minor corrections:

L44-45: "full source contribution": the meaning is not altogether clear. I assume you mean zeroing-out the emissions.
Added: "(i.e., based on 100% emission perturbation)"

L45: "to a 20%" should be "from a 20%"
We changed the "obtained by" to "obtained from". The "to" before "a 20%" goes with "sensitivities".

L47: the meaning of O3 sensitivities is not clear here
It means: sensitivities to a 20% reduction in the EAS anthropogenic emissions. This has been clarified.

L50-51: I would remove the list of possible differences between HTAP1 and HTAP2 results. This list is not all inclusive (for example it could be due to different models or different model designs etc). Is is sufficient, especially in the abstract to say they are different.
We deleted the list here in the abstract per your suggestion, but added "a number of reasons including" before this list in the conclusion. "the different experiment designs of HTAP1 and HTAP2" we include in this list contains the meaning of "different models or different model designs etc".

L177: Goal 1. This goal cannot be realized due to the long list of caveats that the authors have listed. It is probably necessary to run one model configuration from the HTAP1 focus time-period to that of HTAP2 to understand how the LRT impacts change through time. I would suggest changing this goal to comparing the different results between HTAP1 and HTAP2.
The suggestion of "run one model configuration from the HTAP1 focus time-period to that of HTAP2 to understand how the LRT impacts change through time" is great, and it will very likely be accounted for in future studies as discussed at an HTAP2 related workshop at the US EPA. We changed the language to "comparing the differences in…, which could help..".

L201: "Identical" seems a bit strong here due to ambiguities in VOC speciation. The speciation is discussed to some extent further down in the manuscript. It would seem relevant to discuss the VOC speciation in this section.
We agree that the VOC speciation was treated differently by each model. "Identical" here refers to whatever in the provided format, and for NMVOCs meaning the total amount.

L229-230: Biogenic emissions are tacked on here almost as an afterthought, but are discussed in more detail below. It would make sense to include a section on differences in non-anthropogenic emissions and to condense discussion on them. A little detail is given on the supplement for some of the models for non-anthropogenic NOx emissions. Why are STEM lightning NOx emissions offshore masked out? Why are the emissions differences not characterized over a broader domain (as the paper seems to imply much of the difference in the model sensitivities are due to domain boundary conditions)? Why are differences in isoprene emissions not given? It seems that a simple table in the supplement could encapsulate some of the differences in non-anthropogenic emissions between the different models in different source areas.

The materials regarding non-anthropogenic $NO_x$ emissions also supported the discussions of OMI/GEOS-Chem $NO_2$ comparison, and therefore both their spatial distributions and regional total amounts are useful for this paper. We use GEOS-Chem and STEM non-anthropogenic $NO_x$ emission comparisons as an example to make the point that the differences among individual models for individual species/source types are big, partially affecting the model performance. The comparison of GEOS-Chem and STEM were initially done to support Lapina et al. (2014), for which biogenic isoprene emissions were not compared. As we mentioned, non-anthropogenic emissions from all models by species/source types should be quantified and summarized in future studies. STEM offshore lightning $NO_x$ emissions (which were much smaller than over land and many were in pollution export regions) were masked out as at that time we in general have higher confidence in the WRF precipitation over land. Their impacts on the modeled $O_3$ inland are expected to be small.

L253: What is the "NAMALL" simulation? It does not seem to follow the naming convention.
L253: Please specify explicitly what "GLO" stands for.
NAM means the North America source region as shown in Figure 1. We now specified below the equations that "GLO" stands for the "global" source region.

L269: Please make the notation in equation (2) consistent with the notation above. In (2) some terms are written as subscripts while above they are written above in parenthesis (i.e., term 2). Term 3 seems to exclude the "-20%" in some places.
Done.

L558: Are you sure the NOx emissions are overestimated? Doesn't the paper state above that it is not straightforward to draw conclusion with respect to emissions from satellite measurements.
Changed to "the uncertainties in".

L657: Please revisit the Brown-Steiner and Hess paper. After looking at their paper I do not see that they claim a factor of 3 betweeen summer and other seasons.
These refer to results in their Table 2 (US) and 3 (North America) for Spring (MAM) and Summer (JJA) ratios.

L760-763: Please rephrase. I do not understand this point very well from what is written.
It now reads as: "Therefore, the R(MDA8, EAS, 20%) values shown in Figure 10 during the model-based periods of $O_3$ exceedances can represent the sensitivities during the actual periods of $O_3$ compliance in non-western US regions, and may not represent the sensitivities during all actual $O_3$ exceedances in the western US."

L914: "continently"

Changed to "conveniently calculating".

L879-883: I would change "due" to "due in part". This list is not really complete giving all the differences why these values may differ.
Changed to "due to a number of reasons including…"

**Reviewer #3**

The authors have addressed most of my major concerns, and I list below additional suggestions for the authors to strengthen the paper a bit more before publication. The abstract and conclusions could better convey the main findings from the paper. For example, I was surprised that the findings from Figures 10 and 11 aren't discussed.
We added a sentence in the abstract regarding Figure 10-11 findings "The EAS pollution impacts are weaker during observed $O_3$ exceedances than on all days in most US regions except over some high terrain western US rural/remote areas." The paragraph in conclusion starting from L908 discusses about these as well.

L47 not clear what the sensitivities are to (Asian emissions?)
Added: "to the 20% EAS emission perturbations"

L58-60 is confusing as these events aren't occurring at the same times in the eastern US as in the western US. How much did stratospheric and transported EAS pollution influence US ozone during these events?
Referring to at satellite overpass times. Satellite data were used to identify these episodes, distinguish LRT/stratospheric intrusions, and evaluate models, but not to quantify the contributions from each sources.

L498-500 Can you say anything about what this finding implies regarding problems with the model simulations?
We just described the changes in smaller magnitude than in the western US (of >10 ppbv), which was discussed in earlier sentences.

L598-601 is confusing as Mexico is usually considered part of North America. From Figure 1 it's clear that it is separate, but this might be worth pointing out here.
We added: "(not included in the NAM source regions, see Figure 1)"

L626 stronger-than-normal-transport from where to where? And compared to what time period (i.e., what is normal?)?
Added: "trans-Pacific". And "stronger-than-normal" was changed to "stronger" to compare 2010 (HTAP2) conditions with 2000/2001 (HTAP1).

L643-644. Given the comparison with observations, is this estimate expected to be too high or better than the other models?
We do not intend to evaluate the sensitivities here.

L659. Why is nonlinear O3 chemistry stronger outside of summer?

As a reflection of seasonal transition of chemical regime. See more discussions in Wu et al. (2009), Fiore et al. (2009), and Brown-Steiner and Hess (2011) on the ozone responses to $NO_x$ or/and NMVOC perturbation results for earlier years.

L697-698, also repeated L741-742 and 915-916. How exactly will the new satellite help capture high O3 and LRT events? Does it have better sensitivity of ozone near the surface?
We emphasize the benefit from future geostationary satellite's better spatial coverage in smaller footprint sizes (2-5 km). The more frequent sampling than the polar-orbiting satellites is also beneficial. Multi-spectral retrievals would have better sensitivity than single-spectral retrievals in general. A more recent paper on TEMPO was now cited.

L734 Are the PBL depths higher during the LRT episodes? This would help to convince the reader of this interpretation.
The sentence near L734 is on PBLH's diurnal variability. This question is also worth some investigation. We correlated the daily daytime mean WRF PBLH with the STEM EAS influences throughout the period (8 May-30 June) to evaluate the relationships between daily variability of PBLH and the EAS source influences. Only in certain regions, we see medium/strong positive correlations ($r>0.5$), where the PBL depths were higher during the LRT episodes, as the correlations may have been complicated by the relationships between PBLHs-local influences. Some earlier sentences in this paragraph were modified to include this finding.

L744-749. It seems the important message here is that EAS typically isn't contributing to the highest days, yet this doesn't come out til 2 paragraphs later and didn't seem to be highlighted in the abstract or conclusions. The exception of the few sites in the southwestern U.S. in L768-772 is also important to note, and it'd be even better if these impacts are quantified in the abstract/conclusions. Are these sites at higher elevation? It might be worth to report separately high elevation sites from the rest of the western sites.
The paragraph starting from L744 describes Figure 10, which is based on modeled exceedances in all grids. The results here are contrasted with Figure 11 in the following paragraphs to show the impacts of spatial coverage and the biases in modeled exceedances on determined R values. As we mentioned near L770, many of the western sites in Figure 11d-f are at high terrain (Figure 2a; regional mean model/actual elevation in Table 3) rural/remote areas where local influences are less dominant, but it does not seem that the differences between the sensitivities on all days and during exceedances are higher at higher elevation sites (e.g., Colorado contrasting with Arizona and Utah sites as shown in Figure 11).

L794-799. Were these captured in the Lin et al. 2012ab studies? Wondering if this applies to all global models or just the ones used here.
Here we draw these conclusions based only on the three boundary condition models used in this study, not including the AM3 model which was used in Lin et al. (2012a,b). All models included in this study (here and throughout the paper) are those that have data available in the AeroCom database submitted following HTAP2 data submission procedure.

At Grand Canyon NP, Lin et al. (2012a,b)'s AM3 results showed positive ozone biases with a moderate model/observation correlation of 0.49 during spring/summer 2010 (See Lin et al., 2012b, Figure 9, upper-right panel) and time-shifted ozone anomaly around 9 May 2010 (See Lin et al.,

2012a, Figure 11, mid panel). However, we do not find evaluation of AM3 O_3 vertical distributions around 9 May in their papers, and the published Lin et al. (2012a,b) results are not based on HTAP2 emission inputs.

L808. Unless the stratospheric contribution was diagnosed in the model, this attribution to stratospheric is speculative and should be clarified as such or deleted. Is the assimilation only assimilating stratospheric ozone or total ozone? Is it somehow tracking stratospheric ozone? "Such as" was changed to "possibly including". MLS O_3 is from UTLS and above; OMI O_3 is total.

L822 should be clearer that it influenced mid-tropospheric ozone over the Northeast. Modified as suggested.

L828-831. Are the mixing depths shallower here and thus don't entrain as much free tropospheric ozone into the surface layer? Yes, it does seem that the modeled PBLHs over these regions during the event are shallower than over the western US in the daytime, so this might be a reason as well. However, please note our reply to your earlier comment on L734 regarding PBLH-EAS/local influences relationships. The related sentence has been modified.

L863-868 Why not discuss the contribution to events here? Contributions during events are discussed in a later paragraph (~L908). This paragraph focuses on the monthly mean results from STEM and its boundary condition models, as well as from the multi- models.

L871 How many global models? Changed to "eight".

L874-875. It's not clear if this bottom up NOx inventory discussion applies globally or to certain regions In global models, globally; in STEM, just within its regional domain.

L885-886. Is this the forward model GEOS-Chem or the adjoint sensitivities? From the GEOS-Chem adjoint model v35f (initially developed from standard GEOS-Chem v8-02-01 with many updates ever since, details at: http://adjoint.colorado.edu/~yanko/gcadj_std/GC_adj_man.pdf), based on the emission perturbation approach. We cited findings from the adjoint sensitivities in Lapina et al. (2014).

L922 and elsewhere – use of "free-running". Does this mean the models are generating their own weather and thus not expected to match specific observations? I think the authors mean simply that they aren't doing chemical data assimilation, but I'm not sure if this is the accepted use of this term. We added: "(i.e., without chemical data assimilation)" for clarity. Also changed in the abstract.

[revised manuscript text omitted]